# An Asymptotically Optimal Primal-Dual Incremental Algorithm for Contextual Linear Bandits

**Andrea Tirinzoni**[*]
Politecnico di Milano
`andrea.tirinzoni@polimi.it`

**Matteo Pirotta**
Facebook AI Research
`pirotta@fb.com`

**Marcello Restelli**
Politecnico di Milano
`marcello.restelli@polimi.it`

**Alessandro Lazaric**
Facebook AI Research
`lazaric@fb.com`

## Abstract

In the contextual linear bandit setting, algorithms built on the optimism principle fail to exploit the structure of the problem and have been shown to be asymptotically suboptimal. In this paper, we follow recent approaches of deriving asymptotically optimal algorithms from problem-dependent regret lower bounds and we introduce a novel algorithm improving over the state-of-the-art along multiple dimensions. We build on a reformulation of the lower bound, where context distribution and exploration policy are decoupled, and we obtain an algorithm robust to unbalanced context distributions. Then, using an incremental primal-dual approach to solve the Lagrangian relaxation of the lower bound, we obtain a scalable and computationally efficient algorithm. Finally, we remove forced exploration and build on confidence intervals of the optimization problem to encourage a minimum level of exploration that is better adapted to the problem structure. We demonstrate the *asymptotic optimality* of our algorithm, while providing both problem-dependent and worst-case finite-time regret guarantees. Our bounds scale with the logarithm of the number of arms, thus avoiding the linear dependence common in all related prior works. Notably, we establish *minimax optimality* for any learning horizon in the special case of non-contextual linear bandits. Finally, we verify that our algorithm obtains better empirical performance than state-of-the-art baselines.

## 1 Introduction

We study the contextual linear bandit (CLB) setting [e.g., 1], where at each time step $t$ the learner observes a context $X_t$ drawn from a context distribution $\rho$, pulls an arm $A_t$, and receives a reward $Y_t$ drawn from a distribution whose expected value is a linear combination between $d$-dimensional features $\phi(X_t, A_t)$ describing context and arm, and an unknown parameter $\theta^\star$. The objective of the learner is to maximize the reward over time, that is to minimize the cumulative regret w.r.t. an optimal strategy that selects the best arm in each context. This setting formalizes a wide range of problems such as online recommendation systems, clinical trials, dialogue systems, and many others [2]. Popular algorithmic principles, such as optimism-in-face-of-uncertainty and Thompson sampling [3], have been applied to this setting leading to algorithms such as OFUL [4] and LINTS [5, 6] with strong finite-time worst-case regret guarantees. Nonetheless, Lattimore & Szepesvari [7] recently showed that these algorithms are not asymptotically optimal (in a problem-dependent sense) as they fail to adapt to the structure of the problem at hand. In fact, in the CLB setting, the values of

---

[*]Work done while at Facebook.

different arms are tightly connected through the linear assumption and a possibly suboptimal arm may provide a large amount of information about $\theta^\star$ and thus the optimal arm. Optimistic algorithms naturally discard suboptimal arms and thus may miss the chance to acquire information about $\theta^\star$ and significantly reduce the regret.

Early attempts to exploit general structures in MAB either adapted UCB-based strategies [8, 9] or focused on different criteria, such as regret to information ratio [10]. While these approaches succeed in improving the finite-time performance of optimism-based algorithms, they still do not achieve asymptotic optimality. An alternative approach to exploit the problem structure was introduced in [7] for (non-contextual) linear bandits. Inspired by approaches for regret minimization [11, 12, 13] and best-arm identification [14] in MAB, Lattimore & Szepesvari [7] proposed to compute an exploration strategy by solving the (estimated) optimization problem characterizing the asymptotic regret lower bound for linear bandits. While the resulting algorithm matches the asymptotic logarithmic lower bound with tight leading constant, it performs rather poorly in practice. Combes et al. [15] followed a similar approach and proposed OSSB, an asymptotically optimal algorithm for bandit problems with general structure (including, e.g., linear, Lipschitz, unimodal). Unfortunately, once instantiated for the linear bandit case, OSSB suffers from poor empirical performance due to the large dependency on the number of arms. Recently, Hao et al. [16] introduced OAM, an asymptotically optimal algorithm for the CLB setting. While OAM effectively exploits the linear structure and outperforms other bandit algorithms, it suffers from major limitations. From an algorithmic perspective, at each exploration step, OAM requires solving the optimization problem of the regret lower bound, which can hardly scale beyond problems with a handful of contexts and arms. Furthermore, OAM implements a forcing exploration strategy that often leads to long periods of linear regret and introduces a linear dependence on the number of arms $|\mathcal{A}|$. Finally, the regret analysis reveals a critical dependence on the inverse of the smallest probability of a context (i.e., $\min_x \rho(x)$), thus suggesting that OAM may suffer from poor finite-time performance in problems with unbalanced context distributions.[2] Degenne et al. [17] recently introduced SPL, which significantly improves over previous algorithms for MAB problems with general structures. Inspired by algorithms for best-arm identification [18], Degenne et al. reformulate the optimization problem in the lower bound as a saddle-point problem and show how to leverage online learning methods to avoid recomputing the exploration strategy from scratch at each step. Furthermore, SPL removes any form of forced exploration by introducing optimism into the estimated optimization problem. As a result, SPL is computationally efficient and achieves better empirical performance in problems with general structures.

**Contributions.** In this paper, we follow similar steps as in [17] and introduce SOLID, a novel algorithm for the CLB setting. Our main contributions can be summarized as follows.

- We first reformulate the optimization problem associated with the lower bound for contextual linear bandits [15, 19, 16] by introducing an additional constraint to guarantee bounded solutions and by explicitly decoupling the context distribution and the exploration policy. While we bound the bias introduced by the constraint, we also illustrate how the resulting exploration policy is better adapted to unbalanced context distributions.

- Leveraging the Lagrangian dual formulation associated with the constrained lower-bound optimization problem, we derive SOLID, an efficient primal-dual learning algorithm that incrementally updates the exploration strategy at each time step. Furthermore, we replace forced exploration with an optimistic version of the optimization problem by specifically leveraging the linear structure of the problem. Finally, SOLID does not require any explicit tracking step and it samples directly from the current exploration strategy.

- We establish the *asymptotic optimality* of SOLID, while deriving a finite-time problem-dependent regret bound that scales only with $\log|\mathcal{A}|$ and without any dependence on $\min_x \rho(x)$. To this purpose, we introduce a new concentration bound for regularized least-squares that scales as $\mathcal{O}(\log t + d \log\log t)$, hence removing the $d \log t$ dependence of the bound in [4]. Moreover, we establish a $\widetilde{\mathcal{O}}(|\mathcal{X}|\sqrt{dn})$ worst-case regret bound for any CLB problem with $|\mathcal{X}|$ contexts, $d$ features, and horizon $n$. Notably, this is implies that SOLID is the first algorithm to be simultaneously *asymptotically optimal* and *minimax optimal* in non-contextual linear bandits.

- We empirically compare to a number of state-of-the-art methods for contextual linear bandits and show how SOLID is more computationally efficient and often has the smallest regret.

A thorough comparison between SOLID and related work is reported in App. B.

## 2 Preliminaries

We consider the contextual linear bandit setting. Let $\mathcal{X}$ be the set of contexts and $\mathcal{A}$ be the set of arms with cardinality $|\mathcal{X}| < \infty$ and $|\mathcal{A}| < \infty$, respectively. Each context-arm pair is embedded into $\mathbb{R}^d$ through a feature map $\phi : \mathcal{X} \times \mathcal{A} \to \mathbb{R}^d$. For any reward model $\theta \in \mathbb{R}^d$, we denote by $\mu_\theta(x,a) = \phi(x,a)^\mathsf{T}\theta$ the expected reward for each context-arm pair. Let $a_\theta^\star(x) := \mathrm{argmax}_{a \in \mathcal{A}} \mu_\theta(x,a)$ and $\mu_\theta^\star(x) := \max_{a \in \mathcal{A}} \mu_\theta(x,a)$ denote the optimal arm and its value for context $x$ and parameter $\theta$. We define the sub-optimality gap of arm $a$ for context $x$ in model $\theta$ as $\Delta_\theta(x,a) := \mu_\theta^\star(x) - \mu_\theta(x,a)$. We assume that every time arm $a$ is selected in context $x$, a random observation $Y = \phi(x,a)^\mathsf{T}\theta + \xi$ is generated, where $\xi \sim \mathcal{N}(0,\sigma^2)$ is a Gaussian noise.[3] Given two parameters $\theta, \theta' \in \mathbb{R}^d$, we define $d_{x,a}(\theta,\theta') := \frac{1}{2\sigma^2}(\mu_\theta(x,a) - \mu_{\theta'}(x,a))^2$, which corresponds to the Kullback-Leibler divergence between the Gaussian reward distributions of the two models in context $x$ and arm $a$.

At each time step $t \in \mathbb{N}$, the learner observes a context $X_t \in \mathcal{X}$ drawn i.i.d. from a context distribution $\rho$, it pulls an arm $A_t \in \mathcal{A}$, and it receives a reward $Y_t = \phi(X_t, A_t)^\mathsf{T}\theta^\star + \xi_t$, where $\theta^\star \in \mathbb{R}^d$ is unknown to the learner. A bandit strategy $\pi := \{\pi_t\}_{t \geq 1}$ chooses the arm $A_t$ to pull at time $t$ as a measurable function $\pi_t(H_{t-1}, X_t)$ of the current context $X_t$ and of the past history $H_{t-1} := (X_1, Y_1, \ldots, X_{t-1}, Y_{t-1})$. The objective is to define a strategy that minimizes the expected cumulative regret over $n$ steps, $\mathbb{E}_{\xi,\rho}^\pi[R_n(\theta)] := \mathbb{E}_{\xi,\rho}^\pi\left[\sum_{t=1}^n \left(\mu_\theta^\star(X_t) - \mu_\theta(X_t, A_t)\right)\right]$, where $\mathbb{E}_{\xi,\rho}^\pi$ denotes the expectation w.r.t. the randomness of contexts, the noise of the rewards, and any randomization in the algorithm. We denote by $\theta^\star$ the reward model of the bandit problem at hand, and without loss of generality we rely on the following regularity assumptions.

**Assumption 1.** *The realizable parameters belong to a compact subset $\Theta$ of $\mathbb{R}^d$ such that $\|\theta\|_2 \leq B$ for all $\theta \in \Theta$. The features are bounded, i.e., $\|\phi(x,a)\|_2 \leq L$ for all $x \in \mathcal{X}, a \in \mathcal{A}$. The context distribution is supported over the whole context set, i.e., $\rho(x) \geq \rho_{\min} > 0$ for all $x \in \mathcal{X}$. Finally, w.l.o.g. we assume $\theta^\star$ has a unique optimal arm in each context [see e.g., 15, 16].*

**Regularized least-squares estimator.** We introduce the regularized least-square estimate of $\theta^\star$ using $t$ samples as $\widehat{\theta}_t := \overline{V}_t^{-1} U_t$, where $\overline{V}_t := \sum_{s=1}^t \phi(X_s, A_s)\phi(X_s, A_s)^\mathsf{T} + \nu I$, with $\nu \geq \max\{L^2, 1\}$ and $I$ the $d \times d$ identity matrix, and $U_t := \sum_{s=1}^t \phi(X_s, A_s)Y_s$. The estimator $\widehat{\theta}_t$ satisfies the following concentration inequality (see App. J for the proof and exact formulation).

**Theorem 1.** *Let $\delta \in (0,1)$, $n \geq 3$, and $\widehat{\theta}_t$ be a regularized least-square estimator obtained using $t \in [n]$ samples collected using an arbitrary bandit strategy $\pi := \{\pi_t\}_{t \geq 1}$. Then,*

$$\mathbb{P}\left\{\exists t \in [n] : \|\widehat{\theta}_t - \theta^\star\|_{\overline{V}_t} \geq \sqrt{c_{n,\delta}}\right\} \leq \delta,$$

*where $c_{n,\delta}$ is of order $\mathcal{O}(\log(1/\delta) + d \log\log n)$.*

For the usual choice $\delta = 1/n$, $c_{n,1/n}$ is of order $\mathcal{O}(\log n + d \log\log n)$, which illustrates how the dependency on $d$ is on a lower-order term w.r.t. $n$ (as opposed to the well-known concentration bound derived in [4]). This result is the counterpart of [7, Thm. 8] for the concentration on the reward parameter estimation error instead of the prediction error and we believe it is of independent interest.

## 3 Lower Bound

We recall the asymptotic lower bound for multi-armed bandit problems with structure from [20, 15, 19]. We say that a bandit strategy $\pi$ is *uniformly good* if $\mathbb{E}_{\xi,\rho}^\pi[R_n] = o(n^\alpha)$ for any $\alpha > 0$ and any contextual linear bandit problem satisfying Asm. 1.

**Proposition 1.** *Let $\pi := \{\pi_t\}_{t \geq 1}$ by a uniformly good bandit strategy then,*

$$\liminf_{n \to \infty} \frac{\mathbb{E}_{\xi,\rho}^\pi[R_n(\theta^\star)]}{\log(n)} \geq v^\star(\theta^\star), \tag{1}$$

*where $v^\star(\theta^\star)$ is the value of the optimization problem*

$$\inf_{\eta(x,a)\geq 0} \quad \sum_{x\in\mathcal{X}}\sum_{a\in\mathcal{A}}\eta(x,a)\Delta_{\theta^\star}(x,a) \quad \text{s.t.} \quad \inf_{\theta'\in\Theta_{\text{alt}}}\sum_{x\in\mathcal{X}}\sum_{a\in\mathcal{A}}\eta(x,a)d_{x,a}(\theta^\star,\theta')\geq 1, \quad \text{(P)}$$

*where $\Theta_{\text{alt}} := \{\theta'\in\Theta \mid \exists x\in\mathcal{X}, a_{\theta^\star}^\star(x)\neq a_{\theta'}^\star(x)\}$ is the set of alternative reward parameters such that the optimal arm changes for at least a context $x$.[4]*

The variables $\eta(x,a)$ can be interpreted as the number of pulls allocated to each context-arm pair so that enough information is obtained to correctly identify the optimal arm in each context while minimizing the regret. Formulating the lower bound in terms of the solution of (P) is not desirable for two main reasons. First, (P) is not a well-posed optimization problem since the inferior may not be attainable, i.e., the optimal solution may allocate an infinite number of pulls to some optimal arms. Second, (P) removes any dependency on the context distribution $\rho$. In fact, the optimal solution $\eta^\star$ of (P) may prescribe to select a context-arm $(x,a)$ pair a large number of times, despite $x$ having low probability of being sampled from $\rho$. While this has no impact on the asymptotic performance of $\eta^\star$ (as soon as $\rho_{\min} > 0$), building on $\eta^\star$ to design a learning algorithm may lead to poor finite-time performance. In order to mitigate these issues, we propose a variant of the previous lower bound obtained by adding a constraint on the cumulative number of pulls in each context and explicitly decoupling the context distribution $\rho$ and the *exploration policy* $\omega(x,a)$ defining the probability of selecting arm $a$ in context $x$. Given $z\in\mathbb{R}_{>0}$, we define the optimization problem

$$\min_{\omega\in\Omega} \quad z\mathbb{E}_\rho\left[\sum_{a\in\mathcal{A}}\omega(x,a)\Delta_{\theta^\star}(x,a)\right] \quad \text{s.t.} \quad \inf_{\theta'\in\Theta_{\text{alt}}}\mathbb{E}_\rho\left[\sum_{a\in\mathcal{A}}\omega(x,a)d_{x,a}(\theta^\star,\theta')\right]\geq 1/z \quad (\text{P}_z)$$

where $\Omega = \{\omega(x,a)\geq 0 \mid \forall x\in\mathcal{X}: \sum_{a\in\mathcal{A}}\omega(x,a) = 1\}$ is the probability simplex. We denote by $\omega_{z,\theta^\star}^\star$ the optimal solution of $(\text{P}_z)$ and $u^\star(z,\theta^\star)$ its associated value (if the problem is unfeasible we set $u^\star(z,\theta^\star) = +\infty$). Inspecting $(\text{P}_z)$, we notice that $z$ serves as a global constraint on the number of samples. In fact, for any $\omega\in\Omega$, the associated number of samples $\eta(x,a)$ allocated to a context-arm pair $(x,a)$ is now $z\rho(x)\omega(x,a)$. Since $\rho$ is a distribution over $\mathcal{X}$ and $\sum_a\omega(x,a) = 1$ in each context, the total number of samples sums to $z$. As a result, $(\text{P}_z)$ admits a minimum and it is more amenable to designing a learning algorithm based on its Lagrangian relaxation. Furthermore, we notice that $z$ can be interpreted as defining a more "finite-time" formulation of the lower bound. Finally, we remark that the total number of samples that can be assigned to a context $x$ is indeed constrained to $z\rho(x)$. This constraint crucially makes $(\text{P}_z)$ more context aware and forces the solution $\omega$ to be more adaptive to the context distribution. In Sect. 4, we leverage these features to design an incremental algorithm whose finite-time regret does not depend on $\rho_{\min}$, thus improving over previous algorithms [7, 16], as supported by the empirical results in Sect. 6. The following lemma provides a characterization of $(\text{P}_z)$ and its relationship with (P) (see App. C for the proof and further discussion).

**Lemma 1.** *Let* $\underline{z}(\theta^\star) := \min\{z > 0 : (\text{P}_z) \text{ is feasible}\}$, $\overline{z}(\theta^\star) = \max_{x\in\mathcal{X}}\sum_{a\neq a_{\theta^\star}^\star(x)}\frac{\eta^\star(x,a)}{\rho(x)}$ *and* $z^\star(\theta^\star) := \sum_{x\in\mathcal{X}}\sum_{a\neq a_{\theta^\star}^\star(x)}\eta^\star(x,a)$. *Then* $\frac{1}{\underline{z}(\theta^\star)} = \max_{\omega\in\Omega}\inf_{\theta'\in\Theta_{\text{alt}}}\mathbb{E}_\rho\left[\sum_{a\in\mathcal{A}}\omega(x,a)d_{x,a}(\theta^\star,\theta')\right]$ *and there exists a constant $c_\Theta > 0$ such that, for any $z\in(\underline{z}(\theta^\star), +\infty)$,*

$$u^\star(z,\theta^\star) \leq v^\star(\theta^\star) + \frac{2zBL\underline{z}(\theta^\star)}{z - \underline{z}(\theta^\star)}\cdot\begin{cases}1 & \text{if } z < \overline{z}(\theta^\star)\\ \min\left\{\max\left\{\frac{c_\Theta\sqrt{2}z^\star(\theta^\star)}{\sigma\sqrt{z}}, \frac{z^\star(\theta^\star)}{z}\right\}, 1\right\} & \text{otherwise}\end{cases}$$

The first result characterizes the range of $z$ for which $(\text{P}_z)$ is feasible. Interestingly, $\underline{z}(\theta^\star) < +\infty$ is the inverse of the sample complexity of the best-arm identification problem [21] and the associated solution is the one that maximizes the amount of information gathered about the reward model $\theta^\star$. As $z$ increases, $\omega_{z,\theta^\star}^\star$ becomes less aggressive in favoring informative context-arm pairs and more sensitive to the regret minimization objective. The second result quantifies the bias w.r.t. the optimal solution of $(\text{P}_z)$. For $z\geq\overline{z}(\theta^\star)$, the error decreases approximately at a rate $1/\sqrt{z}$ showing that the solution of $(\text{P}_z)$ can be made arbitrarily close to $v^\star(\theta^\star)$.

In designing our learning algorithm, we build on the Lagrangian relaxation of $(P_z)$. For any $\omega \in \Omega$, let $f(\omega; \theta^\star)$ denote the objective function and $g(\omega, z; \theta^\star)$ denote the KL constraint

$$f(\omega; \theta^\star) = \mathbb{E}_\rho \Big[ \sum_{a \in \mathcal{A}} \omega(x, a) \mu_{\theta^\star}(x, a) \Big], \quad g(\omega; z, \theta^\star) = \inf_{\theta' \in \Theta_{\mathrm{alt}}} \mathbb{E}_\rho \Big[ \sum_{a \in \mathcal{A}} \omega(x, a) d_{x, a}(\theta^\star, \theta') \Big] - \frac{1}{z}.$$

We introduce the Lagrangian relaxation problem

$$\min_{\lambda \geq 0} \max_{\omega \in \Omega} \Big\{ h(\omega, \lambda; z, \theta^\star) := f(\omega; \theta^\star) + \lambda g(\omega; z, \theta^\star) \Big\}, \tag{$P_\lambda$}$$

where $\lambda \in \mathbb{R}_{\geq 0}$ is a multiplier. Notice that $f(\omega; \theta^\star)$ is not equal to the objective function of $(P_z)$, since we replaced the gap $\Delta_{\theta^\star}$ by the expected value $\mu_{\theta^\star}$ and we removed the constant multiplicative factor $z$ in the objective function. The associated problem is thus a concave maximization problem. While these changes do not affect the optimality of the solution, they do simplify the algorithmic design. Refer to App. D for details about the Lagrangian formulation.

## 4 Asymptotically Optimal Linear Primal Dual Algorithm

We introduce SOLID (aSymptotic Optimal Linear prImal Dual), which combines a primal-dual approach to incrementally compute the solution of an optimistic estimate of the Lagrangian relaxation $(P_\lambda)$ within a scheme that, depending on the accuracy of the estimate $\widehat{\theta}_t$, separates *exploration* steps, where arms are pulled according to the exploration policy $\omega_t$, and *exploitation* steps, where the greedy arm is selected. The values of the input parameters for which SOLID enjoys regret guarantees are reported in Sect. 5. In the following, we detail the main ingredients composing the algorithm (see Alg. 1).

**Estimation.** SOLID stores and updates the regularized least-square estimate $\widehat{\theta}_t$ using all samples observed over time. To account for the fact that $\widehat{\theta}_t$ may have large norm (i.e., $\|\widehat{\theta}_t\|_2 > B$ and $\widehat{\theta}_t \notin \Theta$), SOLID explicitly projects $\widehat{\theta}_t$ onto $\Theta$. Formally, let $\mathcal{C}_t := \{\theta \in \mathbb{R}^d : \|\theta - \widehat{\theta}_t\|_{\overline{V}_t}^2 \leq \beta_t\}$ be the confidence ellipsoid at time $t$. Then, SOLID computes $\widetilde{\theta}_t := \mathrm{argmin}_{\theta \in \Theta \cap \mathcal{C}_t} \|\theta - \widehat{\theta}_t\|_{\overline{V}_t}^2$. This is a simple convex optimization problem, though it has no closed-form expression.[5] Note that, on those steps where $\theta^\star \notin \mathcal{C}_t$, $\Theta \cap \mathcal{C}_t$ might be empty, in which case we can set $\widetilde{\theta}_t = \widetilde{\theta}_{t-1}$. Then, SOLID uses $\widetilde{\theta}_t$ instead of $\widehat{\theta}_t$ in all steps of the algorithm. SOLID also computes an empirical estimate of the context distribution as $\widehat{\rho}_t(x) = \frac{1}{t} \sum_{s=1}^t \mathbb{1}\{X_s = x\}$.

---

**Algorithm 1:** SOLID

**Input:** Multiplier $\lambda_1$, confidence values $\{\beta_t\}_t$ and $\{\gamma_t\}_t$, maximum multiplier $\lambda_{\max}$, normalization factors $\{z_k\}_{k \geq 0}$, phase lengths $\{p_k\}_{k \geq 0}$, step sizes $\alpha_k^\lambda, \alpha_k^\omega$

Set $\omega_1 \leftarrow \frac{1_{\mathcal{X} \mathcal{A}}}{|\mathcal{A}|}, \overline{V}_0 \leftarrow \nu \boldsymbol{I}, U_0 \leftarrow \boldsymbol{0}, \widetilde{\theta}_0 \leftarrow \boldsymbol{0}, S_0 \leftarrow 0$

Phase index: $K_1 \leftarrow 0$

**for** $t = 1, \ldots, n$ **do**
    Receive context $X_t \sim \rho$
    Set $K_{t+1} \leftarrow K_t$
    **if** $\inf_{\theta' \in \overline{\Theta}_{t-1}} \|\widetilde{\theta}_{t-1} - \theta'\|_{\overline{V}_{t-1}}^2 > \beta_{t-1}$ **then**
        // EXPLOITATION STEP
        $A_t \leftarrow \mathrm{argmax}_{a \in \mathcal{A}} \mu_{\widetilde{\theta}_{t-1}}(X_t, a)$
        $\lambda_{t+1} \leftarrow \lambda_t, \omega_{t+1} \leftarrow \omega_t$
    **else**
        // EXPLORATION STEP
        Sample arm: $A_t \sim \omega_t(X_t, \cdot)$
        Set $S_t \leftarrow S_{t-1} + 1$
        // UPDATE SOLUTION
        Compute $q_t \in \partial h_t(\omega_t, \lambda_t, z_{K_t})$ (see Eq. 4)
        Update policy
$$\omega_{t+1}(x, a) \leftarrow \frac{\omega_t(x, a) e^{\alpha_{K_t}^\omega q_t(x, a)}}{\sum_{a' \in \mathcal{A}} \omega_t(x, a') e^{\alpha_{K_t}^\omega q_t(x, a')}}$$
        Update multiplier
        $\lambda_{t+1} \leftarrow \min\{[\lambda_t - \alpha_{K_t}^\lambda g_t(\omega_t, z_{K_t})]_+, \lambda_{\max}\}$
        // PHASE STOPPING TEST
        **if** $S_t - S_{T_{K_t} - 1} = p_k$ **then**
            Change phase: $K_{t+1} \leftarrow K_t + 1$
            Reset solution: $\omega_{t+1} \leftarrow \omega_1, \lambda_{t+1} \leftarrow \lambda_1$
    Pull $A_t$ and observe outcome $Y_t$
    Update $\overline{V}_t, U_t, \widehat{\theta}_t, \widehat{\rho}_t$ using $X_t, A_t, Y_t$
    Set $\widetilde{\theta}_t := \mathrm{argmin}_{\theta \in \Theta \cap \mathcal{C}_t} \|\theta - \widehat{\theta}_t\|_{\overline{V}_t}$

---

**Accuracy test and tracking.** Similar to previous algorithms leveraging asymptotic lower bounds, we build on the generalized likelihood ratio test [e.g., 18] to verify the accuracy of the estimate $\widehat{\theta}_t$. At the beginning of each step $t$, SOLID first computes $\inf_{\theta' \in \overline{\Theta}_{t-1}} \|\widetilde{\theta}_{t-1} - \theta'\|_{\overline{V}_{t-1}}^2$, where $\overline{\Theta}_{t-1} = \{\theta' \in \Theta \mid \exists x \in \mathcal{X}, a_{\widehat{\theta}_{t-1}}^\star(x) \neq a_{\theta'}^\star(x)\}$ is the set of alternative models. This quantity

measures the accuracy of the algorithm, where the infimum over alternative models defines the problem $\theta'$ that is closest to $\widetilde{\theta}_{t-1}$ and yet different in the optimal arm of at least one context.[6] This serves as a worst-case scenario for the true $\theta^\star$, since if $\theta^* = \theta'$ then selecting arms according to $\widetilde{\theta}_{t-1}$ would lead to linear regret. If the accuracy exceeds a threshold $\beta_{t-1}$, then SOLID performs an exploitation step, where the estimated optimal arm $a^\star_{\widetilde{\theta}_{t-1}}(X_t)$ is selected in the current context.

On the other hand, if the test fails, the algorithm moves to an exploration step, where an arm $A_t$ is sampled according to the estimated exploration policy $\omega_t(X_t, \cdot)$. While this approach is considerably simpler than standard tracking strategies (e.g., selecting the arm with the largest gap between the policy $\omega_t$ and the number of pulls), in Sect. 5 we show that sampling from $\omega_t$ achieves the same level of tracking efficiency.

**Optimistic primal-dual subgradient descent.** At each step $t$, we define an estimated optimistic version of the Lagrangian relaxation (P$_\lambda$) as

$$f_t(\omega) := \sum_{x \in \mathcal{X}} \widehat{\rho}_{t-1}(x) \sum_{a \in \mathcal{A}} \omega(x, a) \left( \mu_{\widetilde{\theta}_{t-1}}(x, a) + \sqrt{\gamma_t} \|\phi(x, a)\|_{\overline{V}_{t-1}^{-1}} \right), \tag{2}$$

$$g_t(\omega, z) := \inf_{\theta' \in \overline{\Theta}_{t-1}} \sum_{x \in \mathcal{X}} \widehat{\rho}_{t-1}(x) \sum_{a \in \mathcal{A}} \omega(x, a) \left( d_{x,a}(\widetilde{\theta}_{t-1}, \theta') + \frac{2BL}{\sigma^2} \sqrt{\gamma_t} \|\phi(x, a)\|_{\overline{V}_{t-1}^{-1}} \right) - \frac{1}{z}, \tag{3}$$

$$h_t(\omega, \lambda, z) := f_t(\omega) + \lambda g_t(\omega, z), \tag{4}$$

where $\gamma_t$ is a suitable parameter defining the size of the confidence interval.

Notice that we do not use optimism on the context distribution, which is simply replaced by its empirical estimate. Therefore, $h_t$ is not necessarily optimistic with respect to the original Lagrangian function $h$. Nonetheless, we prove in Sect. 5 that this level of optimism is sufficient to induce enough exploration to have accurate estimates of $\theta^\star$. This is in contrast with the popular forced exploration strategy [e.g. 7, 15, 19, 16], which prescribes a minimum fraction of pulls $\epsilon$ such that at any step $t$, any of the arms with less than $\epsilon S_t$ pulls is selected, where $S_t$ is the number of exploration rounds so far. While this strategy is sufficient to guarantee a minimum level of accuracy for $\widehat{\theta}_t$ and to obtain asymptotic regret optimality, in practice it is highly inefficient as it requires selecting all arms in each context regardless of their value or amount of information.

At each step $t$, SOLID updates the estimates of the optimal exploration policy $\omega_t$ and the Lagrangian multiplier $\lambda_t$. In particular, given the sub-gradient $q_t$ of $h_t(\omega_t, \lambda_t, z_{K_t})$, SOLID updates $\omega_t$ and $\lambda_t$ by performing one step of projected sub-gradient descent with suitable learning rates $\alpha^\omega_{K_t}$ and $\alpha^\lambda_{K_t}$. In the update of $\omega_t$, we perform the projection onto the simplex $\Omega$ using an entropic metric, while the multiplier is clipped in $[0, \lambda_{\max}]$. While this is a rather standard primal-dual approach to solve the Lagrangian relaxation (P$_\lambda$), the interplay between estimates $\widehat{\theta}_t$, $\rho_t$, the optimism used in $h_t$, and the overall regret performance of the algorithm is at the core of the analysis in Sect. 5.

This approach significantly reduces the computational complexity compared to [15, 16], which require solving problem P at each exploratory step. In Sect. 6, we show that the incremental nature of SOLID allows it to scale to problems with much larger context-arm spaces. Furthermore, we leverage the convergence rate guarantees of the primal-dual gradient descent to show that the incremental nature of SOLID does not compromise the asymptotic optimality of the algorithm (see Sect. 5).

**The $z$ parameter.** While the primal-dual algorithm is guaranteed to converge to the solution of (P$_z$) for any fix $z$, it may be difficult to properly tune $z$ to control the error w.r.t. (P). SOLID leverages the fact that the error scales as $1/\sqrt{z}$ (Lem. 1 for $z$ sufficiently large) and it increases $z$ over time. Given as input two non-decreasing sequences $\{p_k\}_k$ and $\{z_k\}_k$, at each phase $k$, SOLID uses $z_k$ in the computation of the subgradient of $h_t$ and in the definition of $f_t$ and $g_t$. After $p_k$ explorative steps, it resets the policy $\omega_t$ and the multiplier $\lambda_t$ and transitions to phase $k+1$. Since $p_k = S_{T_{k+1}-1} - S_{T_k-1}$ is the number of *explorative* steps of phase $k$ starting at time $T_k$, the actual number of steps during $k$ may vary. Notice that at the end of each phase only the optimization variables are reset, while the learning variables (i.e., $\widehat{\theta}_t$, $\overline{V}_t$, and $\widehat{\rho}_t$) use all the samples collected through phases.

## 5 Regret Analysis

Before reporting the main theoretical result of the paper, we introduce the following assumption.

**Assumption 2.** *The maximum multiplier used by* SOLID *is such that $\lambda_{\max} \geq 2BL\underline{z}(\theta^\star)$.*

While an assumption on the maximum multiplier is rather standard for the analysis of primal-dual projected subgradient [e.g., 22, 23], we conjecture that it may be actually relaxed in our case by replacing the fixed $\lambda_{\max}$ by an increasing sequence as done for $\{z_k\}_k$.

**Theorem 2.** *Consider a contextual linear bandit problem with contexts $\mathcal{X}$, arms $\mathcal{A}$, reward parameter $\theta^\star$, features bounded by $L$, zero-mean Gaussian noise with variance $\sigma^2$ and context distribution $\rho$ satisfying Asm. 1. If* SOLID *is run with confidence values $\beta_{t-1} = c_{n,1/n}$ and $\gamma_t = c_{n,1/S_t^2}$, where $c_{n,\delta}$ is defined as in Thm. 1, learning rates $\alpha_k^\lambda = \alpha_k^\omega = 1/\sqrt{p_k}$ and increasing sequences $z_k = z_0 e^k$ and $p_k = z_k e^{2k}$, for some $z_0 \geq 1$, then it is **asymptotically optimal** with the same constant as in the lower bound of Prop. 1. Furthermore, for any finite $n$ the regret of* SOLID *is bounded as*

$$\mathbb{E}^\pi_{\xi,\rho}\big[R_n(\theta^\star)\big] \leq v^\star(\theta^\star)\frac{c_{n,1/n}}{2\sigma^2} + C_{\log}(\log\log n)^{\frac{1}{2}}(\log n)^{\frac{3}{4}} + C_{\text{const}}, \qquad (5)$$

*where $C_{\log} = lin_{\geq 0}(v^\star(\theta^\star), |\mathcal{X}|, L^2, B^2, \sqrt{d}, 1/\sigma^2)$ and $C_{\text{const}} = v^\star(\theta^\star)\frac{B^2 L^2}{\sigma^2} + lin_{\geq 0}(L, B, z_0(\underline{z}(\theta^\star)/z_0)^3, (\overline{z}(\theta^\star)/z_0)^2).$*[7]

The first result shows that SOLID run with an exponential schedule for $z$ is asymptotic optimal, while the second one provides a bound on the finite-time regret. We can identify three main components in the finite-time regret. **1)** The first term scales with the logarithmic term $c_{n,1/n} = O(\log n + d \log \log n)$ and a leading constant $v^\star(\theta^\star)$, which is optimal as shown in Prop. 1. In most cases, this is the dominant term of the regret. **2)** Lower-order terms in $o(\log n)$. Notably, a regret of order $\sqrt{\log n}$ is due to the incremental nature of SOLID and it is directly inherited from the convergence rate of the primal-dual algorithm we use to optimize $(P_z)$. The larger term $(\log n)^{3/4}$ that we obtain in the final regret is actually due to the schedule of $\{z_k\}$ and $\{p_k\}$. While it is possible to design a different phase schedule to reduce the exponent towards $1/2$, this would negatively impact the constant regret term. **3)** The constant regret $C_{\text{const}}$ is due to the exploitation steps, burn-in phase and the initial value $z_0$. The regret due to $z_0$ takes into account the regime when $(P_z)$ is unfeasible $(z_k < \underline{z}(\theta^\star))$ or when $z_k$ is too small to assess the rate at which $u^\star(z_k, \theta^\star)$ approaches $v^\star(\theta^\star)$ $(z < \overline{z}(\theta^\star))$, see Lem. 1. Notably, the regret due to the initial value $z_0$ vanishes when $z_0 > \overline{z}(\theta^\star)$. A more aggressive schedule for $z_k$ reaching $\overline{z}(\theta^\star)$ in few phases would reduce the initial regret at the cost of a larger exponent in the sub-logarithmic terms.

The sub-logarithmic terms in the regret have only logarithmic dependency on the number of arms. This is better than existing algorithms based on exploration strategies built from lower bounds. OSSB [15] indeed depends on $|\mathcal{A}|$ directly in the main $\mathcal{O}(\log n)$ regret terms. While the regret analysis of OAM is asymptotic, it is possible to identify several lower-order terms depending linearly on $|\mathcal{A}|$. In fact, OAM as well as OSSB require forced exploration on each context-arm pair, which inevitably translates into regret. In this sense, the dependency on $|\mathcal{A}|$ is hard-coded into the algorithm and cannot be improved by a better analysis. SPL depends linearly on $|\mathcal{A}|$ in the explore/exploit threshold (the equivalent of our $\beta_t$) and in other lower-order terms due to the analysis of the tracking rule. On the other hand, SOLID never requires all arms to be repeatedly pulled and we were able to remove the linear dependence on $|\mathcal{A}|$ through a refined analysis of the sampling procedure (see App. E). This is inline with the experimental results where we did not notice any explicit linear dependence on $|\mathcal{A}|$.

The constant regret term depends on the context distribution through $\overline{z}(\theta^\star)$ (Lem. 1). Nonetheless, this dependency disappears whenever $z_0$ is a fraction $\overline{z}(\theta^\star)$. This is in striking contrast with OAM, whose analysis includes several terms depending on the inverse of the context probability $\rho_{\min}$. This confirms that SOLID is able to better adapt to the distribution generating the contexts. While the phase schedule of Thm. 2 leads to an asymptotically-optimal algorithm and sublinear-regret in finite time, it may be possible to find a different schedule having the same asymptotic performance and better finite-time guarantees, although this may depend on the horizon $n$. Refer to App. G.3 for a regret bound highlighting the explicit dependence on the sequences $\{z_k\}$ and $\{p_k\}$.

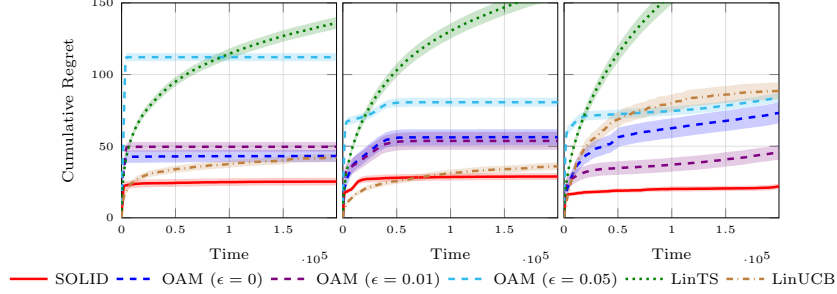

Figure 1: Toy problem with 2 contexts and (left) $\rho(x_1) = .5$, (center) $\rho(x_1) = .9$, (right) $\rho(x_1) = .99$.

As shown in [16], when the features of the optimal arms span $\mathbb{R}^d$, the asymptotic lower bound vanishes (i.e., $v^\star(\theta^\star) = 0$). In this case, selecting optimal arms is already informative enough to correctly estimate $\theta^\star$ and no explicit exploration is needed and SOLID, like OAM, has sub-logarithmic regret.

**Worst-case analysis.** The constant terms in Thm. 2 are due to a naive bound which assumes linear regret in those phases where $z_k$ is small (e.g., when the optimization problem is infeasible). While this simplifies the analysis for asymptotic optimality, we verify that SOLID always suffers sub-linear regret, regardless of the values of $z_k$. For the following result, we do not require Asm. 2 to hold.

**Theorem 3** (Worst-case regret bound). *Let $z_k$ be arbitrary, $p_k = e^{rk}$ for some constant $r \geq 1$, and the other parameters be the same as in Thm. 2. Then, for any $n$ the regret of* SOLID *is bounded as*

$$\mathbb{E}^\pi_{\xi,\rho}\big[R_n(\theta^\star)\big] \leq 3BL\pi^2\left(4 + \frac{\lambda_{\max}BL}{\sigma^2}\right) + \frac{2e^r\lambda_{\max}^2}{r}\sqrt{n} + C_{\text{sqrt}}\left(1 + \frac{\lambda_{\max}BL}{\sigma^2}\right)\log(n)\sqrt{n},$$

*where $C_{\text{sqrt}} = lin_{\geq 0}(|\mathcal{X}|, \sqrt{d}, B, L)$.*

Notably, this bound removes the dependencies on $\underline{z}(\theta^\star)$ and $\overline{z}(\theta^\star)$, while its derivation is agnostic to the values of $z_k$. Interestingly, we could set $\lambda_{\max} = 0$ and the algorithm would completely ignore the KL constraint, thus focusing only on the objective function. This is reflected in the worst-case bound since all terms with a dependence on $\sigma^2$ or a quadratic dependence on $BL$ disappear. The key result is that the objective function alone, thanks to optimism, is sufficient for proving sub-linear regret but not for proving asymptotic optimality. More precisely, the resulting bound is $\widetilde{\mathcal{O}}(|\mathcal{X}|\sqrt{nd})$, which matches the minimax optimal rate apart from the dependence on $|\mathcal{X}|$. The latter could be reduced to $\sqrt{|\mathcal{X}|}$ by a better analysis. It remains an open question how to design an asymptotically optimal algorithm for the contextual case whose regret does not scale with $|\mathcal{X}|$.

## 6 Numerical Simulations

We compare SOLID to LinUCB, LinTS, and OAM. For SOLID, we set $\beta_t = \sigma^2(\log(t) + d\log\log(n))$ and $\gamma_t = \sigma^2(\log(S_t) + d\log\log(n))$ (i.e., we remove all numerical constants) and we use the exponential schedule for phases defined in Thm. 2. For OAM, we use the same $\beta_t$ for the explore/exploit test and we try different values for the forced-exploration parameter $\epsilon$. LinUCB uses the confidence intervals from Thm. 2 in [4] with the log-determinant of the design matrix, and LinTS is as defined in [5] but without the extra-sampling factor $\sqrt{d}$ used to prove its frequentist regret. All plots are the results of 100 runs with 95% Student's t confidence intervals. See App. K for additional details and results on a real dataset.

**Toy contextual linear bandit with structure.** We start with a CLB problem with $|\mathcal{X}| = 2$ and $|\mathcal{A}|, d = 3$. Let $x_i$ ($a_i$) be the $i$-th context (arm). We have $\phi(x_1, a_1) = [1, 0, 0]$, $\phi(x_1, a_2) = [0, 1, 0]$, $\phi(x_1, a_3) = [1 - \xi, 2\xi, 0]$, $\phi(x_2, a_1) = [0, 0.6, 0.8]$, $\phi(x_2, a_2) = [0, 0, 1]$, $\phi(x_2, a_3) = [0, \xi/10, 1 - \xi]$ and $\theta^\star = [1, 0, 1]$. We consider a balanced context distribution $\rho(x_1) = \rho(x_2) = 0.5$. This is a two-context counterpart of the example presented by [7] to show the asymptotic sub-optimality of optimism-based strategies. The intuition is that, for $\xi$ small, an optimistic strategy pulls $a_2$ in $x_1$ and $a_1$ in $x_2$ only a few times since their gap is quite large, and suffers high regret (inversely proportional to $\xi$) to figure out which of the remaining arms is optimal. On the other hand, an asymptotically optimal strategy allocates more pulls to "bad" arms as they bring information to identify $\theta^\star$, which in

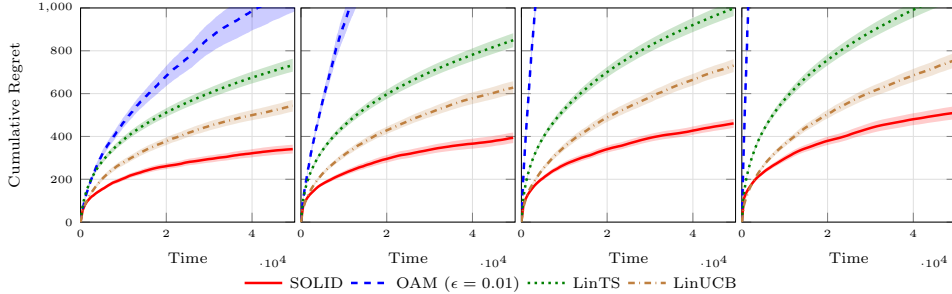

Figure 2: Randomly generated bandit problems with $d = 8, |\mathcal{X}| = 4$, and $|\mathcal{A}| = 4, 8, 16, 32$.

turns avoids a regret scaling with $\xi$. This indeed translates into the empirical performance reported in Fig. 1-(left), where SOLID effectively exploits the structure of the problem and significantly reduces the regret compared to LinTS and LinUCB. Actually, not only the regret is smaller but the "trend" is better. In fact, the regret curves of LinUCB and LinTS have a larger slope than SOLID's, suggesting that the gap may increase further with $n$, thus confirming the theoretical finding that the asymptotic performance of SOLID is better. OAM has a similar behavior, but the actual performance is worse than SOLID and it seems to be very sensitive to the forced exploration parameter, where the best performance is obtained for $\epsilon = 0.0$, which is not theoretically justified.

We also study the influence of the context distribution. We first notice that solving (P) leads to an optimal exploration strategy $\eta^\star$ where the only sub-optimal arm with non-zero pulls is $a_1$ in $x_2$ since it yields lower regret and similar information than $a_2$ in $x_1$. This means that the lower bound prescribes a greedy policy in $x_1$, deferring exploration to $x_2$ alone. In practice, tracking this optimal allocation might lead to poor finite-time performance when the context distribution is unbalanced towards $x_1$, in which case the algorithm would take time proportional to $1/\rho(x_2)$ before performing any meaningful exploration. We verify these intuitions empirically by considering the case of $\rho(x_1) = 0.9$ and $\rho(x_1) = 0.99$ (middle and right plots in Fig. 1 respectively). SOLID is consistently better than all other algorithms, showing that its performance is not negatively affected by $\rho_{\min}$. On the other hand, OAM is more severely affected by the context distribution. In particular, its performance with $\epsilon = 0$ significantly decreases when increasing $\rho(x_1)$ and the algorithm reduces to an almost greedy strategy, thus suffering linear regret in some problems. In this specific case, forcing exploration leads to slightly better finite-time performance since the algorithm pulls the informative arm $a_2$ in $x_1$, which is however not prescribed by the lower bound.

**Random problems.** We evaluate the impact of the number of actions $|\mathcal{A}|$ in randomly generated structured problems with $d = 8$ and $|\mathcal{X}| = 4$. We run each algorithm for $n = 50000$ steps. For OAM, we set forced-exploration $\epsilon = 0.01$ and solve (P) every 100 rounds to speed-up execution as computation becomes prohibitive. The plots in Fig. 2 show the regret over time for $|\mathcal{A}| = 4, 8, 16, 32$. This test confirms the advantage of SOLID over the other methods. Interestingly, the regret of SOLID does not seem to significantly increase as a function of $|\mathcal{A}|$, thus supporting its theoretical analysis. On the other hand, the regret of OAM scales poorly with $|\mathcal{A}|$ since forced exploration pulls all arms in a round robin fashion.

# 7 Conclusion

We introduced SOLID, a novel asymptotically-optimal algorithm for contextual linear bandits with finite-time regret and computational complexity improving over similar methods and better empirical performance w.r.t. state-of-the-art algorithms in our experiments. The main open question is whether SOLID is minimax optimal for contextual problems with $|\mathcal{X}| > 1$. In future work, our method could be extended to continuous contexts, which would probably require a reformulation of the lower bound and the adoption of parametrized policies. Furthermore, it would be interesting to study finite-time lower bounds, especially for problems in which bounded regret is achievable [9, 24, 25]. Finally, we could use algorithmic ideas similar to SOLID to go beyond the realizable linear bandit setting.

## Broader Impact

This work is mainly a theoretical contribution. We believe it does not present any foreseeable societal consequence.

## Funding Transparency Statement

Marcello Restelli was partially funded by the Italian MIUR PRIN 2017 Project ALGADIMAR "Algorithms, Games, and Digital Market".

## Acknowledgements

The authors would like to thank Rémy Degenne, Han Shao, and Wouter Koolen for kindly sharing the draft of their paper before publication. We also would like to thank Pierre Ménard for carefully reading the paper and for providing insightful feedback.

## Footnotes

[2]Interestingly, Hao et al. [16] explicitly mention in their conclusions the importance of properly managing the context distribution to achieve satisfactory finite-time performance.

[3]This assumption can be relaxed by considering sub-Gaussian rewards.

[4] The infimum over this set can be computed in closed-form when the alternative parameters are allowed to lie in the whole $\mathbb{R}^d$ (see App. K.1). When these parameters are forced to have bounded $\ell_2$-norm, the infimum has no closed-form expression, though its computation reduces to a simple convex optimization problem (see [21]).

[5]The projection is required to carry out the analysis, while we ignore it in our implementation (see App. K.1).

[6]In practice, it is more efficient to take the infimum only over problems with different optimal arm in the last observed context $X_t$. This is indeed what we do in our experiments and all our theoretical results follow using this alternative definition with only minor changes.

[7]*$lin(\cdot)$ denotes any function with linear or sublinear dependence on the inputs (ignoring logarithmic terms). For example, $lin_{\geq 0}(x, y^2) \in \{a_0 + a_1 x + a_2 y + a_3 y^2 + a_4 xy^2 : a_i \geq 0\}$.*

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
