[Supplementary Material]

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

[8]Notice that none of the algorithms implement the exact form of the GLRT, but slight variations that provide equivalent guarantees.

[9] While the general structured bandit problem does contain the linear case, it is unclear how it can manage the contextual linear case.

[10]Notice that the constraint directly implies $\sum_{x,a} \eta(x,a) = z$.

[11]We recall that, as discussed in Sect. 3, $z$ introduces a more finite-time flavor into the lower bound, where pulls should now be allocated so as to satisfy the KL-information constraint within $z$ steps.

[12]In the main text we actually state that $(P_\lambda)$ is the Lagrangian relaxation of $(P_z)$ instead of $(\overline{P}_z)$. This is motivated by the fact that $(P_\lambda)$ and $(P_z)$ have the same optimal solution (see Prop. 3), though different optimal objective values.

[13]Recall that the regret of SOLID is not defined in terms of the optimization problem $(\mathrm{P}_z)$ or its Lagrangian, but only in terms of the rewards of the chosen arms compared to those of the optimal arms. This makes it possible to obtain good regret guarantees even when solving an infeasible optimization problem.

[14]Note that the bound on the sum of constraints of App. G.3.3 uses only the properties of the confidence intervals and of the exploitation test. Thus, it is applicable regardless of the feasibility of the optimization problems at each phase.

[15]Recall that, by definition, $\widetilde{\theta}_{t-1} \in \Theta$.

[16]Consider the vector with all components equal to 1, whose norm is $\sqrt{d}$.

[17]Problems that can be solved by a greedy strategy would not reveal any interesting empirical difference between SOLID and the other baselines.

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

# Appendix

## Table of Contents

# A    Notation and Definitions

We provide this table for easy reference. Notation will also be defined as it is introduced.

Table 1: Symbols

| | |
|---|---|
| $\theta^\star$ | The true reward parameter |
| $\mathcal{X}$ | Finite set of contexts |
| $\mathcal{A}$ | Finite set of arms |
| $\sigma^2$ | Variance of the Gaussian reward noise |
| $B$ | Maximum $l_2$-norm of realizable reward parameters |
| $L$ | Maximum $l_2$-norm of the features |
| $\rho$ | Context distribution |
| $\widehat{\rho}_t(x) := \frac{1}{t}\sum_{s=1}^t \mathbb{1}\{X_s = x\}$ | Estimated context distribution |
| $\mu_\theta(x, a)$ | Mean reward of context $x$ and arm $a$ |
| $\Delta_\theta(x, a) := \max_{a' \in \mathcal{A}} \mu_\theta(x, a') - \mu_\theta(x, a)$ | Gap of context $x$ and arm $a$ |
| $a_\theta^\star(x) := \mathrm{argmax}_{a \in \mathcal{A}}\, \mu_\theta(x, a)$ | Optimal arm of context $x$ |
| $\mu_\theta^\star(x) := \max_{a \in \mathcal{A}} \mu_\theta(x, a)$ | Optimal reward value of context $x$ |
| $d_{x,a}(\theta, \theta') := \frac{1}{2\sigma^2}(\mu_\theta(x, a) - \mu_{\theta'}(x, a))^2$ | KL divergence between $\theta$ and $\theta'$ at $x, a$ |
| $\Theta_{\mathrm{alt}} := \{\theta' \in \Theta \mid \exists x \in \mathcal{X},\ a_{\theta^\star}^\star(x) \neq a_{\theta'}^\star(x)\}$ | Set of alternative reward models |
| $\overline{\Theta}_{t-1} = \{\theta' \in \Theta \mid \exists x \in \mathcal{X},\ a_{\widehat{\theta}_{t-1}}^\star(x) \neq a_{\theta'}^\star(x)\}$ | Estimated set of alternative reward models |
| $v^\star(\theta^\star)$ | Optimal value of the optimization problem (P) |
| $\eta^\star$ | Optimal solution of the optimization problem (P) |
| $u^\star(z, \theta^\star)$ | Optimal value of the optimization problem ($\mathrm{P}_z$) |
| $\omega_{z,\theta^\star}^\star$ | Optimal solution of the optimization problem ($\mathrm{P}_z$) |
| $\underline{z}(\theta^\star) := \min\{z > 0 : (\mathrm{P}_z) \text{ is feasible}\}$ | Feasibility threshold of ($\mathrm{P}_z$) |
| $h(\omega, \lambda; z, \theta^\star) := f(\omega; \theta^\star) + \lambda g(\omega; z, \theta^\star)$ | Lagrangian relaxation of ($\mathrm{P}_z$) |
| $f(\omega; \theta^\star)$ | Objective function |
| $f_t(\omega)$ | Estimated (optimistic) objective function (see Eq. 2) |
| $g(\omega; z, \theta^\star)$ | Constraint function |
| $g_t(\omega, z)$ | Estimated (optimistic) constraint (see Eq. 3) |
| $E_t := \mathbb{1}\left\{\inf_{\theta' \in \overline{\Theta}_{t-1}} \|\widetilde{\theta}_{t-1} - \theta'\|_{\overline{V}_{t-1}}^2 \leq \beta_{t-1}\right\}$ | Exploration round |
| $N_t(x, a) := \sum_{s=1}^t \mathbb{1}\{X_t = x, A_t = a\}$ | Total number of visits to $(x, a)$ |
| $N_t^E(x, a) := \sum_{s=1}^t \mathbb{1}\{X_t = x, A_t = a, E_t\}$ | Number of visits to $(x, a)$ in exploration rounds |
| $S_t := \sum_{s=1}^t \mathbb{1}\{E_t\}$ | Total number of exploration rounds |
| $\beta_{t-1} := c_{n,1/n}$ | Theoretical threshold for the exploitation test in SOLID |
| $\gamma_t := c_{n,1/S_t^2}$ | Theoretical value for the confidence intervals in SOLID |
| $K_t \in \{0, 1, \dots\}$ | Phase index at time $t$ |
| $T_k$ | Time at which phase $k$ starts |
| $\mathcal{T}_k := \{t \in [n] : K_t = k\}$ | Time steps in phase $k$ |
| $\mathcal{T}_k^E := \{t \in \mathcal{T}_k : E_t\}$ | Exploration rounds in phase $k$ |
| $\{p_k\}_{k \geq 0}$ | Total number of exploration rounds in each phase |
| $\alpha_k^\lambda, \alpha_k^\omega$ | Step sizes |
| $V_t := \sum_{s=1}^t \phi(X_s, A_s)\phi(X_s, A_s)^\mathsf{T}$ | Design matrix |
| $\overline{V}_t := V_t + \nu I$ | Regularized design matrix ($\nu \geq 1$) |
| $U_t := \sum_{s=1}^t \phi(X_s, A_s)Y_s$ | Sum of reward-weighted features |
| $\widehat{\theta}_t := \overline{V}_t^{-1} U_t$ | Regularized least-squares estimate |
| $\widetilde{\theta}_t := \mathrm{argmin}_{\theta \in \Theta \cap \mathcal{C}_t} \|\theta - \widehat{\theta}_t\|_{\overline{V}_t}^2$ | Projected least-squares estimates |
| $\mathcal{C}_t := \{\theta \in \mathbb{R}^d : \|\theta - \widehat{\theta}_t\|_{\overline{V}_t}^2 \leq \beta_t\}$ | Confidence ellipsoid at time $t$ |
| $G_t$ | Good event (see App. F) |
| $M_n = \sum_{t=1}^n \mathbb{1}\{E_t, \neg G_t\}$ | Number of exploration rounds without good event |
| $M_{n,k} = \sum_{t \in \mathcal{T}_k^E} \mathbb{1}\{\neg G_t\}$ | Number of exploration rounds in phase $k$ without good event |

| Feature/Algorithm | OSSB | OAM | SPL | SOLID |
|---|---|---|---|---|
| *Setting* | general MAB | linear contextual | general MAB | linear contextual |
| *Objective fun.* | constrained | constrained | saddle (ratio) | saddle (Lagrangian) |
| *Opt. variables* | counts | counts | rates | policies |
| *Asympt. optimality* | order-opt | opt | opt | opt |
| *Finite-time bound* | ✓ | ✗ | ✓ | ✓ |
| *Explore/exploit* | tracking test | glrt | glrt | glrt |
| *Tracking* | direct | direct | cumulative | sampling |
| *Optimization* | exact | exact | incr. and best-response | incr. |
| *Exp. level* | forcing | forcing | unstruct. optimism | optimism |
| *Parameters* | forcing, test | forcing, test | gaps clip, test, conf. values | $\lambda_{\max}$, test, conf. values, phases |

Table 2: Comparison of structured bandit algorithms. OSSB [15], OAM [16], SPL [17] and SOLID (this paper).

# B   Comparison to Related Work

In Table 2 we compare several bandit algorithms along several dimensions:

- *Setting* refers to whether the algorithm is designed for general multi-armed bandit (non-contextual) structured problems or it is for the linear contextual case.

- *Objective function* refers to the optimization problem solved by the algorithm. It can be either the original constrained optimization in (P) or a saddle point problem (either obtained by taking the ratio of objective and constraints or the Lagrangian relaxation in ($P_z$)).

- *Optimization variables* refers to the variables that are optimized by the algorithm: *counts* is the $\eta$ variables in (P), *rates* is the ratio fraction of regret, *policies* is the $\omega$ variables in ($P_z$).

- *Asymptotic optimality* is either *order optimal* when only a logarithmic rate is proved with non-optimal constants, or *optimal*, in which case the leading constant is $v^\star(\theta)$ as in Prop. 1.

- *Finite-time bound* is whether finite-time guarantees are reported.

- *Explore/exploit* refers to the separation between exploration and exploitation steps and whether it is based on a *tracking performance test* or on the generalized likelihood ratio test (GLRT).[8]

- *Tracking* refers to how arms are selected during the exploration phase.

- *Optimization* refers to whether the optimization problem is solved exactly at each step or using an incremental method. SPL combines an incremental method using an exact computation of a best response solution.

- *Exploration level* refers to the technique used during exploration steps to guarantee a minimum level of exploration. The first option is *forcing* all arms to satisfy a hard threshold of minimal pulls. The second option is to include a form of *optimism* in the optimization problem.

- *Parameters* list the major parameters in the definition of the algorithm. This is often difficult since some algorithms directly pick theoretical values for some input parameters, while others may provide specific values only during the analysis. OSSB requires tuning the forcing parameter and the parameter used in the exploration/exploitation test. OAM has a forcing parameter and needs to properly tune the GLRT. SPL requires clipping the gap estimates from below, tuning the GLRT, and designing suitable confidence intervals for optimism. SOLID requires an upper bound for the multiplier, tuning of the GLRT, confidence intervals, and phases to tune the normalization factor $z$.

The major insights from this comparison can be summarized as follows:

- *Comparison* SOLID/*OAM:* This is the more direct comparison, since both algorithms are designed for contextual linear (see Sect. 6 for the empirical comparison). SOLID improves over OAM in almost all dimensions. On the theoretical side, we provide explicit finite-time regret bounds showing that SOLID successfully adapts to the context distribution, while the performance of OAM is significantly affected by $\rho_{\min}$. Furthermore, in many lower-order regret terms in the analysis of OAM the cardinality of the arm space appears linearly, while the regret of SOLID only depends on $\log(|\mathcal{A}|)$. On the algorithmic side, SOLID leverages a primal-dual gradient descent that greatly improves the computational complexity compared to the exact solution of the constrained optimization problem done in OAM at each exploration step. Furthermore, replacing the forcing strategy with an optimistic version of the optimization problem allows SOLID to better adapt to the problem and avoid pulling highly suboptimal/non-informative arms.

- *Comparison* SOLID/*SPL:* The comparison is more on the algorithmic and theoretical properties rather than the actual algorithms, since they are designed for different settings.[9] While both algorithms replace the constrained problem in the lower bound by a saddle point problem, SPL takes the ratio between constraints and regret, while in SOLID we take a more straightforward Lagrangian relaxation. As a result, in SOLID we rely on a rather standard primal-dual gradient approach to optimize $(P_z)$, while SPL relies on online learning algorithms for the solution of the saddle-point problem. Finally, both algorithms replace forcing by an optimistic version of the optimization problem. Nonetheless, SPL uses separate confidence intervals for each arm that ignore the structure of the problem, while SOLID relies on confidence intervals build specifically for the linear case. Finally, the regret bound of SPL, similarly to the one of OAM, depends linearly on $|\mathcal{A}|$ in several lower-order terms, even when instantiated for linear structures. SOLID, on the other hand, has only $\log(|\mathcal{A}|)$ dependence.

## C   Lower Bound

### C.1   Proof of Lem. 1

**Feasibility of** $(P_z)$**.**   We start from the first result in Lem. 1, which states the minimal value of $z$ for which $(P_z)$ is feasible. Clearly, the maximal value that the left-hand side of the KL constraint can assume is

$$\max_{\omega \in \Omega} \inf_{\theta' \in \Theta_{\mathrm{alt}}} \mathbb{E}_\rho \left[ \sum_{a \in \mathcal{A}} \omega(x, a) d_{x,a}(\theta^\star, \theta') \right],$$

which can also be interpreted as the solution to the associated pure-exploration (or best-arm identification) problem [e.g., 18]. Therefore,

$$\underline{z}(\theta^\star) := \min \left\{ z > 0 : (P_z) \text{ is feasible} \right\}$$

$$= \min \left\{ z > 0 : \max_{\omega \in \Omega} \inf_{\theta' \in \Theta_{\mathrm{alt}}} \mathbb{E}_\rho \left[ \sum_{a \in \mathcal{A}} \omega(x, a) d_{x,a}(\theta^\star, \theta') \right] \geq \frac{1}{z} \right\}$$

$$= \frac{1}{\max_{\omega \in \Omega} \inf_{\theta' \in \Theta_{\mathrm{alt}}} \mathbb{E}_\rho \left[ \sum_{a \in \mathcal{A}} \omega(x, a) d_{x,a}(\theta^\star, \theta') \right]}.$$

This proves the first statement in Lem. 1.

**Connection between** (P) **and** (P$_z$). In order to prove the second result, let us rewrite (P$_z$) in the following more convenient form:

$$
\begin{aligned}
\underset{\eta(x,a)\geq 0}{\text{minimize}} \quad & \sum_{x\in\mathcal{X}} \rho(x) \sum_{a\in\mathcal{A}} \eta(x,a)\Delta_{\theta^\star}(x,a) \\
\text{subject to} \quad & \inf_{\theta'\in\Theta_{\text{alt}}} \sum_{x\in\mathcal{X}} \rho(x) \sum_{a\in\mathcal{A}} \eta(x,a)d_{x,a}(\theta^\star,\theta') \geq 1, \qquad (\text{P}'_z)\\
& \sum_{a\in\mathcal{A}} \eta(x,a) = z \quad \forall x\in\mathcal{X}.
\end{aligned}
$$

Note that (P$'_z$) is obtained from (P$_z$) in the main paper by performing the change of variables $\eta(x,a) = z\omega(x,a)$, hence the two problems are equivalent. Recall that $v^\star(\theta^\star)$ is the optimal value of (P) and $u^\star(z,\theta^\star)$ is the optimal value of (P$'_z$) and (P$_z$) (if there exists one). We are interested in bounding the deviation between $u^\star(z,\theta^\star)$ and $v^\star(\theta^\star)$ as a function of $z$.

Let us first define the following set of *confusing* models:

$$
\widetilde{\Theta}_{\text{alt}} := \left\{ \theta' \in \Theta_{\text{alt}} : \forall x\in\mathcal{X}, \mu_{\theta^\star}^\star(x) = \mu_{\theta'}(x,a_x^\star) \right\},
$$

where, for the sake of readability, we abbreviate $a_x^\star = a_{\theta^\star}^\star(x)$. These models are indistinguishable from $\theta^\star$ by pulling only optimal arms. The following proposition, which was proved in [17], connects models in the alternative set $\Theta_{\text{alt}}$ with the confusing ones in $\widetilde{\Theta}_{\text{alt}}$.

**Proposition 2** ([17]). *There exists a constant $c_\Theta > 0$ such that, for all $\theta' \in \Theta_{\text{alt}}$, there exists $\theta'' \in \widetilde{\Theta}_{\text{alt}}$ such that,*

$$
\forall x\in\mathcal{X}, a\in\mathcal{A} \qquad |\mu_{\theta'}(x,a) - \mu_{\theta''}(x,a)| \leq c_\Theta |\mu_{\theta^\star}^\star(x) - \mu_{\theta'}(x,a_{\theta^\star}^\star(x))|.
$$

We now prove the bound on $u^\star(z,\theta)$ reported in Lem. 1.

*Proof of Lem. 1.* We start from the Lagrangian version of (P$'_z$).

$$
u^\star(z,\theta) = \min_{\eta\geq 0} \left\{ \sum_{x\in\mathcal{X}} \rho(x) \sum_{a\in\mathcal{A}} \eta(x,a)\Delta_{\theta^\star}(x,a) + \lambda^\star(z,\theta^\star)\left(1 - \inf_{\theta'\in\Theta_{\text{alt}}} \sum_{x\in\mathcal{X}} \rho(x) \sum_{a\in\mathcal{A}} \eta(x,a)d_{x,a}(\theta^\star,\theta')\right) \right\},
$$

subject to $\sum_{a\in\mathcal{A}} \eta(x,a) = z$ for each context $x \in \mathcal{X}$. Here $\lambda^\star(z,\theta^\star)$ is the optimal value of the Lagrange multiplier for the same problem. We distinguish two cases.

**Case 1:** $z < \max_{x\in\mathcal{X}} \frac{1}{\rho(x)} \sum_{a\neq a_{\theta^\star}^\star(x)} \eta^\star(x,a)$. Let

$$
\overline{\eta}(x,a) = z \cdot \begin{cases} \dfrac{\eta^\star(x,a)/\rho(x)}{\max_{x\in\mathcal{X}} \frac{1}{\rho(x)} \sum_{a\neq a_{\theta^\star}^\star(x)} \eta^\star(x,a)} & \text{if } a \neq a_{\theta^\star}^\star(x) \\[3mm] 1 - \dfrac{\sum_{a\neq a_{\theta^\star}^\star(x)} \eta^\star(x,a)/\rho(x)}{\max_{x\in\mathcal{X}} \frac{1}{\rho(x)} \sum_{a\neq a_{\theta^\star}^\star(x)} \eta^\star(x,a)} & \text{otherwise} \end{cases}
$$

where $\eta^\star$ is the optimal solution of (P). Since $\sum_a \overline{\eta}(x,a) = z$, we have that $u^\star(z,\theta^\star)$ is less or equal to the value of the Lagrangian for $\eta = \overline{\eta}$, i.e.,

$$
u^\star(z,\theta^\star) \leq v^\star(\theta^\star) + \lambda^\star(z,\theta^\star)\left(1 - \inf_{\theta'\in\Theta_{\text{alt}}} \sum_{x\in\mathcal{X}} \rho(x) \sum_{a\in\mathcal{A}} \overline{\eta}(x,a)d_{x,a}(\theta^\star,\theta')\right),
$$

where we used the fact that

$$
\sum_{x\in\mathcal{X}} \rho(x) \sum_{a\in\mathcal{A}} \overline{\eta}(x,a)\Delta_{\theta^\star}(x,a) = \underbrace{\frac{z}{\max_{x\in\mathcal{X}} \frac{1}{\rho(x)} \sum_{a\neq a_{\theta^\star}^\star(x)} \eta^\star(x,a)}}_{<1} \underbrace{\sum_{x\in\mathcal{X}} \sum_{a\neq a_{\theta^\star}^\star(x)} \eta^\star(x,a)\Delta_{\theta^\star}(x,a)}_{=v^\star(\theta^\star)}
$$

since $\Delta_{\theta^\star}(x,a_{\theta^\star}^\star(x)) = 0$. Since the KL divergence $d_{x,a}(\theta^\star,\theta')$ is lower-bounded by zero, in case 1 we have

$$
u^\star(z,\theta^\star) \leq v^\star(\theta^\star) + \lambda^\star(z,\theta^\star).
$$

**Case 2:** $z \geq \max_{x \in \mathcal{X}} \frac{1}{\rho(x)} \sum_{a \neq a^\star_{\theta^\star}(x)} \eta^\star(x,a)$. Let

$$\overline{\eta}(x,a) = \begin{cases} \eta^\star(x,a)/\rho(x) & \text{if } a \neq a^\star_{\theta^\star}(x) \\ z - \sum_{a \neq a^\star_{\theta^\star}(x)} \eta^\star(x,a)/\rho(x) & \text{otherwise} \end{cases}$$

where, as before, $\eta^\star$ is the optimal solution of (P). Since $z \geq \sum_{a \neq a^\star_{\theta^\star}(x)} \eta^\star(x,a)/\rho(x)$ for any $x \in \mathcal{X}$, $\overline{\eta}$ is well defined. Since $\overline{\eta}$ also sums to $z$ for each context, we have that $u^\star(z,\theta)$ is less or equal to the value of the Lagrangian for $\eta = \overline{\eta}$, i.e.,

$$u^\star(z,\theta^\star) \leq v^\star(\theta^\star) + \lambda^\star(z,\theta^\star) \left( 1 - \inf_{\theta' \in \Theta_{\text{alt}}} \sum_{x \in \mathcal{X}} \rho(x) \sum_{a \in \mathcal{A}} \overline{\eta}(x,a) d_{x,a}(\theta^\star, \theta') \right).$$

We first lower bound the infimum on the right hand side. We have

$$\inf_{\theta' \in \Theta_{\text{alt}}} \sum_{x \in \mathcal{X}} \rho(x) \sum_{a \in \mathcal{A}} \overline{\eta}(x,a) d_{x,a}(\theta^\star, \theta') = \min \left\{ \underbrace{\inf_{\theta' \in \widetilde{\Theta}_{\text{alt}}} \sum_{x \in \mathcal{X}} \rho(x) \sum_{a \in \mathcal{A}} \overline{\eta}(x,a) d_{x,a}(\theta^\star, \theta')}_{I_{\widetilde{\Theta}_{\text{alt}}}}, \right.$$

$$\left. \underbrace{\inf_{\theta' \in \Theta_{\text{alt}} \setminus \widetilde{\Theta}_{\text{alt}}} \sum_{x \in \mathcal{X}} \rho(x) \sum_{a \in \mathcal{A}} \overline{\eta}(x,a) d_{x,a}(\theta^\star, \theta')}_{I_{\Theta_{\text{alt}} \setminus \widetilde{\Theta}_{\text{alt}}}} \right\}.$$

$$(6)$$

By definition of $\overline{\eta}$ and $\eta^\star$, the infimum over the set of confusing models can be written as

$$I_{\widetilde{\Theta}_{\text{alt}}} = \inf_{\theta' \in \widetilde{\Theta}_{\text{alt}}} \sum_{x \in \mathcal{X}} \rho(x) \sum_{a \in \mathcal{A}} \overline{\eta}(x,a) d_{x,a}(\theta^\star, \theta') = \inf_{\theta' \in \widetilde{\Theta}_{\text{alt}}} \sum_{x \in \mathcal{X}} \sum_{a \neq a^\star_x} \eta^\star(x,a) d_{x,a}(\theta^\star, \theta') \geq 1, \quad (7)$$

where the equality holds since the KLs are zero in the optimal arms, which are the only arms where the values of $\overline{\eta}$ differ from those of $\eta^\star$, and the inequality holds since $\eta^\star$ is feasible. Regarding the infimum over the non-confusing models,

$$I_{\Theta_{\text{alt}} \setminus \widetilde{\Theta}_{\text{alt}}} = \inf_{\theta' \in \Theta_{\text{alt}} \setminus \widetilde{\Theta}_{\text{alt}}} \left( \underbrace{\sum_{x \in \mathcal{X}} \rho(x) \overline{\eta}(x, a^\star_x) d_{x, a^\star_x}(\theta^\star, \theta')}_{(i)} + \underbrace{\sum_{x \in \mathcal{X}} \sum_{a \neq a^\star_x} \eta^\star(x,a) d_{x,a}(\theta^\star, \theta')}_{(ii)} \right). \quad (8)$$

We partition the set of non-confusing models in two subsets:

$$\widetilde{\Theta}^{(1)}_{\text{alt}} := \left\{ \theta' \in \Theta_{\text{alt}} \setminus \widetilde{\Theta}_{\text{alt}} : \forall x \in \mathcal{X}, |\mu^\star_{\theta^\star}(x) - \mu_{\theta'}(x, a^\star_{\theta^\star}(x))| < \epsilon_z \right\}, \quad (9)$$

$$\widetilde{\Theta}^{(2)}_{\text{alt}} := \left\{ \theta' \in \Theta_{\text{alt}} \setminus \widetilde{\Theta}_{\text{alt}} : \exists x \in \mathcal{X}, |\mu^\star_{\theta^\star}(x) - \mu_{\theta'}(x, a^\star_{\theta^\star}(x))| \geq \epsilon_z \right\}. \quad (10)$$

The value of $\epsilon_z$ will be specified later. We have, for $\theta'' \in \widetilde{\Theta}_{\text{alt}}$,

$$\inf_{\theta' \in \widetilde{\Theta}^{(1)}_{\text{alt}}} \sum_{x \in \mathcal{X}} \rho(x) \sum_{a \in \mathcal{A}} \overline{\eta}(x,a) d_{x,a}(\theta^\star, \theta') \overset{(a)}{\geq} \inf_{\theta' \in \widetilde{\Theta}^{(1)}_{\text{alt}}} \sum_{x \in \mathcal{X}} \sum_{a \neq a^\star_x} \eta^\star(x,a) d_{x,a}(\theta^\star, \theta') \quad (11)$$

$$\overset{(b)}{\geq} \inf_{\theta' \in \widetilde{\Theta}^{(1)}_{\text{alt}}} \sum_{x \in \mathcal{X}} \sum_{a \neq a^\star_x} \eta^\star(x,a) \left( d_{x,a}(\theta^\star, \theta'') - \frac{1}{\sigma^2} |\mu_{\theta'}(x,a) - \mu_{\theta''}(x,a)| \right) \quad (12)$$

$$\overset{(c)}{\geq} 1 - \frac{1}{\sigma^2} \sup_{\theta' \in \widetilde{\Theta}^{(1)}_{\text{alt}}} \sum_{x \in \mathcal{X}} \sum_{a \neq a^\star_x} \eta^\star(x,a) |\mu_{\theta'}(x,a) - \mu_{\theta''}(x,a)| \quad (13)$$

$$\overset{(d)}{\geq} 1 - \frac{c_\Theta}{\sigma^2} \sup_{\theta' \in \widetilde{\Theta}^{(1)}_{\text{alt}}} \sum_{x \in \mathcal{X}} \sum_{a \neq a^\star_x} \eta^\star(x,a) \underbrace{|\mu^\star_{\theta^\star}(x) - \mu_{\theta'}(x, a^\star_x)|}_{< \epsilon_z} \quad (14)$$

$$\overset{(e)}{\geq} 1 - \frac{c_\Theta \epsilon_z}{\sigma^2} \sum_{x \in \mathcal{X}} \sum_{a \neq a^\star_x} \eta^\star(x,a), \quad (15)$$

where $(a)$ uses the fact that $(i) \geq 0$ and the definition of $\overline{\eta}$, $(b)$ uses the Lipschitz property of the KL divergence between Gaussians, $(c)$ uses the fact that $\overline{\eta}$ is feasible for confusing models (see Eq. 7), $(d)$ uses Prop. 2 and $(e)$ uses the definition of $\widetilde{\Theta}_{\text{alt}}^{(1)}$. Regarding the second set of alternative models,

$$\inf_{\theta' \in \widetilde{\Theta}_{\text{alt}}^{(2)}} \sum_{x \in \mathcal{X}} \rho(x) \sum_{a \in \mathcal{A}} \overline{\eta}(x,a) d_{x,a}(\theta^\star, \theta') \tag{16}$$

$$\overset{(f)}{\geq} \inf_{\theta' \in \widetilde{\Theta}_{\text{alt}}^{(2)}} \sum_{x \in \mathcal{X}} \rho(x) \left( z - \sum_{a \neq a_{\theta^\star}^\star(x)} \eta^\star(x,a)/\rho(x) \right) d_{x,a_x^\star}(\theta^\star, \theta') \tag{17}$$

$$\overset{(g)}{=} \inf_{\theta' \in \widetilde{\Theta}_{\text{alt}}^{(2)}} \sum_{x \in \mathcal{X}} \rho(x) \left( z - \sum_{a \neq a_{\theta^\star}^\star(x)} \eta^\star(x,a)/\rho(x) \right) \frac{1}{2\sigma^2} \underbrace{(\mu_{\theta^\star}(x,a_x^\star) - \mu_{\theta'}(x,a_x^\star))^2}_{\geq \epsilon_z^2} \tag{18}$$

$$\overset{(k)}{=} \frac{\epsilon_z^2}{2\sigma^2} \left( z - \sum_{x \in \mathcal{X}} \sum_{a \neq a_{\theta^\star}^\star(x)} \eta^\star(x,a) \right). \tag{19}$$

where $(f)$ uses the fact that $(ii) \geq 0$ and the definition of $\overline{\eta}$, $(g)$ uses the definition of KL for Gaussian distributions and $(k)$ uses the definition of $\widetilde{\Theta}_{\text{alt}}^{(2)}$. Let $z^\star(\theta^\star) := \sum_{x \in \mathcal{X}} \sum_{a \neq a_{\theta^\star}^\star(x)} \eta^\star(x,a)$. Putting together the results so far, we have

$$\inf_{\theta' \in \Theta_{\text{alt}}} \sum_{x \in \mathcal{X}} \rho(x) \sum_{a \in \mathcal{A}} \overline{\eta}(x,a) d_{x,a}(\theta^\star, \theta') \geq \min \left\{ 1, 1 - \frac{c_\Theta \epsilon_z z^\star(\theta^\star)}{\sigma^2}, \frac{\epsilon_z^2}{2\sigma^2} \left( z - z^\star(\theta^\star) \right) \right\}. \tag{20}$$

Setting $\epsilon_z = \sqrt{\frac{2\sigma^2}{z}}$,

$$\inf_{\theta' \in \Theta_{\text{alt}}} \sum_{x \in \mathcal{X}} \rho(x) \sum_{a \in \mathcal{A}} \overline{\eta}(x,a) d_{x,a}(\theta^\star, \theta') \geq \max \left\{ \min \left\{ 1 - \frac{c_\Theta \sqrt{2} z^\star(\theta^\star)}{\sigma \sqrt{z}}, 1 - \frac{z^\star(\theta^\star)}{z} \right\}, 0 \right\}, \tag{21}$$

Therefore, in case 2 we have

$$u^\star(z, \theta^\star) \leq v^\star(\theta^\star) + \lambda^\star(z, \theta^\star) \min \left\{ \max \left\{ \frac{c_\Theta \sqrt{2} z^\star(\theta^\star)}{\sigma \sqrt{z}}, \frac{z^\star(\theta^\star)}{z} \right\}, 1 \right\}.$$

**Bounding $\lambda^\star(z, \theta^\star)$.** Finally, we show that the optimal multiplier $\lambda^\star(z, \theta^\star)$ is bounded (regardless of which case $z$ falls into). Let $\underline{\eta} = z\underline{\omega}$, where $\underline{\omega} = \omega_{\underline{z}, \theta^\star}^\star$ is the pure-exploration solution obtained solving problem $(\mathrm{P}_z)$ with $\underline{z}(\theta^\star)$. Recall from the first statement of Lem. 1 that

$$\inf_{\theta' \in \Theta_{\text{alt}}} \sum_{x \in \mathcal{X}} \rho(x) \sum_{a \in \mathcal{A}} \underline{\omega}(x,a) d_{x,a}(\theta^\star, \theta') = \frac{1}{\underline{z}(\theta^\star)}.$$

Thus, $\underline{\eta}$ is strictly feasible for problem $(\widetilde{\mathrm{P}}_z)$ and has constraint value

$$\inf_{\theta' \in \Theta_{\text{alt}}} \sum_{x \in \mathcal{X}} \rho(x) \sum_{a \in \mathcal{A}} \underline{\eta}(x,a) d_{x,a}(\theta^\star, \theta') = \frac{z}{\underline{z}(\theta^\star)} > 1 \tag{22}$$

since $z > \underline{z}(\theta^\star)$ by assumption. Using the Slater's condition (see e.g., Lem. 3 in [22]),

$$0 \leq \lambda^\star(z, \theta^\star) \leq \frac{\sum_{x \in \mathcal{X}} \rho(x) \sum_{a \in \mathcal{A}} \Delta_{\theta^\star}(x,a)(\underline{\eta}(x,a) - \eta_z^\star(x,a))}{\inf_{\theta' \in \Theta_{\text{alt}}} \sum_{x \in \mathcal{X}} \rho(x) \sum_{a \in \mathcal{A}} \underline{\eta}(x,a) d_{x,a}(\theta^\star, \theta') - 1} \tag{23}$$

$$\leq \frac{z}{\frac{z}{\underline{z}(\theta^\star)} - 1} \sum_{x \in \mathcal{X}} \rho(x) \sum_{a \in \mathcal{A}} \underbrace{\Delta_{\theta^\star}(x,a)}_{\geq 0} \left( \underbrace{\underline{\omega}(x,a) - \eta_z^\star(x,a)/z}_{\geq 0} \right) \tag{24}$$

$$\leq \frac{z}{\frac{z}{\underline{z}(\theta^\star)} - 1} \sum_{x \in \mathcal{X}} \rho(x) \sum_{a \in \mathcal{A}} \underbrace{\Delta_{\theta^\star}(x,a)}_{\in [0, 2BL]} \underline{\omega}(x,a) \leq 2BL \frac{z\underline{z}(\theta^\star)}{z - \underline{z}(\theta^\star)}. \tag{25}$$

$\square$

## C.2 Discussion About Problem $(P_z)$

In this section we provide more intuition about the effect of explicitly adding the context distribution in the formulation of the lower bound. As mentioned in Sect. 3 the infimum in the original problem (P) may not be attainable, thus making it difficult to solve it and build a learning algorithm around it. A simple way to address this issue is to introduce a global constraint so that the sum of $\eta$ is constrained to a parameter $z$. This leads to the optimization

$$\inf_{\eta(x,a)\geq 0} \quad \sum_{x\in\mathcal{X}}\sum_{a\in\mathcal{A}} \eta(x,a)\Delta_{\theta^\star}(x,a)$$

$$\text{s.t.} \quad \inf_{\theta'\in\Theta_{\text{alt}}} \sum_{x\in\mathcal{X}}\sum_{a\in\mathcal{A}} \eta(x,a)d_{x,a}(\theta^\star,\theta') \geq 1 \qquad (\widetilde{P}_z)$$

$$\sum_{x,a} \eta(x,a) = z$$

Let $\widetilde{\eta}_z^\star$ be the optimal solution of $(\widetilde{P}_z)$ and $\widetilde{u}_z^\star$ be its associated optimal value. On the other hand, the problem $(P_z)$ we propose can be easily rewritten as

$$\inf_{\eta(x,a)\geq 0} \quad \sum_{x\in\mathcal{X}}\sum_{a\in\mathcal{A}} \eta(x,a)\Delta_{\theta^\star}(x,a)$$

$$\text{s.t.} \quad \inf_{\theta'\in\Theta_{\text{alt}}} \sum_{x\in\mathcal{X}}\sum_{a\in\mathcal{A}} \eta(x,a)d_{x,a}(\theta^\star,\theta') \geq 1 \qquad (P_z)$$

$$\sum_a \eta(x,a) = z\rho(x) \quad \forall x \in \mathcal{X}$$

where the constraint is now on each context and it depends on the context distribution ($\omega(x,a) = \frac{\eta(x,a)}{z\rho(x)}$).[10] The crucial difference w.r.t. $(\widetilde{P}_z)$ is that now the number of samples prescribed by $\eta$ needs to be "compatible" with the amount of samples that can be collected within $z$ steps from each context $x$ depending on its probability $\rho(x)$. Let $\eta_z^\star$ be the optimal solution of $(P_z)$ and $u_z^\star$ be its associated objective value. In order to understand how this difference may translate into a different behavior when integrated in an actual algorithm, let compare the two solutions $\widetilde{\eta}_z^\star$ and $\eta_z^\star$ if executed for $z$ steps.[11] Since neither of them can be "played" (i.e., only one arm can be selected at each step), we need to define a specific *execution strategy* to "realize" an allocation $\eta$. For the ease of exposition, let consider a simple strategy where in each context $x$, an arm $a$ is pulled at random proportionally to $\eta(x,a)$. Let $\widetilde{\zeta}_z(x,a)$ and $\zeta_z(x,a)$ the expected number of samples generated in each context-arm pair $(x,a)$ when sampling from $\widetilde{\eta}_z^\star$ and $\eta_z^\star$ respectively. Then we have

$$\widetilde{\zeta}_z(x,a) = \widetilde{\eta}_z^\star(x,a) \overbrace{\frac{z\rho(x)}{\sum_{a'}\widetilde{\eta}_z^\star(x,a')}}^{\text{mismatch } \alpha_z(x,a)} \qquad (26)$$

$$\zeta_z(x,a) = z\rho(x)\frac{\eta_z^\star(x,a)}{\sum_{a'}\eta_z^\star(x,a')} = \eta_z^\star(x,a) \qquad (27)$$

which reveals how $\widetilde{\eta}_z^\star(x,a)$, which was explicitly optimized under the constraint that the total number of samples was $z$, may not really be "realizable" in practice, since it ignores the context distribution and the number of samples that can be actually generated at each context $x$. On the other hand, on average the desired allocation $\eta_z^\star$ can always be realized within $z$ steps. Interestingly, the mismatch between $\widetilde{\eta}_z^\star(x,a)$ and $\widetilde{\zeta}_z(x,a)$ would no longer guarantee neither the performance $\widetilde{u}_z^\star$ "promised" by $\widetilde{\eta}_z^\star$ nor the feasibility for $(\widetilde{P}_z)$ (i.e., $\widetilde{\zeta}_z(x,a)$ may not satisfy the KL-information constraint). This would make considerably more difficult to build a learning algorithm on $\widetilde{\eta}_z^\star$ than on $\eta_z^\star$.

As it can be noticed in Eq. 26, the level mismatch is due to the execution strategy used to realize the allocation $\widetilde{\eta}_z^\star$ (in this case, a simple sampling approach) and better solutions may exist. We could even consider to directly optimize the execution strategy so as to achieve a mismatch $\alpha_z(x,a)$ that induce

an allocation $\widetilde{\zeta}_z(x, a)$ that performs best in terms of regret minimization under the KL-information constraint. Given the $\widetilde{\eta}_z^\star$ obtained from $(\widetilde{P}_z)$, we define the optimization problem

$$\inf_{\alpha(x,a) \geq 0} \quad \sum_{x \in \mathcal{X}} \sum_{a \in \mathcal{A}} \widetilde{\eta}_z^\star(x, a) \alpha(x, a) \Delta_\theta(x, a)$$

$$\text{s.t.} \quad \inf_{\theta' \in \Theta_{\mathrm{alt}}} \sum_{x \in \mathcal{X}} \sum_{a \in \mathcal{A}} \widetilde{\eta}_z^\star(x, a) \alpha(x, a) d_{x,a}(\theta, \theta') \geq 1 \qquad (\widetilde{P}_\alpha)$$

$$\sum_a \widetilde{\eta}_z^\star(x, a) \alpha(x, a) = z\rho(x)$$

Interestingly, a simple change of variables reveals that $(\widetilde{P}_\alpha)$ does coincide with $(P_z)$ that we originally introduced (i.e., $\alpha^\star(x, a) = \frac{\eta_z^\star(x,a)}{\widetilde{\eta}_z^\star(x,a)}$ minimizes the problem). This illustrates that solving $(P_z)$ indeed leads to the optimal allocation compatible with the context distribution and the constraint of $z$ realizations.

## D    Lagrangian Formulation

We discuss in more details the Lagrangian formulation presented in Section 3. Consider the following variant of $(P_z)$:

$$\max_{\omega \in \Omega} \quad \mathbb{E}_\rho \left[ \sum_{a \in \mathcal{A}} \omega(x, a) \mu_{\theta^\star}(x, a) \right] \quad \text{s.t.} \quad \inf_{\theta' \in \Theta_{\mathrm{alt}}} \mathbb{E}_\rho \left[ \sum_{a \in \mathcal{A}} \omega(x, a) d_{x,a}(\theta^\star, \theta') \right] \geq 1/z \qquad (\overline{P}_z)$$

This problem differs from $(P_z)$ since we replaced the action gaps with the means in the objective function and avoided scaling the latter by $z$. Let $\overline{\omega}_{z,\theta^\star}^\star$ the optimal solution of $(\overline{P}_z)$ and $\overline{u}^\star(z, \theta^\star)$ be its associated value (if the problem is unfeasible we set $\overline{u}^\star(z, \theta^\star) = +\infty$). Since the feasibility set is equivalent in $(P_z)$ and $(\overline{P}_z)$ as we only changed the objective function, the following proposition is immediate.

**Proposition 3.** *The following properties hold:*

1. *Both $(P_z)$ and $(\overline{P}_z)$ are feasible for $z \geq \underline{z}(\theta^\star)$;*

2. *$\omega_{z,\theta^\star}^\star = \overline{\omega}_{z,\theta^\star}^\star$.*

3. *$u^\star(z, \theta^\star) = z \left( \mu^\star - \overline{u}^\star(z, \theta^\star) \right)$ where $\mu^\star = \mathbb{E}_\rho[\mu_{\theta^\star}^\star(x)]$;*

Due to the equivalence demonstrated in Prop. 3, in the remaining we shall occasionally write $\omega_z^\star$ to denote both $\omega_{z,\theta^\star}^\star$ and $\overline{\omega}_{z,\theta^\star}^\star$.

We recall the Lagrangian relaxation problem of Sec. 3. For any $\omega \in \Omega$, let $f(\omega; \theta^\star)$ denote the objective function and $g(\omega, z; \theta^\star)$ denote the KL constraint

$$f(\omega; \theta^\star) = \mathbb{E}_\rho \left[ \sum_{a \in \mathcal{A}} \omega(x, a) \mu_{\theta^\star}(x, a) \right], \quad g(\omega; z, \theta^\star) = \inf_{\theta' \in \Theta_{\mathrm{alt}}} \mathbb{E}_\rho \left[ \sum_{a \in \mathcal{A}} \omega(x, a) d_{x,a}(\theta^\star, \theta') \right] - \frac{1}{z}.$$

The Lagrangian relaxation problem of $(\overline{P}_z)$ is[12]

$$\min_{\lambda \geq 0} \max_{\omega \in \Omega} \left\{ h(\omega, \lambda; z, \theta^\star) := f(\omega; \theta^\star) + \lambda g(\omega; z, \theta^\star) \right\}, \qquad (P_\lambda)$$

where $\lambda \in \mathbb{R}_{\geq 0}$ is a multiplier. We denote by $\lambda^\star(z, \theta^\star)$ the optimal multiplier for problem $(P_\lambda)$. We note that $f$ is linear in $\omega$, while $g$ is concave since it is an infimum of affine functions. Hence, the maximization in $(P_\lambda)$ is a non-smooth concave optimization problem.

**Strong duality.** We now verify that strong duality holds for the Lagrangian formulation ($P_\lambda$) (with respect to ($\overline{P}_z$)) when $z > \underline{z}(\theta^\star)$. This is immediate from the existence of a Slater point, as shown in the following proposition.

**Proposition 4** (Slater Condition). *For any $z > \underline{z}(\theta^\star)$, there exists a strictly feasible solution $\underline{\omega}$, i.e., $g(\underline{\omega}; z, \theta^\star) > 0$.*

*Proof.* This is a direct consequence of the fact that

$$\max_{\omega \in \Omega} \inf_{\theta' \in \Theta_{\mathrm{alt}}} \mathbb{E}_\rho \left[ \sum_{a \in \mathcal{A}} \omega(x, a) d_{x,a}(\theta^\star, \theta') \right] = \frac{1}{\underline{z}(\theta^\star)} > \frac{1}{z}. \tag{28}$$

See Lem. 1 and App. C. $\qquad\square$

Thus, the optimal solution of ($P_\lambda$) is $(\lambda^\star(z, \theta^\star), \omega_z^\star)$.

**Boundedness of the optimal multipliers.** We recall the following basic result.

**Lemma 2** (Lemma 3 of [22]). *For any $z > \underline{z}(\theta^\star)$, if $\overline{\omega}_z$ is a Slater point for ($\overline{P}_z$),*

$$\lambda^\star(z, \theta^\star) \leq \frac{f(\omega_z^\star; \theta^\star) - f(\overline{\omega}_z; \theta^\star)}{g(\overline{\omega}_z; z, \theta^\star)}$$

Using Lemma 2, we can prove the following result which will be very useful for the regret analysis.

**Lemma 3.** *For any $z \geq 2\underline{z}(\theta^\star)$,*

$$\lambda^\star(z, \theta^\star) \leq 2BL\underline{z}(\theta^\star). \tag{29}$$

*Proof.* From Prop. 4, $\underline{\omega}$ (the solution of the associated pure-exploration problem) is a Slater point for problem ($P_z$). Then, by Lemma 2,

$$\lambda^\star(z, \theta^\star) \leq \frac{f(\omega_z^\star; \theta^\star) - f(\underline{\omega}; \theta^\star)}{g(\underline{\omega}; z, \theta^\star)}.$$

Let $\mathrm{kl}(\underline{\omega})$ denote the expected KL of $\underline{\omega}$, so that $g(\underline{\omega}; z, \theta^\star) = \mathrm{kl}(\underline{\omega}) - 1/z$. Then,

$$\frac{f(\omega_z^\star; \theta^\star) - f(\underline{\omega}; \theta^\star)}{\mathrm{kl}(\underline{\omega}) - 1/z} \leq \frac{f(\omega_z^\star; \theta^\star)}{\mathrm{kl}(\underline{\omega}) - 1/z} \leq \frac{BL}{\mathrm{kl}(\underline{\omega}) - 1/z}. \tag{30}$$

Furthermore, since $\mathrm{kl}(\underline{\omega}) = 1/\underline{z}(\theta^\star)$,

$$\lambda^\star(z, \theta^\star) \leq \frac{BLz\underline{z}(\theta^\star)}{z - \underline{z}(\theta^\star)} \leq 2BL\underline{z}(\theta^\star),$$

where the last inequality holds for $z \geq 2\underline{z}(\theta^\star)$. This concludes the proof. $\qquad\square$

# E  Action Sampling

SOLID does not use standard tracking approaches for action selection (e.g., cumulative tracking [14, 18] or direct tracking [15, 16]) but a sampling strategy. Despite being simpler and more practical than tracking, we show that sampling from $\omega_t$ enjoys nice theoretical guarantees.

In the following lemmas we define the filtration $\mathcal{F}_t$ as the $\sigma$-algebra generated by the $t$-step history, $H_t = (X_1, A_1, Y_1, \ldots, X_t, A_t, Y_t)$.

**Lemma 4.** *Let $\{\omega_t\}_{t \geq 1}$ be such that $\omega_t \in \Omega$ and $\omega_t$ is $\mathcal{F}_{t-1}$-measurable. Let $\{X_t\}_{t \geq 1}$ be a sequence of i.i.d. contexts distributed according to $\rho$ and $\{A_t\}_{t \geq 1}$ be such that $A_t \sim \omega_t(X_t, \cdot)$. Then,*

$$\sum_{t \geq 1} \sum_{x \in \mathcal{X}} \sum_{a \in \mathcal{A}} \mathbb{P} \left\{ E_t, \left| N_t^E(x, a) - \rho(x) \sum_{s \leq t : E_s} \omega_s(x, a) \right| > \sqrt{\frac{S_t}{2} \log \left( S_t^2 |\mathcal{X}||\mathcal{A}| \right)} \right\} \leq \frac{\pi^2}{3}.$$

*Proof.* Let $Z_t := \mathbb{1}\{X_t = x, A_t = a\}$ and $\tau_s$ be a random variable such that the $s$-th exploration round occurs at time $\tau_s + 1$. Notice that $\{\tau_s\}_{s\geq 1}$ is a strictly-increasing sequence (i.e., $\tau_{s+1} > \tau_s$) of stopping times w.r.t. $\{\mathcal{F}_t\}_{t\geq 1}$. Furthermore, define

$$W_s := Z_{\tau_s+1} - \rho(x)\omega_{\tau_s+1}(x, a)$$

and let $\mathcal{G}_s := \mathcal{F}_{\tau_{s+1}}$. Using Lem. 10 in [26], we have that $\{W_s, \mathcal{G}_s\}_{s\geq 1}$ is a martingale difference sequence. Therefore, by Azuma's inequality

$$\mathbb{P}\left\{\left|\sum_{i=1}^{s} W_i\right| > \sqrt{\frac{s}{2}\log\frac{2}{\delta}}\right\} \leq \delta.$$

Let $a_t := \sqrt{\frac{S_t}{2}\log\left(S_t^2|\mathcal{X}||\mathcal{A}|\right)}$ and rewrite $N_t^E(x, a) = \sum_{s\leq t:E_s} Z_s$. Fix any $\bar{t} \geq 1$. Then,

$$\sum_{t=1}^{\bar{t}}\mathbb{1}\left\{E_t, \left|\sum_{s\leq t:E_s}(Z_s - \rho(x)\omega_s(x, a))\right| > a_t\right\}$$

$$\leq \sum_{s\geq 1}\mathbb{1}\left\{\left|\sum_{i=1}^{s}(Z_{\tau_i+1} - \rho(x)\omega_{\tau_i+1}(x, a))\right| > a_{\tau_s+1}, \tau_s + 1 \leq \bar{t}\right\}$$

$$\leq \sum_{s\geq 1}\mathbb{1}\left\{\left|\sum_{i=1}^{s} W_i\right| > \sqrt{\frac{s}{2}\log\left(s^2|\mathcal{X}||\mathcal{A}|\right)}\right\}.$$

In the last inequality, we used the fact that $a_{\tau_s+1} = \sqrt{s\log s}$. Taking expectations and applying Azuma's inequality with $\delta = \frac{2}{s^2|\mathcal{X}||\mathcal{A}|}$,

$$\sum_{t=1}^{\bar{t}}\mathbb{P}\left\{E_t, \left|\sum_{s\leq t:E_s}(Z_s - \rho(x)\omega_s(x, a))\right| > a_t\right\} \leq \frac{1}{|\mathcal{X}||\mathcal{A}|}\sum_{s\geq 1}\frac{2}{s^2} = \frac{\pi^2}{3|\mathcal{X}||\mathcal{A}|}.$$

The results holds for all $\bar{t}$, and the proof is concluded by summing over contexts and arms. $\square$

**Lemma 5.** *Let $\{\omega_t\}_{t\geq 1}$ be such that $\omega_t \in \Omega$ and $\omega_t$ is $\mathcal{F}_{t-1}$-measurable. Let $\{X_t\}_{t\geq 1}$ be a sequence of i.i.d. contexts distributed according to $\rho$ and $\{A_t\}_{t\geq 1}$ be such that $A_t \sim \omega_t(X_t, \cdot)$. Let $\{\varphi_t^i\}_{t\geq 1, i\in[m]}$ be a sequence of functions $\varphi_t^i : \mathcal{X} \times \mathcal{A} \to [-b, b]$ such that $\varphi_t^i(x, a)$ is $\mathcal{F}_{t-1}$-measurable for all $i \in [m]$. Then,*

$$\sum_{t\geq 1}\sum_{i=1}^{m}\mathbb{P}\left\{E_t, \left|\sum_{s\leq t:E_s}\left(\varphi_s^i(X_s, A_s) - \sum_{x\in\mathcal{X}}\rho(x)\sum_{a\in\mathcal{A}}\omega_s(x, a)\varphi_s^i(x, a)\right)\right| > b\sqrt{\frac{S_t}{2}\log(mS_t^2)}\right\} \leq \frac{\pi^2}{3}.$$

*Proof.* The proof follows the same steps as the one of Lemma 4. Fix $i \in [m]$. Let $Z_t := \varphi_t^i(X_t, A_t)$ and $\tau_s$ be a random variable such that the $s$-th exploration round occurs at time $\tau_s + 1$. Notice that $\{\tau_s\}_{s\geq 1}$ is a strictly-increasing sequence (i.e., $\tau_{s+1} > \tau_s$) of stopping times w.r.t. $\{\mathcal{F}_t\}_{t\geq 1}$. Furthermore, define

$$W_s := Z_{\tau_s+1} - \sum_{x\in\mathcal{X}}\sum_{a\in\mathcal{A}}\rho(x)\omega_{\tau_s+1}(x, a)\varphi_{\tau_s+1}^i(x, a)$$

and let $\mathcal{G}_s := \mathcal{F}_{\tau_{s+1}}$. Using Lem. 10 in [26], we have that $\{W_s, \mathcal{G}_s\}_{s\geq 1}$ is a martingale difference sequence (with differences bounded by $b$). Therefore, by Azuma's inequality

$$\mathbb{P}\left\{\left|\sum_{i=1}^{s} W_i\right| > b\sqrt{\frac{s}{2}\log\frac{2}{\delta}}\right\} \leq \delta.$$

Let $a_t := b\sqrt{\frac{S_t}{2}\log\left(mS_t^2\right)}$ and fix some $\bar{t} \geq 1$. Then,

$$
\sum_{t=1}^{\bar{t}} \mathbb{1}\left\{ E_t, \left| \sum_{s \leq t: E_s} \left( Z_s - \sum_{x \in \mathcal{X}} \sum_{a \in \mathcal{A}} \rho(x)\omega_s(x,a)\varphi_s^i(x,a) \right) \right| > a_t \right\}
$$

$$
\leq \sum_{s \geq 1} \mathbb{1}\left\{ \left| \sum_{j=1}^{s} \left( Z_{\tau_j+1} - \sum_{x \in \mathcal{X}} \sum_{a \in \mathcal{A}} \rho(x)\omega_{\tau_j+1}(x,a)\varphi_{\tau_j+1}^i(x,a) \right) \right| > a_{\tau_s+1}, \tau_s + 1 \leq \bar{t} \right\}
$$

$$
\leq \sum_{s \geq 1} \mathbb{1}\left\{ \left| \sum_{j=1}^{s} W_j \right| > b\sqrt{\frac{s}{2}\log(ms^2)} \right\}.
$$

In the last inequality, we used the fact that $a_{\tau_s+1} = b\sqrt{\frac{s}{2}\log(ms^2)}$. Taking expectations and applying Azuma's inequality with $\delta = \frac{2}{ms^2}$,

$$
\sum_{t=1}^{\bar{t}} \mathbb{P}\left\{ E_t, \left| \sum_{s \leq t: E_s} \left( Z_s - \sum_{x \in \mathcal{X}} \sum_{a \in \mathcal{A}} \rho(x)\omega_s(x,a)\varphi_s^i(x,a) \right) \right| > a_t \right\} \leq \sum_{s \geq 1} \frac{2}{ms^2} = \frac{\pi^2}{3m}.
$$

The results holds for all $\bar{t}$ and the proof follows by summing over all $i \in [m]$. $\square$

**Discussion.** Lemma 4 provides an analogous result to those obtained by tracking strategies, where the empirical pull counts are shown close to the sequence of conditional probabilities computed by the optimizer. Despite being simpler, our sampling rule achieves similar efficiency as existing tracking rules. In particular, our bound scales with $\log|\mathcal{A}|$, a factor that appears in the tightest known analysis of cumulative tracking [17]. The factor $\sqrt{S_t \log S_t}$ is not typically found in tracking strategies for MABs. However, we note that such dependency would naturally appear when generalizing these strategies to the contextual case.

Lemma 5 extends Lemma 4 to bound the deviation between expectations of measurable functions under the sequence of conditional probabilities and the same functions evaluated at the observed contexts/arms. This result will be very useful in the regret analysis to avoid undesirable linear dependencies on the number of arms.

# F High-Probability Events

In this section, we report the high-probability events used through the paper. Refer to App. I.1 for concentration inequalities.

Let $\Phi_{x,a} := \phi(x,a)\phi(x,a)^T$. We define the following events:

*true regret close to objective values*

$$G_t^\Delta := \left\{ \left| \sum_{s \le t: E_s} \left( \Delta_{\theta^\star}(X_s, A_s) - \sum_{x \in \mathcal{X}} \rho(x) \sum_{a \in \mathcal{A}} \omega_s(x,a) \Delta_{\theta^\star}(x,a) \right) \right| \le 2LB\sqrt{S_t \log S_t} \right\},$$
(31)

*true confidence intervals close to expected confidence intervals*

$$G_t^\phi := \left\{ \left| \sum_{s \le t: E_s} \left( \|\phi(X_s, A_s)\|_{\bar{V}_{s-1}^{-1}} - \sum_{x \in \mathcal{X}} \rho(x) \sum_{a \in \mathcal{A}} \omega_s(x,a) \|\phi(x,a)\|_{\bar{V}_{s-1}^{-1}} \right) \right| \le \frac{L}{\nu}\sqrt{S_t \log S_t} \right\},$$
(32)

*true design matrix close to expected design matrix*

$$G_t^d := \left\{ \left\| \sum_{s \le t: E_s} \left( \Phi_{X_s, A_s} - \sum_{x \in \mathcal{X}} \rho(x) \sum_{a \in \mathcal{A}} \omega_s(x,a) \Phi_{x,a} \right) \right\|_\infty \le L^2\sqrt{S_t \log (dS_t)} \right\},$$
(33)

*well-estimated context distribution*

$$G_t^\rho := \left\{ \forall x \in \mathcal{X} : |\hat{\rho}_{t-1}(x) - \rho(x)| \le 2\max\left( \sqrt{\frac{\log(|\mathcal{X}|S_t^2)}{2S_t}}, \frac{2}{t} \right) \right\},$$
(34)

*well-estimated parameters*

$$G_t^\theta := \left\{ \|\hat{\theta}_{t-1} - \theta^\star\|_{\bar{V}_{t-1}} \le \sqrt{\gamma_t} \right\}.$$
(35)

Furthermore, we define $G_t := \{G_t^\Delta, G_t^\phi, G_t^d, , G_t^\rho, G_t^\theta\}$ as the *"good"* event and let $M_t = \sum_{s=1}^t \mathbb{1}\{E_s, \neg G_s\}$ be the number of exploration rounds in which the good event does not hold. This can be bounded in expectation as follows.

**Lemma 6.** *Let $M_t = \sum_{s=1}^t \mathbb{1}\{E_s, \neg G_s\}$ be the number of exploration rounds in which the good event does not hold, then*

$$\mathbb{E}[M_t] \le \frac{3\pi^2}{2}.$$

*Proof.* Using the definition of $G_s$ together with the union bound,

$$\mathbb{E}[M_t] = \sum_{s=1}^t \mathbb{P}\{E_s, \neg G_s\} \le \sum_{s=1}^t \mathbb{P}\{E_s, \neg G_s^\Delta\} + \sum_{s=1}^t \mathbb{P}\{E_s, \neg G_s^\phi\} + \sum_{s=1}^t \mathbb{P}\{E_s, \neg G_s^d\}$$
$$+ \sum_{s=1}^t \mathbb{P}\{E_s, \neg G_s^\rho\} + \sum_{s=1}^t \mathbb{P}\{E_s, \neg G_s^\theta\}.$$

The first and second term can be bounded by Lemma 5 by noticing that $\Delta_{\theta^\star}(x,a) \le 2LB$ and that $\|\phi(x,a)\|_{\bar{V}_{s-1}^{-1}}$ is $\mathcal{F}_{s-1}$-measurable and upper-bounded by $\frac{L}{\nu}$ at all time steps. Thus,

$$\sum_{s=1}^t \mathbb{P}\{E_s, \neg G_s^\Delta\} + \sum_{s=1}^t \mathbb{P}\{E_s, \neg G_s^\phi\} \le \frac{2\pi^2}{3}.$$

Similarly, the third term can be bounded by Lemma 5 by taking a union bound over all elements of $\Phi_{x,a}$ (for a total of $d^2$ elements) and noting that each term is bounded by $L^2$. Thus,

$$\sum_{s=1}^t \mathbb{P}\{E_s, \neg G_s^d\} \le \frac{\pi^2}{3}.$$

The fourth term is

$$\sum_{s=1}^{t} \mathbb{P}\left\{E_s, \neg G_s^{\rho}\right\} \leq \sum_{x \in \mathcal{X}} \sum_{s \geq 1} \mathbb{P}\left\{E_s, |\widehat{\rho}_{s-1}(x) - \rho(x)| > 2 \max\left(\sqrt{\frac{\log(|\mathcal{X}|S_s^2)}{2S_s}}, \frac{2}{s}\right)\right\}$$

$$\leq \sum_{x \in \mathcal{X}} \sum_{s \geq 1} \mathbb{P}\left\{E_s, |\widehat{\rho}_{s-1}(x) - \widehat{\rho}_s(x)| + |\widehat{\rho}_s(x) - \rho(x)| > 2 \max\left(\sqrt{\frac{\log(|\mathcal{X}|S_s^2)}{2S_s}}, \frac{2}{s}\right)\right\}$$

$$\leq \sum_{x \in \mathcal{X}} \sum_{s \geq 1} \mathbb{P}\left\{E_s, |\widehat{\rho}_{s-1}(x) - \widehat{\rho}_s(x)| > \frac{2}{s}\right\} + \sum_{x \in \mathcal{X}} \sum_{s \geq 1} \mathbb{P}\left\{E_s, |\widehat{\rho}_s(x) - \rho(x)| > \sqrt{\frac{\log(|\mathcal{X}|S_s^2)}{2S_s}}\right\}$$

$$\leq \frac{\pi^2}{3}.$$

Here we used the fact that the absolute difference between two consecutive empirical means with samples bounded by 1 cannot be larger than $\frac{2}{s}$. We also used Lemma 7 to bound the second term. Finally, the fifth term can be directly bounded by Lemma 8:

$$\sum_{s=1}^{t} \mathbb{P}\left\{E_s, \neg G_s^{\theta}\right\} \leq \frac{\pi^2}{6}.$$

Combining the five bounds concludes the proof. □

# G  Regret Proof

We start decomposing the regret based on whether $E_t$ holds or not:

$$R_n = \sum_{t=1}^{n} \Delta_{\theta^\star}(X_t, A_t) \mathbb{1}\left\{\neg E_t\right\} + \sum_{t=1}^{n} \Delta_{\theta^\star}(X_t, A_t) \mathbb{1}\left\{E_t\right\} = R_n^{\text{exploit}} + R_n^{\text{explore}}.$$

Throughout the proof, as stated in the main theorem, we use $\beta_{t-1} := c_{n,1/n}$ and $\gamma_t := c_{n,1/S_t^2}$.

## G.1  Outline

An outline of our proof is as follows.

**Step 1**. (App. G.2) Using the confidence set derived in App. J, we show that the regret suffered when the algorithm enters the exploitation step is finite;

**Step 2**. (App. G.3.1) Using the properties of our action sampling strategy, we reduce the regret incurred during exploration rounds to the sum of objective values of the policies computed incrementally by primal-dual gradient ascent;

**Step 3**. (App. G.3.2) By combining standard tools from convex optimization with the properties of our confidence intervals, we relate the sum of objective values at each phase to the corresponding optimal value and constraint violations;

**Step 4**. (App. G.3.3) We relate the sum of constraints to the exploitation test used by SOLID. In particular, using the fact that the algorithm is not in the exploitation step, we show that the sum of constraints cannot be larger than $\mathcal{O}(\log n)$;

**Step 5**. (App. G.3.4) We combine the results obtained in the previous steps to show a first bound on the expected regret suffered during the exploration rounds. Our bound has the optimal dependency on $v^\star(\theta^\star) \log n$ but scales with the expected number $\mathbb{E}[K_n]$ of phases executed by the algorithm;

**Step 6**. (App. G.3.5) By relating the upper bound on the sum of constraints computed at Step 3 and a lower bound on the same quantity, we obtain an upper bound on $K_n$ as a function of the chosen sequences $p_k, z_k$;

**Step 7**. (App. G.3.6) We derive the final result by combining the bound on $K_n$ of Step 5 using the exponential schedule for $p_k, z_k$ with the partial regret bound of Step 4.

## G.2 Regret during Exploitation

We show that the regret suffered when exploitation occurs is finite. Let $\beta_{t-1} := c_{n,1/n}$, where $c_{n,\delta}$ was defined in Thm. 1. Then $F_t := \mathbb{1}\left\{\|\widehat{\theta}_{t-1} - \theta^\star\|^2_{\overline{V}_{t-1}} \leq c_{n,1/n}\right\}$ is the event under which the true model belongs to the confidence set, which holds with probability at least $1 - 1/n$ by the same theorem. We leverage this to decompose the regret during exploitation as:

$$R_n^{\text{exploit}} = \sum_{t=1}^n \Delta_{\theta^\star}(X_t, A_t)\mathbb{1}\{\neg E_t, F_t\} + \sum_{t=1}^n \Delta_{\theta^\star}(X_t, A_t)\mathbb{1}\{\neg E_t, \neg F_t\}.$$

The expectation of the second term is bounded by

$$\mathbb{E}\left[\sum_{t=1}^n \underbrace{\Delta_{\theta^\star}(X_t, A_t)}_{\leq 2LB}\mathbb{1}\{\neg E_t, \neg F_t\}\right] \leq 2LB \cdot \mathbb{E}\left[\sum_{t=1}^n \mathbb{1}\{\neg F_t\}\right] \leq 2LB\sum_{t=1}^n \underbrace{\mathbb{P}\{\neg F_t\}}_{\leq 1/n} \leq 2LB,$$

where we bounded $\mathbb{P}\{\neg F_t\} \leq \frac{1}{n}$ by using Thm. 1 with $\delta = 1/n$. Regarding the first term, we have two possible cases. If $a^\star_{\widetilde{\theta}_{t-1}}(X_t) = a^\star_{\theta^\star}(X_t)$, then the algorithm suffers no regret since by definition it pulls the empirically optimal arm (which is the optimal arm in this case). If $a^\star_{\widetilde{\theta}_{t-1}}(X_t) \neq a^\star_{\theta^\star}(X_t)$, then it must be that $\theta^\star \in \overline{\Theta}_{t-1}$, that is, the true model is in the set of alternative models for the current context. Under $\neg E_t$, this implies that

$$\|\widehat{\theta}_{t-1} - \theta^\star\|^2_{\overline{V}_{t-1}} \geq \|\widetilde{\theta}_{t-1} - \theta^\star\|^2_{\overline{V}_{t-1}} \geq \inf_{\theta'\in\overline{\Theta}_{t-1}}\|\widehat{\theta}_{t-1} - \theta'\|^2_{\overline{V}_{t-1}} > \beta_{t-1} = c_{n,1/n},$$

where the first inequality is due to the fact that the good event $F_t$ holds and Cor. 1. This is a contradiction with respect to $F_t$. Therefore, $\neg E_t$ and $F_t$ cannot hold at the same time and the algorithm suffers no regret. Combining these results, we conclude

$$\mathbb{E}\left[R_n^{\text{exploit}}\right] \leq 2LB.$$

## G.3 Regret under Exploration

The key challenge is to bound the regret during the exploration rounds. We proceed by following the steps outlined in App. G.1.

### G.3.1 From Regret to Objective Values

We decompose the regret incurred during exploration as

$$R_n^{\text{explore}} := \sum_{t=1}^n \Delta_{\theta^\star}(X_t, A_t)\mathbb{1}\{E_t\} \leq \sum_{t=1}^n \Delta_{\theta^\star}(X_t, A_t)\mathbb{1}\{E_t, G_t\} + 2LB\underbrace{\sum_{t=1}^n \mathbb{1}\{E_t, \neg G_t\}}_{:=M_n}.$$

Refer to App. F for the definition of $G_t$. The second term is $M_n$, the number of exploration rounds in which the good event does not hold, and can be bounded in expectation by using Lem. 6. The first one can be bounded by using the good event. Suppose, without loss of generality, that $E_n$ and $G_n$ hold (if they do not, the following reasoning can be repeated for the last time step at which these events hold). Then, using $G_t^\Delta$ (see App. F),

$$\sum_{t=1}^n \Delta_{\theta^\star}(X_t, A_t)\mathbb{1}\{E_t, G_t\} = \sum_{t\leq n:E_t} \Delta_{\theta^\star}(X_t, A_t)$$

$$\leq \sum_{t\leq n:E_t}\sum_{x\in\mathcal{X}}\rho(x)\sum_{a\in\mathcal{A}}\omega_t(x,a)\Delta_{\theta^\star}(x,a) + 2LB\sqrt{S_n\log S_n}.$$

Using the definition of phase, we can rewrite the first summation as

$$\sum_{t\leq n:E_t}\sum_{x\in\mathcal{X}}\rho(x)\sum_{a\in\mathcal{A}}\omega_t(x,a)\Delta_{\theta^\star}(x,a) = \sum_{k=0}^{K_n}\sum_{t\in\mathcal{T}_k^E}\sum_{x\in\mathcal{X}}\rho(x)\sum_{a\in\mathcal{A}}\omega_t(x,a)\Delta_{\theta^\star}(x,a).$$

Recall that $K_t$ is the (random) phase index at time $t$, while $\mathcal{T}_k^E$ is the set of exploration rounds in phase $k$. See App. A for a summary of notation. Let $\underline{k} := \min\{k \in \mathbb{N} | z_k \geq 2\underline{z}(\theta^\star)\}$. We split the sum into phases before and after $\underline{k}$. For those before, we have

$$\sum_{k<\underline{k}} \sum_{t\in\mathcal{T}_k^E} \sum_{x\in\mathcal{X}} \rho(x) \sum_{a\in\mathcal{A}} \omega_t(x,a)\Delta_{\theta^\star}(x,a) \leq 2LB\sum_{k<\underline{k}} |\mathcal{T}_k^E| \leq 2LB\sum_{k<\underline{k}} p_k,$$

which yields at most finite regret since $\{p_k\}$ is increasing. Let us now fix a phase $k \geq \underline{k}$ and bound the regret during its exploration rounds ($\mathcal{T}_k^E$). Note that the optimization problem in each phase $k \geq \underline{k}$ is feasible (see App. D). We have

$$\sum_{t\in\mathcal{T}_k^E} \sum_{x\in\mathcal{X}} \rho(x) \sum_{a\in\mathcal{A}} \omega_t(x,a)\Delta_{\theta^\star}(x,a)$$

$$= \sum_{t\in\mathcal{T}_k^E:G_t} \sum_{x\in\mathcal{X}} \rho(x) \sum_{a\in\mathcal{A}} \omega_t(x,a)\Delta_{\theta^\star}(x,a) + \sum_{t\in\mathcal{T}_k^E:\neg G_t} \sum_{x\in\mathcal{X}} \rho(x) \sum_{a\in\mathcal{A}} \omega_t(x,a)(\mu_{\theta^\star}^\star(x) - \mu_{\theta^\star}(x,a))$$

$$\leq \sum_{t\in\mathcal{T}_k^E:G_t} \sum_{x\in\mathcal{X}} \rho(x) \sum_{a\in\mathcal{A}} \omega_t(x,a)\Delta_{\theta^\star}(x,a) + M_{n,k}\mu^\star - \sum_{t\in\mathcal{T}_k^E:\neg G_t} \sum_{x\in\mathcal{X}} \rho(x) \sum_{a\in\mathcal{A}} \omega_t(x,a)\mu_{\theta^\star}(x,a).$$

Here we defined $\mu^\star := \sum_{x\in\mathcal{X}} \rho(x)\mu_{\theta^\star}^\star(x)$ and $M_{n,k}$ as the number of exploration rounds during phase $k$ where the good event does not hold. The last term can be bounded by $M_{n,k}BL$. Regarding the remaining two,

$$\sum_{t\in\mathcal{T}_k^E:G_t} \sum_{x\in\mathcal{X}} \rho(x) \sum_{a\in\mathcal{A}} \omega_t(x,a)\Delta_{\theta^\star}(x,a) + M_{n,k}\mu^\star$$

$$= (p_k - M_{n,k})\mu^\star + M_{n,k}\mu^\star - \sum_{t\in\mathcal{T}_k^E:G_t} \sum_{x\in\mathcal{X}} \rho(x) \sum_{a\in\mathcal{A}} \omega_t(x,a)\mu_{\theta^\star}(x,a)$$

$$= p_k\mu^\star + \underbrace{\sum_{t\in\mathcal{T}_k^E:G_t} \sum_{x\in\mathcal{X}} (\hat{\rho}_{t-1}(x) - \rho(x)) \sum_{a\in\mathcal{A}} \omega_t(x,a)\mu_{\theta^\star}(x,a)}_{(a)} - \underbrace{\sum_{t\in\mathcal{T}_k^E:G_t} \sum_{x\in\mathcal{X}} \hat{\rho}_{t-1}(x) \sum_{a\in\mathcal{A}} \omega_t(x,a)\mu_{\theta^\star}(x,a)}_{(b)}.$$

Term (a) can be bounded by

$$(a) \leq LB \underbrace{\sum_{t\in\mathcal{T}_k^E:G_t} \sum_{x\in\mathcal{X}} |\hat{\rho}_{t-1}(x) - \rho(x)|}_{\zeta_{n,k}}.$$

The second term $\zeta_{n,k}$ will be bounded shortly over all phases by means of Lemma 12. We now provide a lower bound to term (b). The first step is to relate this to the objective function optimized by the algorithm. Using the definition of $G_t$ and Lem. 10,

$$(b) \geq \sum_{t\in\mathcal{T}_k^E:G_t} \sum_{x\in\mathcal{X}} \hat{\rho}_{t-1}(x) \sum_{a\in\mathcal{A}} \omega_t(x,a)\left(\mu_{\widetilde{\theta}_{t-1}}(x,a) - \sqrt{\gamma_t}\|\phi(x,a)\|_{\bar{V}_{t-1}^{-1}}\right)$$

$$\pm \sum_{t\in\mathcal{T}_k^E:\neg G_t} \sum_{x\in\mathcal{X}} \hat{\rho}_{t-1}(x) \sum_{a\in\mathcal{A}} \omega_t(x,a)\underbrace{\mu_{\widetilde{\theta}_{t-1}}(x,a)}_{|\cdot|\leq LB} \pm \sum_{t\in\mathcal{T}_k^E} \sum_{x\in\mathcal{X}} \hat{\rho}_{t-1}(x) \sum_{a\in\mathcal{A}} \omega_t(x,a)\sqrt{\gamma_t}\|\phi(x,a)\|_{\bar{V}_{t-1}^{-1}}$$

$$\geq \sum_{t\in\mathcal{T}_k^E} f_t(\omega_t) - M_{n,k}BL - 2\sum_{t\in\mathcal{T}_k^E} \sum_{x\in\mathcal{X}} \hat{\rho}_{t-1}(x) \sum_{a\in\mathcal{A}} \omega_t(x,a)\sqrt{\gamma_t}\|\phi(x,a)\|_{\bar{V}_{t-1}^{-1}}$$

$$\geq \sum_{t\in\mathcal{T}_k^E} f_t(\omega_t) - M_{n,k}BL - 2\sqrt{\gamma_n}\Psi_{n,k}. \tag{36}$$

In the last step, we used $\sqrt{\gamma_t} \leq \sqrt{\gamma_n}$ (which is by definition $\mathcal{O}(\log S_n)$) and defined $\Psi_{n,k} := \sum_{t\in\mathcal{T}_k^E} \sum_{x\in\mathcal{X}} \hat{\rho}_{t-1}(x) \sum_{a\in\mathcal{A}} \omega_t(x,a)\|\phi(x,a)\|_{\bar{V}_{t-1}^{-1}}$.

To wrap-up the regret bound we have obtained so far, summing over all phases,

$$R_n^{\text{explore}} \leq 2LB \sum_{k < \underline{k}} p_k + \sum_{k \geq \underline{k}}^{K_n} p_k \mu^\star + LB \underbrace{\sum_{k \geq \underline{k}}^{K_n} \zeta_{n,k}}_{\leq \zeta_n} - \sum_{k \geq \underline{k}}^{K_n} \sum_{t \in \mathcal{T}_k^E} f_t(\omega_t)$$

$$+ 2LB \underbrace{\sum_{k \geq \underline{k}}^{K_n} M_{n,k}}_{\leq M_n} + 2LBM_n + 2\sqrt{\gamma_n} \underbrace{\sum_{k \geq \underline{k}}^{K_n} \Psi_{n,k}}_{\leq \Psi_n} + 2LB\sqrt{S_n \log S_n}.$$

Here we defined

$$\zeta_n := \sum_{t \leq n : E_t, G_t} \sum_{x \in \mathcal{X}} |\hat{\rho}_{t-1}(x) - \rho(x)|$$

and

$$\Psi_n := \sum_{t \leq n : E_t} \sum_{x \in \mathcal{X}} \hat{\rho}_{t-1}(x) \sum_{a \in \mathcal{A}} \omega_t(x, a) \|\phi(x, a)\|_{\bar{V}_{t-1}^{-1}}.$$

$\zeta_n$ can be bounded by Lemma 12 and $\Psi_n$ by Lemma 13. Both terms are of order $\mathcal{O}(\sqrt{S_n \log S_n})$. In order to simplify notation, we keep the specific bounds implicit in the remaining. Therefore, our partial regret bound is

$$R_n^{\text{explore}} \leq 2LB \sum_{k < \underline{k}} p_k + \sum_{k \geq \underline{k}}^{K_n} p_k \mu^\star - \sum_{k \geq \underline{k}}^{K_n} \sum_{t \in \mathcal{T}_k^E} f_t(\omega_t)$$

$$+ 4LBM_n + 2\sqrt{\gamma_n}\Psi_n + LB\zeta_n + 2LB\sqrt{S_n \log S_n}. \tag{37}$$

### G.3.2 Bounding the Sum of Objective Values

Our goal here is to lower bound the sum of objective values. As before, fix some phase index $k \geq \underline{k}$ and let $\lambda \geq 0$ be arbitrary. By recalling that the optimization process is reset at the beginning of each phase and using Corollary 2 with $\alpha_k^\lambda = \alpha_k^\omega = 1/\sqrt{p_k}$ and $\omega = \omega_{z_k}^\star$ (the optimal solution of problem $(P_{z_k})$),

$$\sum_{t \in \mathcal{T}_k^E} f_t(\omega_t) \geq \sum_{t \in \mathcal{T}_k^E} h_t(\omega_{z_k}^\star, \lambda_t, z_k) - \lambda \sum_{t \in \mathcal{T}_k^E} g_t(\omega_t, z_k) - \left( \log|\mathcal{A}| + \frac{b_\omega^2 + b_\lambda^2}{2} + \frac{(\lambda - \lambda_1)^2}{2} \right) \sqrt{p_k}.$$

$$\tag{38}$$

We recall that $b_\lambda$ and $b_\omega$ are the maximum sub-gradients in $\lambda$ and $\omega$, respectively. We now lower-bound the first term on the right-hand side. Since $h_t(\omega_{z_k}^\star, \lambda_t, z_k) = f_t(\omega_{z_k}^\star) + \lambda_t g_t(\omega_{z_k}^\star, z_k)$, $f_t(\omega_{z_k}^\star) \geq -LB$, $g_t(\omega_{z_k}^\star, z_k) \geq -\frac{1}{z_k}$, and $\lambda_t \leq \lambda_{\max}$, this term, evaluated on those steps where $G_t$ does not hold, can be lower-bounded by $\sum_{t \in \mathcal{T}_k^E : \neg G_t} h_t(\omega_{z_k}^\star, \lambda_t, z_k) \geq -(LB + \lambda_{\max}/z_k)M_{n,k}$. For any step $t \in \mathcal{T}_k^E$ in which $G_t$ holds, the optimism property (Lemma 11) yields

$$f_t(\omega_{z_k}^\star) \geq \sum_{x \in \mathcal{X}} (\hat{\rho}_{t-1}(x) - \rho(x)) \underbrace{\sum_{a \in \mathcal{A}} \omega_{z_k}^\star(x, a) \mu_{\theta^\star}(x, a)}_{|\cdot| \leq LB} + f(\omega_{z_k}^\star)$$

$$\geq f(\omega_{z_k}^\star) - LB \sum_{x \in \mathcal{X}} |\hat{\rho}_{t-1}(x) - \rho(x)|,$$

and

$$g_t(\omega_{z_k}^\star, z_k) \geq \inf_{\theta' \in \Theta_{alt}} \sum_{x \in \mathcal{X}} \hat{\rho}_{t-1}(x) \sum_{a \in \mathcal{A}} \omega_{z_k}^\star(x, a) d_{x,a}(\theta^\star, \theta') - \frac{1}{z_k} \pm g(\omega_{z_k}^\star)$$

$$\geq \inf_{\theta' \in \Theta_{alt}} \sum_{x \in \mathcal{X}} (\hat{\rho}_{t-1}(x) - \rho(x)) \sum_{a \in \mathcal{A}} \omega_{z_k}^\star(x, a) d_{x,a}(\theta^\star, \theta') + g(\omega_{z_k}^\star)$$

$$\geq g(\omega_{z_k}^\star) - \frac{2L^2 B^2}{\sigma^2} \sum_{x \in \mathcal{X}} |\hat{\rho}_{t-1}(x) - \rho(x)|.$$

Combining these two and using $\lambda_t \leq \lambda_{\max}$,

$$\sum_{t \in \mathcal{T}_k^E : G_t} h_t(\omega_{z_k}^\star, \lambda_t, z_k) \geq \sum_{t \in \mathcal{T}_k^E : G_t} \left( f(\omega_{z_k}^\star) + \lambda_t g(\omega_{z_k}^\star) \right) - LB \left( 1 + \frac{2LB\lambda_{\max}}{\sigma^2} \right) \zeta_{n,k}.$$

Note that $g(\omega_{z_k}^\star) \geq 0$ since by assumption $\omega_{z_k}^\star$ is feasible for the optimization problem $(P_{z_k})$. Furthermore, $\sum_{t \in \mathcal{T}_k^E : G_t} f(\omega_{z_k}^\star) = \sum_{t \in \mathcal{T}_k^E} f(\omega_{z_k}^\star) - \sum_{t \in \mathcal{T}_k^E : \neg G_t} \underbrace{f(\omega_{z_k}^\star)}_{|\cdot| \leq LB} \geq p_k f(\omega_{z_k}^\star) - LBM_{n,k}$.

Therefore, we obtain the following lower-bound on the sum of optimal objective values:

$$\sum_{t \in \mathcal{T}_k^E} h_t(\omega_{z_k}^\star, \lambda_t, z_k) \geq p_k f(\omega_{z_k}^\star) - LB \left( 1 + \frac{2LB\lambda_{\max}}{\sigma^2} \right) \zeta_{n,k} - (2LB + \lambda_{\max}/z_k)M_{n,k}.$$

Plugging this back into (38),

$$\sum_{t \in \mathcal{T}_k^E} f_t(\omega_t) \geq p_k f(\omega_{z_k}^\star) - \lambda \sum_{t \in \mathcal{T}_k^E} g_t(\omega_t, z_k) - a_\lambda \sqrt{p_k}$$

$$- LB \left( 1 + \frac{2LB\lambda_{\max}}{\sigma^2} \right) \zeta_{n,k} - (2LB + \lambda_{\max}/z_k)M_{n,k}, \qquad (39)$$

where, for simplicity, we defined $a_\lambda := \left( \log|\mathcal{A}| + \frac{b_\omega^2 + b_\lambda^2}{2} + \frac{(\lambda - \lambda_1)^2}{2} \right)$. Summing over all phases,

$$\sum_{k \geq \underline{k}}^{K_n} \sum_{t \in \mathcal{T}_k^E} f_t(\omega_t) \geq \sum_{k \geq \underline{k}}^{K_n} p_k f(\omega_{z_k}^\star) - \lambda \sum_{k \geq \underline{k}}^{K_n} \sum_{t \in \mathcal{T}_k^E} g_t(\omega_t, z_k) - a_\lambda \sum_{k \geq \underline{k}}^{K_n} \sqrt{p_k}$$

$$- LB \left( 1 + \frac{2LB\lambda_{\max}}{\sigma^2} \right) \zeta_n - (2LB + \lambda_{\max})M_n, \qquad (40)$$

where we used $\sum_{k \geq \underline{k}}^{K_n} M_{n,k} \leq M_n$, $\sum_{k \geq \underline{k}}^{K_n} \zeta_{n,k} \leq \zeta_n$, and $z_k \geq 1$.

### G.3.3 Bounding the sum of constraints

Our next step is to upper bound $\sum_{k \geq \underline{k}}^{K_n} \sum_{t \in \mathcal{T}_k^E} g_t(\omega_t, z_k)$, the sum of constraints of the policies played by the algorithm during feasible phases (those with $z_k \geq 2\underline{z}(\theta^\star)$). The intuition is that this term cannot be large (i.e., it cannot be above $\mathcal{O}(\log n)$), otherwise the exploitation test would trigger and we would not be exploring at step $n$. Using the definition of $g_t(\omega, z_k)$ (Eq. 3) and splitting the sum based on the good event

$$\sum_{k \geq \underline{k}}^{K_n} \sum_{t \in \mathcal{T}_k^E} g_t(\omega_t, z_{K_t})$$

$$\leq \sum_{t \leq n : E_t} \inf_{\theta' \in \widetilde{\Theta}_{t-1}} \sum_{x \in \mathcal{X}} \hat{\rho}_{t-1}(x) \sum_{a \in \mathcal{A}} \omega_t(x, a) d_{x,a}(\widetilde{\theta}_{t-1}, \theta') + \frac{2LB}{\sigma^2} \sqrt{\gamma_n} \Psi_n - \sum_{k \geq \underline{k}}^{K_n} \sum_{t \in \mathcal{T}_k^E} \frac{1}{z_k}$$

$$\leq \underbrace{\sum_{t \leq n : E_t, G_t} \inf_{\theta' \in \widetilde{\Theta}_{t-1}} \sum_{x \in \mathcal{X}} \hat{\rho}_{t-1}(x) \sum_{a \in \mathcal{A}} \omega_t(x, a) d_{x,a}(\widetilde{\theta}_{t-1}, \theta')}_{\textcircled{1}} + \frac{2L^2B^2}{\sigma^2} M_n + \frac{2LB}{\sigma^2} \sqrt{\gamma_n} \Psi_n - \sum_{k \geq \underline{k}}^{K_n} \frac{p_k}{z_k}.$$

Note that in the first step above we implicitly upper bounded the sum of KLs on the feasible phases with the sum of KLs over all exploration rounds. We can use the definition of $G_t$ and the optimism

(Lemma 11) to upper bound the first sum by

$$① \leq \sum_{t \leq n: E_t, G_t} \inf_{\theta' \in \bar{\Theta}_{t-1}} \sum_{x \in \mathcal{X}} \hat{\rho}_{t-1}(x) \sum_{a \in \mathcal{A}} \omega_t(x,a) d_{x,a}(\theta^\star, \theta') + \frac{2LB}{\sigma^2} \sqrt{\gamma_n} \Psi_n$$

$$\leq \underbrace{\sum_{t \leq n: E_t, G_t} \inf_{\theta' \in \bar{\Theta}_{t-1}} \sum_{x \in \mathcal{X}} \rho(x) \sum_{a \in \mathcal{A}} \omega_t(x,a) d_{x,a}(\theta^\star, \theta')}_{②} + \frac{2L^2 B^2}{\sigma^2} \underbrace{\sum_{t \leq n: E_t, G_t} \sum_{x \in \mathcal{X}} |\rho(x) - \hat{\rho}_{t-1}(x)|}_{=\zeta_n}$$

$$+ \frac{2LB}{\sigma^2} \sqrt{\gamma_n} \Psi_n.$$

Furthermore, the first term can be upper bounded by replacing each set $\bar{\Theta}_{t-1}$ over which the infimum is taken by $\Theta_{alt}$ (if the two sets were different, such term would be zero). Therefore,

$$② \leq \sum_{t \leq n: E_t, G_t} \inf_{\theta' \in \Theta_{alt}} \sum_{x \in \mathcal{X}} \rho(x) \sum_{a \in \mathcal{A}} \omega_t(x,a) d_{x,a}(\theta^\star, \theta')$$

$$\leq \underbrace{\inf_{\theta' \in \Theta_{alt}} \sum_{t \leq n: E_t} \sum_{x \in \mathcal{X}} \rho(x) \sum_{a \in \mathcal{A}} \omega_t(x,a) d_{x,a}(\theta^\star, \theta')}_{③}, \tag{41}$$

where we moved the infimum outside the outer sum and added the remaining steps where $G_t$ does not hold. Let $\Phi_{x,a} := \phi(x,a)\phi(x,a)^T$ and $V_{n,e} := \sum_{t \leq n: E_t} \Phi_{X_t, A_t}$ be the design matrix of the exploration rounds. Using the definition of $d_{x,a}$,

$$③ = \frac{1}{2\sigma^2} \inf_{\theta' \in \Theta_{alt}} (\theta^\star - \theta')^T \left( \sum_{t \leq n: E_t} \sum_{x \in \mathcal{X}} \rho(x) \sum_{a \in \mathcal{A}} \omega_t(x,a) \Phi_{x,a} \pm V_{n,e} \right) (\theta^\star - \theta')$$

$$\leq \inf_{\theta' \in \Theta_{alt}} \left\{ \frac{1}{2\sigma^2} (\theta^\star - \theta')^T V_{n,e} (\theta^\star - \theta') + \frac{1}{2\sigma^2} \|\theta^\star - \theta'\|_2^2 \left\| \sum_{t \leq n: E_t} \sum_{x \in \mathcal{X}} \rho(x) \sum_{a \in \mathcal{A}} \omega_t(x,a) \Phi_{x,a} - V_{n,e} \right\|_2 \right\}$$

$$\leq \inf_{\theta' \in \Theta_{alt}} \sum_{x \in \mathcal{X}} \sum_{a \in \mathcal{A}} N_n^E(x,a) d_{x,a}(\theta^\star, \theta') + \frac{2B^2}{\sigma^2} \left\| \sum_{t \leq n: E_t} \sum_{x \in \mathcal{X}} \rho(x) \sum_{a \in \mathcal{A}} \omega_t(x,a) \Phi_{x,a} - V_{n,e} \right\|_2$$

$$\leq \inf_{\theta' \in \Theta_{alt}} \sum_{x \in \mathcal{X}} \sum_{a \in \mathcal{A}} N_n^E(x,a) d_{x,a}(\theta^\star, \theta') + \frac{2B^2 \sqrt{d}}{\sigma^2} \left\| \sum_{t \leq n: E_t} \sum_{x \in \mathcal{X}} \rho(x) \sum_{a \in \mathcal{A}} \omega_t(x,a) \Phi_{x,a} - V_{n,e} \right\|_\infty.$$

Recall that $G_n$ holds. Then, by using the definition of $G^d$ to bound the norm,

$$③ \leq \underbrace{\inf_{\theta' \in \Theta_{alt}} \sum_{x \in \mathcal{X}} \sum_{a \in \mathcal{A}} N_{n-1}^E(x,a) d_{x,a}(\theta^\star, \theta')}_{④} + \frac{2B^2 L^2}{\sigma^2} + \frac{2B^2 L^2}{\sigma^2} \sqrt{dS_n \log(dS_n)}.$$

Here we used $N_n(x,a) = N_{n-1}(x,a) + \mathbb{1}\{X_n = x, A_n = a\}$ and upper bounded the KL at round $n$ by its maximum value. Moreover, similarly to Lem. 11 we can show that

$$④ \leq \inf_{\theta' \in \Theta_{alt}} \sum_{x \in \mathcal{X}} \sum_{a \in \mathcal{A}} N_{n-1}^E(x,a) d_{x,a}(\widetilde{\theta}_{n-1}, \theta') + \frac{2LB\sqrt{\gamma_n}}{\sigma^2} \underbrace{\sum_{x \in \mathcal{X}} \sum_{a \in \mathcal{A}} N_{n-1}^E(x,a) \|\phi(x,a)\|_{\bar{V}_{n-1}^{-1}}}_{\leq \Psi_n}.$$

The upper bound on the second term can be extracted from the proof of Lemma 13. The first term can be finally related to the exploitation test:

$$\inf_{\theta' \in \Theta_{alt}} \sum_{x \in \mathcal{X}} \sum_{a \in \mathcal{A}} N_{n-1}^E(x,a) d_{x,a}(\widetilde{\theta}_{n-1}, \theta') \leq \inf_{\theta' \in \overline{\Theta}_{n-1}} \sum_{x \in \mathcal{X}} \sum_{a \in \mathcal{A}} N_{n-1}^E(x,a) d_{x,a}(\widetilde{\theta}_{n-1}, \theta')$$

$$= \frac{1}{2\sigma^2} \inf_{\theta' \in \overline{\Theta}_{n-1}} \|\widetilde{\theta}_{n-1} - \theta'\|_{V_{n-1}}^2$$

$$\leq \frac{1}{2\sigma^2} \inf_{\theta' \in \overline{\Theta}_{n-1}} \|\widetilde{\theta}_{n-1} - \theta'\|_{\overline{V}_{n-1}}^2 \leq \frac{\beta_{n-1}}{2\sigma^2},$$

where the second-last inequality holds since $\overline{V}_{n-1} \succeq V_{n-1}$, and the last inequality holds since the algorithm is exploring at step $n$. By gathering all the results together, we get

$$\sum_{k \geq \underline{k}} \sum_{t \in \mathcal{T}_K^E : E_t} g_t(\omega_t, z_{K_t}) \leq \frac{\beta_{n-1}}{2\sigma^2} - \sum_{k \geq \underline{k}}^{K_n} \frac{p_k}{z_k} + \frac{2L^2 B^2}{\sigma^2} M_n + \frac{6LB}{\sigma^2} \sqrt{\gamma_n} \Psi_n + \frac{2B^2 L^2}{\sigma^2} \zeta_n$$

$$+ \frac{2B^2 L^2}{\sigma^2} \left( \sqrt{dS_n \log(dS_n)} + 1 \right). \tag{42}$$

### G.3.4    Back to the regret during exploration

So far we have (1) reduced the total regret during exploration to the sum of objective values (Eq. 37), (2) related this quantity to the optimal values of each phase (Eq. 40), and (3) derived an upper bound to the total sum of constraints (Eq. 42). We now combine all these results. If we first plug (40) into (37),

$$R_n^{\text{explore}} \leq 2LB \sum_{k < \underline{k}} p_k + \sum_{k \geq \underline{k}}^{K_n} p_k \mu^\star - \sum_{k \geq \underline{k}}^{K_n} p_k f(\omega_{z_k}^\star) + \lambda \sum_{k \geq \underline{k}}^{K_n} \sum_{t \in \mathcal{T}_k^E} g_t(\omega_t, z_k) + a_\lambda \sum_{k \geq \underline{k}}^{K_n} \sqrt{p_k}$$

$$+ (6LB + \lambda_{\max}) M_n + 2\sqrt{\gamma_n} \Psi_n + LB \left( 2 + \frac{2LB\lambda_{\max}}{\sigma^2} \right) \zeta_n + 2LB\sqrt{S_n \log S_n}. \tag{43}$$

Then, plugging (42) into this inequality,

$$R_n^{\text{explore}} \leq 2LB \sum_{k < \underline{k}} p_k + \sum_{k \geq \underline{k}}^{K_n} p_k \mu^\star - \sum_{k \geq \underline{k}}^{K_n} p_k f(\omega_{z_k}^\star) + \lambda \frac{\beta_{n-1}}{2\sigma^2} - \lambda \sum_{k \geq \underline{k}}^{K_n} \frac{p_k}{z_k} + a_\lambda \sum_{k \geq \underline{k}}^{K_n} \sqrt{p_k}$$

$$+ \left( \lambda \frac{2L^2 B^2}{\sigma^2} + 6LB + \lambda_{\max} \right) M_n + \left( 2 + \frac{6LB\lambda}{\sigma^2} \right) \sqrt{\gamma_n} \Psi_n + 2LB\sqrt{S_n \log S_n}$$

$$+ LB \left( 2 + \frac{2LB(\lambda_{\max} + \lambda)}{\sigma^2} \right) \zeta_n + \frac{2\lambda B^2 L^2}{\sigma^2} \left( \sqrt{dS_n \log(dS_n)} + 1 \right). \tag{44}$$

Let us simplify this expression so that it becomes more readable. First, we note that

$$\sum_{k \geq \underline{k}}^{K_n} p_k \mu^\star - \sum_{k \geq \underline{k}}^{K_n} p_k f(\omega_{z_k}^\star) = \sum_{k \geq \underline{k}}^{K_n} \frac{p_k}{z_k} \underbrace{z_k(\mu^\star - f(\omega_{z_k}^\star))}_{= u^\star(z_k, \theta^\star)} = \sum_{k \geq \underline{k}}^{K_n} \frac{p_k}{z_k} u^\star(z_k, \theta^\star).$$

Taking the expectation of both sides, we obtain

$$\mathbb{E}\left[R_n^{\text{explore}}\right] \leq 2LB \sum_{k < \underline{k}} p_k + \mathbb{E}\left[ \sum_{k \geq \underline{k}}^{K_n} \frac{p_k}{z_k} u^\star(z_k, \theta^\star) \right] + \lambda \frac{\beta_{n-1}}{2\sigma^2} - \lambda \mathbb{E}\left[ \sum_{k \geq \underline{k}}^{K_n} \frac{p_k}{z_k} \right]$$

$$+ a_\lambda \mathbb{E}\left[ \sum_{k \geq \underline{k}}^{K_n} \sqrt{p_k} \right] + \mathbb{E}\left[ \mathcal{O}(\sqrt{S_n \log S_n}) \right].$$

The remaining expectations on the right-hand side are due to the fact that $K_n$ (hence $S_n$) is still random. Setting $\lambda = v^\star(\theta^\star)$ and combining the second and fourth terms, we get

$$\sum_{k \geq \underline{k}}^{K_n} \frac{p_k}{z_k} u^\star(z_k, \theta^\star) - \lambda \sum_{k \geq \underline{k}}^{K_n} \frac{p_k}{z_k} = \sum_{k \geq \underline{k}}^{K_n} \frac{p_k}{z_k} \left( u^\star(z_k, \theta^\star) - v^\star(\theta^\star) \right)$$

$$= \sum_{k \geq \underline{k}: z_k < \bar{z}(\theta^\star)} \frac{p_k}{z_k} \left( u^\star(z_k, \theta^\star) - v^\star(\theta^\star) \right) + \sum_{k: z_k \geq \bar{z}(\theta^\star)}^{K_n} \frac{p_k}{z_k} \left( u^\star(z_k, \theta^\star) - v^\star(\theta^\star) \right),$$

where $\bar{z}(\theta^\star) := \max_{x \in \mathcal{X}} \sum_{a \neq a^\star_{\theta^\star}(x)} \frac{\eta^\star(x,a)}{\rho(x)}$ was defined in Lem. 1. For $k \geq \underline{k}$, we can use the perturbation bound (Lem. 1) on both terms. We obtain,

$$\sum_{k \geq \underline{k}: z_k < \bar{z}(\theta^\star)} \frac{p_k}{z_k} \left( u^\star(z_k, \theta^\star) - v^\star(\theta^\star) \right) \leq BL\underline{z}(\theta^\star) \sum_{k \geq \underline{k}: z_k < \bar{z}(\theta^\star)} \frac{p_k}{z_k - \underline{z}(\theta^\star)}$$

and

$$\sum_{k \geq \underline{k}: z_k \geq \bar{z}(\theta^\star)}^{K_n} \frac{p_k}{z_k} \left( u^\star(z_k, \theta^\star) - v^\star(\theta^\star) \right) \leq BL\underline{z}(\theta^\star) z^\star(\theta^\star) \sum_{k \geq \underline{k}: z_k \geq \bar{z}(\theta^\star)}^{K_n} \frac{p_k}{z_k - \underline{z}(\theta^\star)} \max\left\{ \frac{c_\Theta \sqrt{2}}{\sigma \sqrt{z_k}}, \frac{1}{z_k} \right\}.$$

**Partial regret bound** Plugging these bounds into the expected regret,

$$\mathbb{E}\left[ R_n^{\text{explore}} \right] \leq \underbrace{2BL \sum_{k < \underline{k}} p_k}_{\text{I}} + \underbrace{BL\underline{z}(\theta^\star) \sum_{k \geq \underline{k}: z_k < \bar{z}(\theta^\star)} \frac{p_k}{z_k - \underline{z}(\theta^\star)}}_{\text{II}} + \underbrace{v^\star(\theta^\star) \frac{\beta_{n-1}}{2\sigma^2}}_{\text{III}} + \underbrace{a_\lambda \mathbb{E}\left[ \sum_{k \geq \underline{k}}^{K_n} \sqrt{p_k} \right]}_{\text{IV}}$$

$$+ \underbrace{BL\underline{z}(\theta) z^\star(\theta^\star) \mathbb{E}\left[ \sum_{k: z_k \geq \bar{z}(\theta^\star)}^{K_n} \frac{p_k}{z_k - \underline{z}(\theta^\star)} \max\left\{ \frac{c_\Theta \sqrt{2}}{\sigma \sqrt{z_k}}, \frac{1}{z_k} \right\} \right]}_{\text{V}} + \underbrace{\mathbb{E}\left[ \mathcal{O}(\sqrt{S_n \log S_n}) \right]}_{\text{VI}}. \qquad (45)$$

The six terms constituting the bound are (from left to right):

    I. finite regret suffered in the phases where the optimization problem is infeasible;

    II. finite regret suffered in the phases in which we do not know much about the convergence rate of $u^\star(z, \theta^\star)$ to $v^\star(\theta^\star)$. This term is likely an artefact of the analysis;

    III. asymptotically-optimal regret rate;

    IV. regret suffered due to the incremental gradient updates and inversely proportional to the step sizes;

    V. regret suffered due to the fact that we solve ($P_z$) instead of (P);

    VI. other low-order terms mostly due to the concentration bounds.

Note that, since $\beta_{n-1} = c_{n,1/n}$ and $c_{n,1/n} \to 2\sigma^2 \log n$ as $n \to \infty$,

$$\limsup_{n \to \infty} \frac{v^\star(\theta^\star) \beta_{n-1}}{2\sigma^2 \log n} = v^\star(\theta^\star),$$

which is the asymptotically-optimal regret rate as prescribed by (P).

### G.3.5 Bounding the total number of phases

So far we proved an upper bound on the regret incurred during exploration which depends on the (random) number of phases. We now upper bound this random variable as a function of $z_k$ and $p_k$. In particular, we achieve this by focusing on the constraints only. The intuition is that, if the primal-dual algorithm works, then the sequence of policies played cannot violate the constraints *at each phase* too much. At the same time, these policies cannot satisfy the constraints too much, otherwise the

exploitation test would trigger and the algorithm would not be exploring at step $n$. Relating these two we obtain a bound on $K_n$.

Recall that, as we assumed before, $n$ is an exploration step in which the good event $G_n$ holds. Using (41) and the equations thereafter, we have

$$\inf_{\theta' \in \Theta_{alt}} \sum_{t \leq n : E_t} \sum_{x \in \mathcal{X}} \rho(x) \sum_{a \in \mathcal{A}} \omega_t(x, a) d_{x,a}(\theta^\star, \theta')$$

$$\leq \frac{\beta_{n-1}}{2\sigma^2} + \frac{2LB}{\sigma^2} \sqrt{\gamma_n} \Psi_n + \frac{2B^2 L^2}{\sigma^2} \left( \sqrt{dS_n \log (dS_n)} + 1 \right). \tag{46}$$

where the last two terms are $\mathcal{O}(\sqrt{S_n \log S_n})$.

We now provide a lower-bound on the same quantity. Fix a phase index $k \geq \underline{k}$. From (39), we have

$$\sum_{t \in \mathcal{T}_k^E} (f_t(\omega_t) + \lambda g_t(\omega_t, z_k)) \geq p_k f(\omega_{z_k}^\star) - a_\lambda \sqrt{p_k} - (2LB + \lambda_{\max}) M_{n,k}$$

$$- LB \left( 1 + \frac{2LB\lambda_{\max}}{\sigma^2} \right) \zeta_{n,k}, \tag{47}$$

The left-hand side can be upper-bounded by using the optimism property to obtain the true objective and constraint. Regarding the objective function, we have

$$\sum_{t \in \mathcal{T}_k^E} f_t(\omega_t) = \sum_{t \in \mathcal{T}_k^E : G_t} f_t(\omega_t) + \sum_{t \in \mathcal{T}_k^E : \neg G_t} f_t(\omega_t)$$

$$\leq \sum_{t \in \mathcal{T}_k^E : G_t} \sum_{x \in \mathcal{X}} \hat{\rho}_{t-1}(x) \sum_{a \in \mathcal{A}} \omega_t(x, a) \mu_{\widetilde{\theta}_{t-1}}(x, a) + \sum_{t \in \mathcal{T}_k^E : \neg G_t} \sum_{x \in \mathcal{X}} \hat{\rho}_{t-1}(x) \sum_{a \in \mathcal{A}} \omega_t(x, a) \mu_{\widetilde{\theta}_{t-1}}(x, a) + \sqrt{\gamma_n} \Psi_{n,k}$$

$$\leq \underbrace{\sum_{t \in \mathcal{T}_k^E : G_t} \sum_{x \in \mathcal{X}} \hat{\rho}_{t-1}(x) \sum_{a \in \mathcal{A}} \omega_t(x, a) \mu_{\widetilde{\theta}_{t-1}}(x, a)}_{(a)} + BLM_{n,k} + \sqrt{\gamma_n} \Psi_{n,k}.$$

Regarding the sum over the good events, using Lem. 11,

$$(a) \leq \sum_{t \in \mathcal{T}_k^E : G_t} \sum_{x \in \mathcal{X}} \hat{\rho}_{t-1}(x) \sum_{a \in \mathcal{A}} \omega_t(x, a) \mu_{\theta^\star}(x, a) + \sqrt{\gamma_n} \Psi_{n,k} \tag{48}$$

$$\leq \sum_{t \in \mathcal{T}_k^E} \underbrace{\sum_{x \in \mathcal{X}} \rho(x) \sum_{a \in \mathcal{A}} \omega_t(x, a) \mu_{\theta^\star}(x, a)}_{=f(\omega_t)} + BL \underbrace{\sum_{t \in \mathcal{T}_k^E : G_t} \sum_{x \in \mathcal{X}} |\hat{\rho}_t(x) - \rho(x)|}_{\zeta_{n,k}} + \sqrt{\gamma_n} \Psi_{n,k}. \tag{49}$$

Therefore,

$$\sum_{t \in \mathcal{T}_k^E} f_t(\omega_t) \leq \sum_{t \in \mathcal{T}_k^E} f(\omega_t) + BL\zeta_{n,k} + 2\sqrt{\gamma_n} \Psi_{n,k} + BLM_{n,k}.$$

We can follow the same reasoning to upper bound the sum of constraints. Since the KLs are upper-bounded by $2B^2 L^2/\sigma^2$,

$$\sum_{t \in \mathcal{T}_k^E} g_t(\omega_t, z_k) \leq \sum_{t \in \mathcal{T}_k^E} g(\omega_t, z_k) + \frac{2B^2 L^2}{\sigma^2} \zeta_{n,k} + \frac{4BL}{\sigma^2} \sqrt{\gamma_n} \Psi_{n,k} + \frac{2B^2 L^2}{\sigma^2} M_{n,k}.$$

Combining the bounds on $f$ and $g$,

$$\sum_{t \in \mathcal{T}_k^E} (f(\omega_t) + \lambda g(\omega_t, z_k)) \geq p_k f(\omega_{z_k}^\star) - \left( 3BL + \lambda_{\max} + \lambda \frac{2B^2 L^2}{\sigma^2} \right) M_{n,k} - a_\lambda \sqrt{p_k}$$

$$- 2BL \left( 1 + \frac{(\lambda_{\max} + \lambda)BL}{\sigma^2} \right) \zeta_{n,k} - \left( 2 + \frac{4BL\lambda}{\sigma^2} \right) \sqrt{\gamma_n} \Psi_{n,k}.$$

Let $\bar{\omega}_{t,k} := \frac{1}{p_k} \sum_{t \in \mathcal{T}_k^E} \omega_t$ be the average policy played in phase $k$. Since $f$ is linear and $g$ is concave, $\sum_{t \in \mathcal{T}_k^E} (f(\omega_t) + \lambda g(\omega_t, z_k)) \le p_k f(\bar{\omega}_{t,k}) + \lambda p_k g(\bar{\omega}_{t,k}, z_k)$. We now set

$$\lambda = \begin{cases} 2\lambda_{\max} & \text{if } [g(\bar{\omega}_{t,k}, z_k)]_- \neq 0 \\ 0 & \text{otherwise} \end{cases}$$

where $[x]_- = \min\{x, 0\}$. Therefore,

$$p_k \left( f(\bar{\omega}_{t,k}) - f(\omega_{z_k}^\star) + 2\lambda_{\max}[g(\bar{\omega}_{t,k}, z_k)]_- \right) \ge - \left( 3BL + \lambda_{\max} + \lambda_{\max} \frac{4B^2 L^2}{\sigma^2} \right) M_{n,k} - a_{\lambda_{\max}} \sqrt{p_k}$$
$$- 2BL \left( 1 + \frac{3\lambda_{\max} BL}{\sigma^2} \right) \zeta_{n,k} - \left( 2 + \frac{8BL\lambda_{\max}}{\sigma^2} \right) \sqrt{\gamma_n} \Psi_{n,k}.$$

Lemma 3 together with Asm. 2 ensures that, for $k \ge \underline{k}$, $\lambda^\star(z_k, \theta^\star) \le \lambda_{\max}$. Thus, we can apply Theorem 42 of [23] and obtain

$$p_k g(\bar{\omega}_{t,k}, z_k) \ge p_k [g(\bar{\omega}_{t,k}, z_k)]_- \ge - \left( 3BL + \lambda_{\max} + \lambda_{\max} \frac{4B^2 L^2}{\sigma^2} \right) \frac{M_{n,k}}{2\lambda_{\max}} - \frac{a_{\lambda_{\max}} \sqrt{p_k}}{2\lambda_{\max}}$$
$$- 2BL \left( 1 + \frac{3\lambda_{\max} BL}{\sigma^2} \right) \frac{\zeta_{n,k}}{2\lambda_{\max}} - \left( 2 + \frac{8BL\lambda_{\max}}{\sigma^2} \right) \frac{\sqrt{\gamma_n} \Psi_{n,k}}{2\lambda_{\max}}.$$

Summing both sides over all phases,

$$\sum_{k \ge \underline{k}}^{K_n} p_k g(\bar{\omega}_{t,k}, z_k) = \sum_{k \ge \underline{k}}^{K_n} p_k \left( \inf_{\theta' \in \Theta_{alt}} \sum_{x \in \mathcal{X}} \rho(x) \sum_{a \in \mathcal{A}} \bar{\omega}_{t,k}(x, a) d_{x,a}(\theta^\star, \theta') - \frac{1}{z_k} \right)$$
$$= \sum_{k \ge \underline{k}}^{K_n} \left( \inf_{\theta' \in \Theta_{alt}} \sum_{t \in \mathcal{T}_k^E} \sum_{x \in \mathcal{X}} \rho(x) \sum_{a \in \mathcal{A}} \omega_t(x, a) d_{x,a}(\theta^\star, \theta') - \frac{p_k}{z_k} \right)$$
$$\le \inf_{\theta' \in \Theta_{alt}} \sum_{t \le n: E_t} \sum_{x \in \mathcal{X}} \rho(x) \sum_{a \in \mathcal{A}} \omega_t(x, a) d_{x,a}(\theta^\star, \theta') - \sum_{k \ge \underline{k}}^{K_n} \frac{p_k}{z_k}.$$

Therefore,

$$\inf_{\theta' \in \Theta_{alt}} \sum_{t \le n: E_t} \sum_{x \in \mathcal{X}} \rho(x) \sum_{a \in \mathcal{A}} \omega_t(x, a) d_{x,a}(\theta^\star, \theta') \ge \sum_{k \ge \underline{k}}^{K_n} \frac{p_k}{z_k} - \left( 3BL + \lambda_{\max} + \lambda_{\max} \frac{4B^2 L^2}{\sigma^2} \right) \frac{M_n}{2\lambda_{\max}}$$
$$- \frac{a_{\lambda_{\max}} \sqrt{p_k}}{2\lambda_{\max}} - 2BL \left( 1 + \frac{3\lambda_{\max} BL}{\sigma^2} \right) \frac{\zeta_n}{2\lambda_{\max}} - \left( 2 + \frac{8BL\lambda_{\max}}{\sigma^2} \right) \frac{\sqrt{\gamma_n} \Psi_n}{2\lambda_{\max}}.$$

Combining this with (46), we obtain the following inequality:

$$\sum_{k \ge \underline{k}}^{K_n} \frac{p_k}{z_k} \le \frac{\beta_{n-1}}{2\sigma^2} + \frac{a_{\lambda_{\max}} \sqrt{p_k}}{2\lambda_{\max}} + \frac{2LB}{\sigma^2} \sqrt{\gamma_n} \Psi_n + \frac{2B^2 L^2}{\sigma^2} \sqrt{dS_n \log(dS_n)}$$
$$+ \left( 3BL + \lambda_{\max} + \lambda_{\max} \frac{4B^2 L^2}{\sigma^2} \right) \frac{M_n}{2\lambda_{\max}}$$
$$+ 2BL \left( 1 + \frac{3\lambda_{\max} BL}{\sigma^2} \right) \frac{\zeta_n}{2\lambda_{\max}} + \left( 2 + \frac{8BL\lambda_{\max}}{\sigma^2} \right) \frac{\sqrt{\gamma_n} \Psi_n}{2\lambda_{\max}}.$$

Recall that, by definition, $S_n = \sum_{k=0}^{K_n} p_k$. Furthermore, by Cauchy-Schwartz inequality, $\sum_{k=0}^{K_n} \sqrt{p_k} \le \sqrt{K_n \sum_{k=0}^{K_n} p_k}$. Simplifying this a little,

$$\sum_{k \ge \underline{k}}^{K_n} \frac{p_k}{z_k} \le \frac{\beta_{n-1}}{2\sigma^2} + \mathcal{O} \left( \sqrt{K_n \sum_{k=0}^{K_n} p_k} \right) + \mathcal{O} \left( \sqrt{\left( \sum_{k=0}^{K_n} p_k \right) \log \left( \sum_{k=0}^{K_n} p_k \right)} \right). \tag{50}$$

### G.3.6 Choosing $z_k$ and $p_k$

We choose the exponential schedule $z_k = z_0 e^k$ and $p_k = z_k e^{rk}$, where $r$ will be specified later. The left-hand side of (50) is

$$\sum_{k \geq \underline{k}}^{K_n} \frac{p_k}{z_k} = \sum_{k \geq \underline{k}}^{K_n} e^{rk} \geq e^{rK_n},$$

while the right-hand side is

$$\frac{\beta_{n-1}}{2\sigma^2} + \mathcal{O}\left(\sqrt{K_n \sum_{k=0}^{K_n} e^{(r+1)k}}\right) + \mathcal{O}\left(\sqrt{\left(\sum_{k=0}^{K_n} e^{(r+1)k}\right)\log\left(\sum_{k=0}^{K_n} e^{(r+1)k}\right)}\right)$$

$$\leq \frac{\beta_{n-1}}{2\sigma^2} + \mathcal{O}\left(\sqrt{K_n^2 e^{(r+1)K_n}}\right).$$

For $r > 1$, the resulting inequality yields $K_n \leq \mathcal{O}(\frac{1}{r}\log\beta_{n-1})$, i.e., $K_n \leq \mathcal{O}(\frac{1}{r}\log\log n)$ by definition of $\beta_{n-1}$. Let us recall (45):

$$\mathbb{E}\left[R_n^{\mathrm{explore}}\right] \leq \underbrace{2BL\sum_{k<\underline{k}} p_k}_{\mathrm{I}} + \underbrace{BL\underline{z}(\theta^\star)\sum_{k\geq\underline{k}:z_k<\bar{z}} \frac{p_k}{z_k - \underline{z}(\theta^\star)}}_{\mathrm{II}} + \underbrace{v^\star(\theta^\star)\frac{\beta_{n-1}}{2\sigma^2}}_{\mathrm{III}} + \underbrace{a_\lambda\mathbb{E}\left[\sum_{k\geq\underline{k}}^{K_n}\sqrt{p_k}\right]}_{\mathrm{IV}}$$

$$+ \underbrace{BL\underline{z}(\theta)z^\star(\theta^\star)\mathbb{E}\left[\sum_{k:z_k\geq\bar{z}(\theta^\star)}^{K_n}\frac{p_k}{z_k - \underline{z}(\theta^\star)}\max\left\{\frac{c_\Theta\sqrt{2}}{\sigma\sqrt{z_k}}, \frac{1}{z_k}\right\}\right]}_{\mathrm{V}} + \underbrace{\mathbb{E}\left[\mathcal{O}(\sqrt{S_n\log S_n})\right]}_{\mathrm{VI}}. \quad (51)$$

We bound the remaining terms separately.

**Term I**

$$\sum_{k<\underline{k}} p_k = z_0\sum_{k<\underline{k}} e^{(r+1)k} \leq z_0 e^{(r+1)\log\left(\frac{2\underline{z}(\theta^\star)}{z_0}\right)}\log(2\underline{z}(\theta^\star)/z_0) = z_0(2\underline{z}(\theta^\star)/z_0)^{r+1}\log(2\underline{z}(\theta^\star)/z_0),$$

where we used that, from the definition of $\underline{k}$ and $z_k$, it must be that $k < \log(2\underline{z}(\theta^\star)/z_0)$. Thus,

$$\mathrm{I} \leq 2BLz_0(2\underline{z}(\theta^\star)/z_0)^{r+1}\log(2\underline{z}(\theta^\star)/z_0).$$

**Term II**

$$\sum_{k\geq\underline{k}:z_k<\bar{z}(\theta^\star)} \frac{p_k}{z_k - \underline{z}(\theta^\star)} = \sum_{\log\left(\frac{2\underline{z}(\theta^\star)}{z_0}\right)\leq k<\log\left(\frac{\bar{z}(\theta^\star)}{z_0}\right)} \frac{z_0 e^{(r+1)k}}{z_0 e^k - \underline{z}(\theta^\star)}$$

$$= \sum_{\log\left(\frac{2\underline{z}(\theta^\star)}{z_0}\right)\leq k<\log\left(\frac{\bar{z}(\theta^\star)}{z_0}\right)} \underbrace{\frac{z_0 e^k}{z_0 e^k - \underline{z}(\theta^\star)}}_{\leq 2} e^{rk} \leq 2(\bar{z}(\theta^\star)/z_0)^r\log(\bar{z}(\theta^\star)/z_0).$$

Thus,

$$\mathrm{II} \leq 2BL\underline{z}(\theta^\star)(\bar{z}(\theta^\star)/z_0)^r\log(\bar{z}(\theta^\star)/z_0).$$

**Term IV** The total number of exploration rounds is

$$S_n = \sum_{k=0}^{K_n} p_k = z_0\sum_{k=0}^{K_n} e^{(r+1)k} \leq z_0 e^{(r+1)(K_n+1)} \leq \mathcal{O}((\log n)^{\frac{r+1}{r}}).$$

Therefore,

$$\mathrm{IV} \leq \sqrt{K_n\sum_{k=0}^{K_n} p_k} \leq \mathcal{O}((\log\log n)^{1/2}(\log n)^{\frac{r+1}{2r}}).$$

**Term V** We consider two cases, based on which of the inner terms is the maximum. In the first case, we need to bound

$$\sum_{k:z_k\geq\bar{z}(\theta^\star)}\frac{p_k}{(z_k-\underline{z}(\theta^\star))\sqrt{z_k}}=\sum_{k\geq\log\left(\frac{\bar{z}(\theta^\star)}{z_0}\right)}^{K_n}\frac{z_0e^{(r+1)k}}{(z_0e^k-\underline{z}(\theta^\star))\sqrt{z_0e^k}}$$

$$=\frac{1}{\sqrt{z_0}}\sum_{k\geq\log\left(\frac{\bar{z}(\theta^\star)}{z_0}\right)}^{K_n}\underbrace{\frac{z_0e^k}{(z_0e^k-\underline{z}(\theta^\star))}}_{\leq 2}e^{(r-1/2)k}\leq\frac{2}{\sqrt{z_0}}\sum_{k\geq\log\left(\frac{\bar{z}(\theta^\star)}{z_0}\right)}^{K_n}e^{(r-1/2)k}$$

$$\leq\frac{2}{\sqrt{z_0}}\int_{\log\left(\frac{\bar{z}(\theta^\star)}{z_0}\right)}^{K_n+1}e^{(r-1/2)k}\mathrm{d}k=\frac{2}{\sqrt{z_0}}\left[\frac{e^{(r-1/2)k}}{r-1/2}\right]_{\log\left(\frac{\bar{z}(\theta^\star)}{z_0}\right)}^{K_n+1}$$

$$=\frac{2}{(r-1/2)\sqrt{z_0}}\left(e^{(r-1/2)(K_n+1)}-(\bar{z}(\theta^\star)/z_0)^{r-1/2}\right).$$

Since $K_n\leq\mathcal{O}(\frac{1}{r}\log\log n)$, this term is $\mathcal{O}((\log n)^{\frac{r-1/2}{r}})$. If the other term is the maximum, then the same procedure yields a $\mathcal{O}((\log n)^{\frac{r-1}{r}})$ dependency. Thus,

$$\mathrm{V}\leq\mathcal{O}((\log n)^{\frac{r-1/2}{r}}).$$

**Term VI** We have $\mathrm{VI}\leq\mathcal{O}((\log n)^{\frac{r+1}{2r}})$ as in Term IV.

**Final Bound** Using $r=2$, we obtain the following bound on the expected regret during exploration:

$$\mathbb{E}\left[R_n^{\text{explore}}\right]\leq 2BLz_0(2\underline{z}(\theta^\star)/z_0)^3\log(2\underline{z}(\theta^\star)/z_0)$$

$$+2BL\underline{z}(\theta^\star)(\bar{z}(\theta^\star)/z_0)^2\log(\bar{z}(\theta^\star)/z_0)+v^\star(\theta^\star)\frac{\beta_{n-1}}{2\sigma^2}+\mathcal{O}((\log\log n)^{\frac{1}{2}}(\log n)^{\frac{3}{4}}),$$

which is asymptotically optimal.

## H Worst-case Analysis (Proof of Thm. 3)

### H.1 Outline

The proof follows a similar argument as the one of Thm. 2 but it is considerably simpler and shorter. In particular, the main simplifications come from two worst-case arguments. (1) While bounding the regret during exploration rounds, we use the naive bound $S_n\leq n$. This is equivalent to assuming that SOLID never enters the exploitation step and it allows us to entirely avoid the bound on the number of phases of App. G.3.5. (2) We completely ignore the sequence $z_k$ and proceed as if the optimization problem $(\mathrm{P}_z)$ was infeasible in all phases. This makes the multiplier saturate to $\lambda_{\max}$ and facilitate the analysis of the resulting Lagrangian[13]. An outline of the proof, together with the main differences w.r.t. the one of Thm. 2, is as follows.

1. We decompose the regret suffered during exploitation and exploration rounds. Using the same steps as in App. G, we bound the former by a constant and reduce the latter to the sum of objective values.

2. Instead of relating to the objective values of the optimal policies $\omega_{z_k}^\star$ at each phase $k$ (as was done in App. G.3.2, we reduce our bound to the optimal solution of our bandit problem, i.e., the policy that only pulls optimal arms. This makes the sum of objective values cancel since the optimal policy achieves zero regret.

3. Using the results of App. G.3.3, we show that the sum of constraints is $\mathcal{O}(\log n)$.

4. We use the naive bound $S_n\leq n$ to conclude the proof.

## H.2 Proof

We start from the same regret decomposition as in App. G,

$$R_n = \sum_{t=1}^n \Delta_{\theta^\star}(X_t, A_t) \mathbb{1}\left\{\neg E_t\right\} + \sum_{t=1}^n \Delta_{\theta^\star}(X_t, A_t) \mathbb{1}\left\{E_t\right\} = R_n^{\text{exploit}} + R_n^{\text{explore}}.$$

The regret suffered during the exploitation rounds was bounded in App. G.2 as $\mathbb{E}\left[R_n^{\text{exploit}}\right] \leq 2LB$. Regarding the regret suffered during the exploration rounds, we have

$$R_n^{\text{explore}} := \sum_{t=1}^n \Delta_{\theta^\star}(X_t, A_t) \mathbb{1}\left\{E_t\right\} \leq \sum_{t=1}^n \Delta_{\theta^\star}(X_t, A_t) \mathbb{1}\left\{E_t, G_t\right\} + 2LB \underbrace{\sum_{t=1}^n \mathbb{1}\left\{E_t, \neg G_t\right\}}_{:=M_n}.$$

$$(52)$$

Refer to App. F for the definition of $G_t$. The second term is $M_n$, the number of exploration rounds in which the good event does not hold, and can be bounded in expectation by using Lem. 6. The first one can be bounded by using the good event. Suppose, without loss of generality, that $E_n$ and $G_n$ hold (if they do not, the following reasoning can be repeated for the last time step at which these events hold). Then, using $G_t^\Delta$ (see App. F),

$$\sum_{t=1}^n \Delta_{\theta^\star}(X_t, A_t) \mathbb{1}\left\{E_t, G_t\right\} \leq \sum_{t \leq n: E_t} \Delta_{\theta^\star}(X_t, A_t)$$

$$\leq \sum_{t \leq n: E_t} \sum_{x \in \mathcal{X}} \rho(x) \sum_{a \in \mathcal{A}} \omega_t(x, a) \Delta_{\theta^\star}(x, a) + 2LB\sqrt{S_n \log S_n}. \quad (53)$$

We now proceed using similar steps as in App. G.3.1, except that we ignore the phases. We decompose the first term as

$$\sum_{t \leq n: E_t} \sum_{x \in \mathcal{X}} \rho(x) \sum_{a \in \mathcal{A}} \omega_t(x, a) \Delta_{\theta^\star}(x, a)$$

$$= \sum_{t \leq n: E_t, G_t} \sum_{x \in \mathcal{X}} \rho(x) \sum_{a \in \mathcal{A}} \omega_t(x, a) \Delta_{\theta^\star}(x, a) + \sum_{t \leq n: E_t, \neg G_t} \sum_{x \in \mathcal{X}} \rho(x) \sum_{a \in \mathcal{A}} \omega_t(x, a)(\mu_{\theta^\star}^\star(x) - \mu_{\theta^\star}(x, a))$$

$$\leq \sum_{t \leq n: E_t, G_t} \sum_{x \in \mathcal{X}} \rho(x) \sum_{a \in \mathcal{A}} \omega_t(x, a) \Delta_{\theta^\star}(x, a) + M_n \mu^\star - \sum_{t \leq n: E_t, \neg G_t} \sum_{x \in \mathcal{X}} \rho(x) \sum_{a \in \mathcal{A}} \omega_t(x, a)\mu_{\theta^\star}(x, a).$$

Here we defined

$$\mu^\star := \sum_{x \in \mathcal{X}} \rho(x)\mu_{\theta^\star}^\star(x). \quad (54)$$

The last term can be bounded by $M_n BL$. Regarding the remaining two,

$$\sum_{t \leq n: E_t, G_t} \sum_{x \in \mathcal{X}} \rho(x) \sum_{a \in \mathcal{A}} \omega_t(x, a) \Delta_{\theta^\star}(x, a) + M_n \mu^\star$$

$$= (S_n - M_n)\mu^\star + M_n \mu^\star - \sum_{t \leq n: E_t, G_t} \sum_{x \in \mathcal{X}} \rho(x) \sum_{a \in \mathcal{A}} \omega_t(x, a)\mu_{\theta^\star}(x, a)$$

$$= S_n \mu^\star + \underbrace{\sum_{t \leq n: E_t, G_t} \sum_{x \in \mathcal{X}} (\widehat{\rho}_{t-1}(x) - \rho(x)) \sum_{a \in \mathcal{A}} \omega_t(x, a)\mu_{\theta^\star}(x, a)}_{(a)} - \underbrace{\sum_{t \leq n: E_t, G_t} \sum_{x \in \mathcal{X}} \widehat{\rho}_{t-1}(x) \sum_{a \in \mathcal{A}} \omega_t(x, a)\mu_{\theta^\star}(x, a)}_{(b)}.$$

Term (a) can be bounded as

$$(a) \leq LB \underbrace{\sum_{t \leq n: E_t, G_t} \sum_{x \in \mathcal{X}} |\widehat{\rho}_{t-1}(x) - \rho(x)|}_{\zeta_n}.$$

For the sake of readability, we keep the dependence on $\zeta_n$ explicit. We will bound this term by Lem. 12 at the end of the proof. Regarding term (b), using the definition of $G_t$ and Lem. 10,

$$(b) \geq \sum_{t \leq n: E_t, G_t} \sum_{x \in \mathcal{X}} \widehat{\rho}_{t-1}(x) \sum_{a \in \mathcal{A}} \omega_t(x,a) \left( \mu_{\widetilde{\theta}_{t-1}}(x,a) - \sqrt{\gamma_t} \|\phi(x,a)\|_{\bar{V}_{t-1}^{-1}} \right)$$

$$\pm \sum_{t \leq n: E_t, \neg G_t} \sum_{x \in \mathcal{X}} \widehat{\rho}_{t-1}(x) \sum_{a \in \mathcal{A}} \omega_t(x,a) \underbrace{\mu_{\widetilde{\theta}_{t-1}}(x,a)}_{|\cdot| \leq LB} \pm \sum_{t \leq n: E_t} \sum_{x \in \mathcal{X}} \widehat{\rho}_{t-1}(x) \sum_{a \in \mathcal{A}} \omega_t(x,a) \sqrt{\gamma_t} \|\phi(x,a)\|_{\bar{V}_{t-1}^{-1}}$$

$$\geq \sum_{t \leq n: E_t} f_t(\omega_t) - M_n BL - 2 \sum_{t \leq n: E_t} \sum_{x \in \mathcal{X}} \widehat{\rho}_{t-1}(x) \sum_{a \in \mathcal{A}} \omega_t(x,a) \sqrt{\gamma_t} \|\phi(x,a)\|_{\bar{V}_{t-1}^{-1}}$$

$$\geq \sum_{t \leq n: E_t} f_t(\omega_t) - M_n BL - 2\sqrt{\gamma_n} \Psi_n.$$

We recall that $\sqrt{\gamma_t} \leq \sqrt{\gamma_n}$ and $\Psi_n := \sum_{t \leq n: E_t} \sum_{x \in \mathcal{X}} \widehat{\rho}_{t-1}(x) \sum_{a \in \mathcal{A}} \omega_t(x,a) \|\phi(x,a)\|_{\bar{V}_{t-1}^{-1}}$. As for $\zeta_n$, we keep the dependence on $\Psi_n$ explicit and defer bounding this term to the end of the proof. Using the bounds on (a) and (b) and plugging everything back into (53) and then into (52), we obtain

$$R_n^{\text{explore}} \leq S_n \mu^\star - \sum_{t \leq n: E_t} f_t(\omega_t) + 4M_n BL + \zeta_n BL + 2\sqrt{\gamma_n} \Psi_n + 2BL\sqrt{S_n \log S_n}. \quad (55)$$

We now lower bound the sum of objective values. Here we proceed in a slightly different way with respect to the proof of the asymptotically optimal regret bound. Instead of relating to the objective values of the optimal policies $\omega_{z_k}^\star$ at each phase $k$, we reduce our bound to the optimal solution of our bandit problem, i.e., the policy that only pulls optimal arms. Let

$$\omega_{\theta^\star}^\star(x,a) := \begin{cases} 1 & \text{if } a = a_{\theta^\star}^\star(x) \\ 0 & \text{otherwise} \end{cases} \quad (56)$$

Recall that $\sum_{t \leq n: E_t} f_t(\omega_t) = \sum_{k=0}^{K_n} \sum_{t \in \mathcal{T}_k^E} f_t(\omega_t)$. Fix some phase index $k \geq 0$ and let $\lambda \geq 0$ be arbitrary. Using Corollary 2 with $\alpha_k^\lambda = \alpha_k^\omega = 1/\sqrt{p_k}$ and $\omega = \omega_{\theta^\star}^\star$,

$$\sum_{t \in \mathcal{T}_k^E} f_t(\omega_t) \geq \sum_{t \in \mathcal{T}_k^E} h_t(\omega_{\theta^\star}^\star, \lambda_t, z_k) - \lambda \sum_{t \in \mathcal{T}_k^E} g_t(\omega_t, z_k) - a_\lambda \sqrt{p_k}, \quad (57)$$

where $a_\lambda := \left( \log|\mathcal{A}| + \frac{b_\omega^2 + b_\lambda^2}{2} + \frac{(\lambda - \lambda_1)^2}{2} \right)$ and $b_\lambda$ and $b_\omega$ are the maximum sub-gradients in $\lambda$ and $\omega$, respectively. Note that, since we apply Corollary 2 to bound the sum of objective values over the whole phase, we have $S_{n,k} = p_k$. We now lower-bound the first term on the right-hand side. We have

$$\sum_{t \in \mathcal{T}_k^E} h_t(\omega_{\theta^\star}^\star, \lambda_t, z_k) \overset{(c)}{=} \sum_{t \in \mathcal{T}_k^E} f_t(\omega_{\theta^\star}^\star) + \sum_{t \in \mathcal{T}_k^E} \lambda_t g_t(\omega_{\theta^\star}^\star, z_k) \overset{(d)}{\geq} \sum_{t \in \mathcal{T}_k^E} f_t(\omega_{\theta^\star}^\star) - \sum_{t \in \mathcal{T}_k^E} \frac{\lambda_t}{z_k}$$

$$\overset{(e)}{\geq} \sum_{t \in \mathcal{T}_k^E} f_t(\omega_{\theta^\star}^\star) - \frac{\lambda_{\max} S_{n,k}}{z_k}, \quad (58)$$

where (c) uses the definition of $h_t$ and $g_t$ (see Eq. 3 and Eq. 4), (d) uses the positivity of KL divergences and confidence intervals, and (e) uses $\lambda_t \leq \lambda_{\max}$ and $S_{n,k} := |\mathcal{T}_k^E|$. Let us focus on the sum of objective values. Since $f_t(\omega_{\theta^\star}^\star) \geq -LB$, we have $\sum_{t \in \mathcal{T}_k^E: \neg G_t} f_t(\omega_{\theta^\star}^\star) \geq -M_{n,k} BL$. For any step $t \in \mathcal{T}_k^E$ in which $G_t$ holds, the optimism property (see App. F and Lem. 11) yields

$$\sum_{t \in \mathcal{T}_k^E: G_t} f_t(\omega_{\theta^\star}^\star) \geq \sum_{t \in \mathcal{T}_k^E: G_t} \sum_{x \in \mathcal{X}} \widehat{\rho}_{t-1}(x) \sum_{a \in \mathcal{A}} \omega_{\theta^\star}^\star(x,a) \mu_{\theta^\star}(x,a)$$

$$= \sum_{t \in \mathcal{T}_k^E: G_t} \sum_{x \in \mathcal{X}} (\widehat{\rho}_{t-1}(x) - \rho(x)) \underbrace{\sum_{a \in \mathcal{A}} \omega_{\theta^\star}^\star(x,a) \mu_{\theta^\star}(x,a)}_{|\cdot| \leq BL} + \sum_{t \in \mathcal{T}_k^E: G_t} \underbrace{f(\omega_{\theta^\star}^\star)}_{=\mu^\star}$$

$$\geq (S_{n,k} - M_{n,k}) \mu^\star - BL \underbrace{\sum_{t \in \mathcal{T}_k^E: G_t} \sum_{x \in \mathcal{X}} |\widehat{\rho}_{t-1}(x) - \rho(x)|}_{:=\zeta_{n,k}},$$

where we used the fact that $f(\omega_{\theta^\star}^\star) = \mu^\star$ by definition (56) and (54) and $\sum_{t \in \mathcal{T}_k^E} \mathbb{1}\{G_t\} = \sum_{t \in \mathcal{T}_k} \mathbb{1}\{E_t\} - \sum_{t \in \mathcal{T}_k} \mathbb{1}\{E_t, \neg G_t\} = S_{n,k} - M_{n,k}$. Plugging this back into (58) and then into (57),

$$\sum_{t \in \mathcal{T}_k^E} f_t(\omega_t) \geq (S_{n,k} - M_{n,k})\mu^\star - BL\zeta_{n,k} - \frac{\lambda_{\max}S_{n,k}}{z_k} - \lambda \sum_{t \in \mathcal{T}_k^E} g_t(\omega_t, z_k) - a_\lambda\sqrt{p_k} - M_{n,k}BL.$$

Summing over all phases and recalling that $\sum_{k=0}^{K_n} S_{n,k} = S_n$, $\sum_{k=0}^{K_n} M_{n,k} = M_n$, and $\sum_{k=0}^{K_n} \zeta_{n,k} = \zeta_n$, we obtain

$$\sum_{t \leq n:E_t} f_t(\omega_t) \geq (S_n - M_n)\mu^\star - BL\zeta_n - \sum_{k=0}^{K_n} \frac{\lambda_{\max}S_{n,k}}{z_k} - \lambda \sum_{t \leq n:E_t} g_t(\omega_t, z_{K_t}) - a_\lambda \sum_{k=0}^{K_n} \sqrt{p_k} - M_nBL. \tag{59}$$

Using the definition of $g_t$ (see Eq. 3),

$$\sum_{t \leq n:E_t} g_t(\omega_t, z_{K_t}) := \sum_{t \leq n:E_t} \inf_{\theta' \in \overline{\Theta}_{t-1}} \sum_{x \in \mathcal{X}} \widehat{\rho}_{t-1}(x) \sum_{a \in \mathcal{A}} \omega_t(x,a) \left( d_{x,a}\left(\widetilde{\theta}_{t-1}, \theta'\right) + \frac{2LB}{\sigma^2}\sqrt{\gamma_t}\|\phi(x,a)\|_{\overline{V}_{t-1}^{-1}} \right)$$
$$- \sum_{t \leq n:E_t} \frac{1}{z_{K_t}}.$$

By the definition of phase, the second term is $\sum_{t \leq n:E_t} \frac{1}{z_{K_t}} = \sum_{k=0}^{K_n} \frac{S_{n,k}}{z_k}$. The first term can be bounded using exactly the same steps as in App. G.3.3.[14] We obtain

$$\sum_{t \leq n:E_t} g_t(\omega_t, z_{K_t}) \leq \frac{\beta_{n-1}}{2\sigma^2} - \sum_{k=0}^{K_n} \frac{S_{n,k}}{z_k} + \frac{2L^2B^2}{\sigma^2}M_n + \frac{6LB}{\sigma^2}\sqrt{\gamma_n}\Psi_n + \frac{2L^2B^2}{\sigma^2}\zeta_n$$
$$+ \frac{2B^2L^2}{\sigma^2}\left(\sqrt{dS_n\log(dS_n)} + 1\right). \tag{60}$$

If we now set $\lambda = \lambda_{\max}$ and plug (60) into (59),

$$\sum_{t \leq n:E_t} f_t(\omega_t) \geq (S_n - M_n)\mu^\star - BL\left(1 + \frac{2\lambda_{\max}BL}{\sigma^2}\right)(\zeta_n + M_n) + \underbrace{\sum_{k=0}^{K_n} \frac{\lambda_{\max}S_{n,k}}{z_k} - \sum_{k=0}^{K_n} \frac{\lambda_{\max}S_{n,k}}{z_k}}_{=0}$$

$$- \frac{\lambda_{\max}\beta_{n-1}}{2\sigma^2} - a_{\lambda_{\max}}\sum_{k=0}^{K_n}\sqrt{p_k} - \frac{6\lambda_{\max}LB}{\sigma^2}\sqrt{\gamma_n}\Psi_n - \frac{2\lambda_{\max}B^2L^2}{\sigma^2}\left(\sqrt{dS_n\log(dS_n)} + 1\right).$$

We can finally plug this into (55), thus obtaining

$$R_n^{\text{explore}} \leq M_n \underbrace{\mu^\star}_{|\cdot| \leq BL} + BL\left(5 + \frac{2\lambda_{\max}BL}{\sigma^2}\right)(\zeta_n + M_n) + \frac{\lambda_{\max}\beta_{n-1}}{2\sigma^2} + a_{\lambda_{\max}}\sum_{k=0}^{K_n}\sqrt{p_k}$$

$$+ \left(2 + \frac{6\lambda_{\max}LB}{\sigma^2}\right)\sqrt{\gamma_n}\Psi_n + \frac{2\lambda_{\max}B^2L^2}{\sigma^2}\left(\sqrt{dS_n\log(dS_n)} + 1\right) + 2BL\sqrt{S_n\log S_n}.$$

Let $\bar{k}_n := \min\{k : p_k \geq n\}$, then $K_n \leq \bar{k}_n$. Using the exponential schedule $p_k = e^{rk}$, $\bar{k}_n = \lceil\frac{1}{r}\log n\rceil$ and

$$\sum_{k=0}^{K_n}\sqrt{p_k} \leq \sum_{k=0}^{\bar{k}_n} e^{\frac{r}{2}k} \leq \int_0^{\bar{k}_n+1} e^{\frac{r}{2}x}\mathrm{d}x = \left[\frac{2}{r}e^{\frac{r}{2}x}\right]_0^{\bar{k}_n+1} = \frac{2}{r}e^{\frac{r}{2}(\lceil\frac{1}{r}\log n\rceil+1)} - \frac{2}{r} \leq \frac{2e^r}{r}\sqrt{n}.$$

Taking expectations of both sides of the regret bound above and using $S_n \leq n$ and $\mathbb{E}[M_n] \leq \frac{3\pi^2}{2}$ by Lem. 6,

$$\mathbb{E}\left[R_n^{\text{explore}}\right] \leq \frac{3BL\pi^2}{2}\left(6 + \frac{2\lambda_{\max}BL}{\sigma^2}\right) + \frac{\lambda_{\max}\beta_{n-1}}{2\sigma^2} + \frac{2e^r a_{\lambda_{\max}}}{r}\sqrt{n} + 2BL\sqrt{n\log n}$$

$$+ \left(2 + \frac{6\lambda_{\max}LB}{\sigma^2}\right)\mathbb{E}\left[\sqrt{\gamma_n}\Psi_n\right] + \frac{2\lambda_{\max}B^2L^2}{\sigma^2}\left(\sqrt{nd\log(nd)} + 1\right) + BL\left(5 + \frac{2\lambda_{\max}BL}{\sigma^2}\right)\mathbb{E}[\zeta_n].$$

After bounding $S_n \leq n$, by Lem. 13, $\Psi_n \leq \mathcal{O}\left(L|\mathcal{X}|\sqrt{n\log n} + \sqrt{nd\log n}\right)$ while, by Lem. 12, $\zeta_n \leq \mathcal{O}\left(|\mathcal{X}|\sqrt{n\log n}\right)$. Therefore, recalling that the regret during exploitation rounds was bounded by $2BL$ and noting that $2 < \frac{3\pi^2}{2}$,

$$\mathbb{E}[R_n] \leq 3BL\pi^2\left(4 + \frac{\lambda_{\max}BL}{\sigma^2}\right) + \frac{2e^r\lambda_{\max}^2}{r}\sqrt{n} + C_{\text{sqrt}}\left(1 + \frac{\lambda_{\max}BL}{\sigma^2}\right)\log(n)\sqrt{n},$$

where $C_{\text{sqrt}} = lin_{\geq 0}(|\mathcal{X}|, \sqrt{d}, B, L)$. Here we included $\frac{\lambda_{\max}\beta_{n-1}}{2\sigma^2}$ and the components of $a_{\lambda_{\max}}$ (except $\lambda_{\max}^2$ which is kept explicit) into the last term above. This concludes the proof.

# I  Auxiliary Results

## I.1  Concentration Inequalities

**Lemma 7** (Concentration of $\rho$ during exploration). *For any context $x \in \mathcal{X}$,*

$$\sum_{t \geq 1}\sum_{x \in \mathcal{X}}\mathbb{P}\left\{E_t, |\widehat{\rho}_t(x) - \rho(x)| > \sqrt{\frac{\log(|\mathcal{X}|S_t^2)}{2S_t}}\right\} \leq \frac{\pi^2}{3}. \tag{61}$$

*Proof.* The proof follows Lem. B.1 in [27]. Fix some $\bar{t} \geq 1$ and $x \in \mathcal{X}$. Then,

$$\sum_{t=1}^{\bar{t}}\mathbb{1}\left\{E_t, |\widehat{\rho}_t(x) - \rho(x)| > \sqrt{\frac{\log(|\mathcal{X}|S_t^2)}{2S_t}}\right\} \leq \sum_{s \geq 1}\mathbb{1}\left\{|\widehat{\rho}_{\tau_s}(x) - \rho(x)| > \sqrt{\frac{\log(|\mathcal{X}|s^2)}{2s}}, \tau_s \leq \bar{t}\right\}.$$

where $\tau_s$ is the random time the $s$-th exploration round occurs. Thus, by taking the expectation of both sides,

$$\sum_{t=1}^{\bar{t}}\mathbb{P}\left\{E_t, |\widehat{\rho}_t(x) - \rho(x)| > \sqrt{\frac{\log(|\mathcal{X}|S_t^2)}{2S_t}}\right\} \leq \sum_{s \geq 1}\mathbb{P}\left\{|\widehat{\rho}_{\tau_s}(x) - \rho(x)| > \sqrt{\frac{\log(|\mathcal{X}|s^2)}{2s}}, \tau_s \leq \bar{t}\right\}.$$

Since $\tau_s$ is a stopping-time upper bounded by $\bar{t}$ and the number of samples used to compute $\widehat{\rho}_{\tau_s}(x)$ is at least $s$, we can apply Lemma 4.3 of [27]:

$$\sum_{t=1}^{\bar{t}}\mathbb{P}\left\{E_t, |\widehat{\rho}_t(x) - \rho(x)| > \sqrt{\frac{\log(|\mathcal{X}|S_t^2)}{2S_t}}\right\} \leq \sum_{s \geq 1} 2e^{-2s\frac{\log(|\mathcal{X}|s^2)}{2s}} = \frac{2}{|\mathcal{X}|}\sum_{s \geq 1}\frac{1}{s^2} = \frac{\pi^2}{3|\mathcal{X}|}.$$

The reasoning above holds for any $\bar{t}$ and $x \in \mathcal{X}$. Summing over $\mathcal{X}$ concludes the proof. $\square$

**Lemma 8** (Confidence set for exploration). *With some abuse of notation, let $\gamma_t := c_{n,1/S_t^2}$. Then, under the same conditions as in Theorem 1,*

$$\sum_{t=1}^{n}\mathbb{P}\left\{E_t, \|\widehat{\theta}_{t-1} - \theta^\star\|_{\overline{V}_{t-1}} > \sqrt{\gamma_t}\right\} \leq \frac{\pi^2}{6}.$$

*Proof.* Let $\{\tau_s\}_{s \geq 1}$ be a sequence of stopping times with respect to $\mathcal{F}$ such that if $\tau_s = t$, then the $s$-th exploration round occurs at time $t + 1$. Then,

$$\sum_{t=1}^{n}\mathbb{1}\left\{E_t, \|\widehat{\theta}_{t-1} - \theta^\star\|_{\overline{V}_{t-1}} > \sqrt{\gamma_t}\right\} \leq \sum_{s \geq 1}\mathbb{1}\left\{\|\widehat{\theta}_{\tau_s} - \theta^\star\|_{\overline{V}_{\tau_s}} > \sqrt{\gamma_{\tau_s+1}}, \tau_s \leq n\right\}. \tag{62}$$

Since $S_{\tau_s+1} = s$, we have $\gamma_{\tau_s+1} = c_{n,1/s^2}$. Taking expectations and applying Theorem 1,

$$\sum_{s \geq 1} \mathbb{P}\left\{ \|\widehat{\theta}_{\tau_s} - \theta^\star\|_{\overline{V}_{\tau_s}} > \sqrt{\gamma_{\tau_s+1}}, \tau_s \leq n \right\} \leq \sum_{s \geq 1} \frac{1}{s^2} = \frac{\pi^2}{6}.$$

$\square$

## I.2  Supporting Lemmas

The following result shows that any projection onto a non-empty convex set using a norm weighted by a positive definite matrix is a *non-expansion*. That is, the distance (in the chosen weighted norm) between the projected vector and any point in the set cannot increase w.r.t. the unprojected vector. We are not sure about a suitable citation for this result, so we include its proof.

**Lemma 9** (Non-expansion of weighted projection). *Let $\widehat{\theta} \in \mathbb{R}^d$ be any vector, $V \in \mathbb{R}^{d \times d}$ be a positive definite matrix, and $\mathcal{B} \subset \mathbb{R}^d$ be a non-empty convex set. Let $\widetilde{\theta}$ be the weighted projection of $\widehat{\theta}$ onto $\mathcal{B}$,*

$$\widetilde{\theta} := \underset{\theta \in \mathcal{B}}{\operatorname{argmin}} \|\theta - \widehat{\theta}\|_V. \tag{63}$$

*Then, for all $\theta \in \mathcal{B}$,*

$$\|\widetilde{\theta} - \theta\|_V \leq \|\widehat{\theta} - \theta\|_V. \tag{64}$$

*Proof.* Let $f : \mathbb{R}^d \to \mathbb{R}$ be defined as $f(x) := \|x - \widehat{\theta}\|_V^2$, so that $\widetilde{\theta} = \operatorname{argmin}_{x \in \mathcal{B}} f(x)$. Note that $f$ is a convex function that is differentiable on $\mathbb{R}^d$. Therefore, using the first-order optimality conditions for convex functions (see, e.g., Theorem 2.8 in [28]), we have $\widetilde{\theta} = \operatorname{argmin}_{x \in \mathcal{B}} f(x)$ if and only if

$$\forall \theta \in \mathcal{B} : \langle \nabla f(\widetilde{\theta}), \theta - \widetilde{\theta} \rangle \geq 0. \tag{65}$$

Since $\nabla f(x) = 2V(x - \widehat{\theta})$,

$$\forall \theta \in \mathcal{B} : \langle V(\widetilde{\theta} - \widehat{\theta}), \theta - \widetilde{\theta} \rangle \geq 0. \tag{66}$$

Fix any $\theta \in \mathcal{B}$. We have

$$\|\widehat{\theta} - \theta\|_V^2 = \|\widehat{\theta} \pm \widetilde{\theta} - \theta\|_V^2 = \|\widehat{\theta} - \widetilde{\theta}\|_V^2 + \|\widetilde{\theta} - \theta\|_V^2 + 2(\widehat{\theta} - \widetilde{\theta})^T V(\widetilde{\theta} - \theta) \geq \|\widetilde{\theta} - \theta\|_V^2.$$

This concludes the proof. $\square$

**Corollary 1.** *Let $t \in [n]$ be any time step in which the good event $G_t$ holds. Then,*

$$\|\widetilde{\theta}_{t-1} - \theta^\star\|_{\overline{V}_{t-1}} \leq \|\widehat{\theta}_{t-1} - \theta^\star\|_{\overline{V}_{t-1}}. \tag{67}$$

*Proof.* If $G_t$ holds, then $\theta^\star \in \mathcal{C}_{t-1}$. Since $\|\theta^\star\|_2 \leq B$ by definition, the set $\mathcal{C}_{t-1} \cap \Theta$ is non-empty (it contains $\theta^\star$ itself). Then, the result follows from Lem. 9. $\square$

The following result is immediate from the definition of good event and the non-expansion property of the projection used to compute $\widetilde{\theta}_t$.

**Lemma 10.** *Let $t \in [n]$ be any time step in which the good event $G_t$ holds. Then,*

$$\forall x \in \mathcal{X}, a \in \mathcal{A} : |\mu_{\widetilde{\theta}_{t-1}}(x, a) - \mu_{\theta^\star}(x, a)| \leq \sqrt{\gamma_t}\|\phi(x, a)\|_{\overline{V}_{t-1}^{-1}}.$$

*Proof.* Fix any $x \in \mathcal{X}$ and $a \in \mathcal{A}$. Then,

$$|\mu_{\widetilde{\theta}_{t-1}}(x, a) - \mu_{\theta^\star}(x, a)| = |\phi(x, a)^T(\widetilde{\theta}_{t-1} - \theta^\star)| = |\phi(x, a)^T \bar{V}_{t-1}^{-1/2} \bar{V}_{t-1}^{1/2}(\widetilde{\theta}_{t-1} - \theta^\star)|$$

$$\overset{(a)}{\leq} \|\phi(x, a)\|_{\bar{V}_{t-1}^{-1}} \|\widetilde{\theta}_{t-1} - \theta^\star\|_{\bar{V}_{t-1}}$$

$$\overset{(b)}{\leq} \|\phi(x, a)\|_{\bar{V}_{t-1}^{-1}} \|\widehat{\theta}_{t-1} - \theta^\star\|_{\bar{V}_{t-1}} \overset{(c)}{\leq} \sqrt{\gamma_t}\|\phi(x, a)\|_{\bar{V}_{t-1}^{-1}},$$

where (a) is from Cauchy-Schwartz inequality, (b) from Cor. 1, and (c) from the definition of $G_t$. $\square$

**Lemma 11.** *Let $\gamma_t := c_{n,1/S_t^2}$ and $n \geq 3$. Then, for any time step $t$ in which the good event $G_t$ (see App. F) holds,*

$$f_t(\omega) := \sum_{x \in \mathcal{X}} \widehat{\rho}_{t-1}(x) \sum_{a \in \mathcal{A}} \omega(x,a) \left( \mu_{\widetilde{\theta}_{t-1}}(x,a) + \sqrt{\gamma_t} \|\phi(x,a)\|_{\overline{V}_{t-1}^{-1}} \right)$$

$$\geq \sum_{x \in \mathcal{X}} \widehat{\rho}_{t-1}(x) \sum_{a \in \mathcal{A}} \omega(x,a) \mu_{\theta^\star}(x,a), \tag{68}$$

*and*

$$g_t(\omega) := \inf_{\theta' \in \overline{\Theta}_{t-1}} \sum_{x \in \mathcal{X}} \widehat{\rho}_{t-1}(x) \sum_{a \in \mathcal{A}} \omega(x,a) \left( d_{x,a}\left(\widetilde{\theta}_{t-1}, \theta'\right) + \frac{2LB}{\sigma^2} \sqrt{\gamma_t} \|\phi(x,a)\|_{\overline{V}_{t-1}^{-1}} \right)$$

$$\geq \inf_{\theta' \in \Theta_{alt}} \sum_{x \in \mathcal{X}} \widehat{\rho}_{t-1}(x) \sum_{a \in \mathcal{A}} \omega(x,a) d_{x,a}(\theta^\star, \theta'). \tag{69}$$

*Proof.* Since $\widehat{\rho}$ and $\omega$ are non-negative, the first inequality is trivial by upper bounding the true mean $\mu_{\theta^\star}(x,a)$ for each $x, a$ by using the definition of $G_t^\theta$ and Lemma 10. Let us prove the second one. Fix any model $\theta' \in \Theta$. By using the definition of KL divergence of Gaussians with fixed variance, we have that:

$$d_{x,a}(\theta^\star, \theta') = \frac{(\mu_{\theta'}(x,a) - \mu_{\theta^\star}(x,a))^2}{2\sigma^2} \leq d_{x,a}(\widetilde{\theta}_{t-1}, \theta') + \frac{2LB}{\sigma^2} |\mu_{\widetilde{\theta}_{t-1}}(x,a) - \mu_{\theta^\star}(x,a)|$$

$$\leq d_{x,a}(\widetilde{\theta}_{t-1}, \theta') + \frac{2LB}{\sigma^2} \sqrt{\gamma_t} \|\phi(x,a)\|_{\overline{V}_{t-1}^{-1}},$$

where the first inequality is from $|(a-c)^2 - (b-c)^2| = |(a+b-2c)(a-b)| \leq 4LB|a-b|$ and the second one is once again from the definition of $G_t$ and Lemma 10. Therefore,

$$\inf_{\theta' \in \Theta_{alt}} \sum_{x \in \mathcal{X}} \widehat{\rho}_{t-1}(x) \sum_{a \in \mathcal{A}} \omega(x,a) d_{x,a}(\theta^\star, \theta') \tag{70}$$

$$\leq \inf_{\theta' \in \Theta_{alt}} \sum_{x \in \mathcal{X}} \widehat{\rho}_{t-1}(x) \sum_{a \in \mathcal{A}} \omega(x,a) \left( d_{x,a}(\widetilde{\theta}_{t-1}, \theta') + \frac{2LB}{\sigma^2} \sqrt{\gamma_t} \|\phi(x,a)\|_{\overline{V}_{t-1}^{-1}} \right).$$

We now upper bound the infimum over models in the alternative set. Note that such set can be fully specified once we assign an optimal arm to each context. Let $\{a_x\}_{x \in \mathcal{X}}$ and define

$$\Theta(\{a_x\}_{x \in \mathcal{X}}) = \{\theta' \in \Theta | \exists x \in \mathcal{X} : a_{\theta'}^\star(x) \neq a_x\}.$$

Note that $\Theta_{alt} = \Theta(\{a_{\theta^\star}^\star(x)\}_{x \in \mathcal{X}})$. Then,

$$\inf_{\theta' \in \Theta_{alt}} \sum_{x \in \mathcal{X}} \widehat{\rho}_{t-1}(x) \sum_{a \in \mathcal{A}} \omega(x,a) d_{x,a}(\widetilde{\theta}_{t-1}, \theta') \tag{71}$$

$$\leq \max_{\{a_x\}_{x \in \mathcal{X}}} \inf_{\theta' \in \Theta(\{a_x\}_{x \in \mathcal{X}})} \sum_{x \in \mathcal{X}} \widehat{\rho}_{t-1}(x) \sum_{a \in \mathcal{A}} \omega(x,a) d_{x,a}(\widetilde{\theta}_{t-1}, \theta') \tag{72}$$

$$\leq \inf_{\theta' \in \overline{\Theta}_{t-1}} \sum_{x \in \mathcal{X}} \widehat{\rho}_{t-1}(x) \sum_{a \in \mathcal{A}} \omega(x,a) d_{x,a}(\widetilde{\theta}_{t-1}, \theta'). \tag{73}$$

To see the last inequality, note that for all $\{a_x\}_{x \in \mathcal{X}}$ which do not contain only the optimal arms of $\widetilde{\theta}_{t-1}$ (i.e., $\{a_x\}_{x \in \mathcal{X}} \neq \{a_{\widetilde{\theta}_{t-1}}^\star(x)\}_{x \in \mathcal{X}}$), we have $\widetilde{\theta}_{t-1} \in \Theta(\{a_x\}_{x \in \mathcal{X}})$[15], and therefore the infimum is zero. Thus, the maximum must be attained by $\{a_{\widetilde{\theta}_{t-1}}^\star(x)\}_{x \in \mathcal{X}}$, which yields $\Theta(\{a_{\widetilde{\theta}_{t-1}}^\star(x)\}_{x \in \mathcal{X}}) = \overline{\Theta}_{t-1}$. This concludes the proof. $\qquad\square$

**Lemma 12.** *For all time steps $t$,*

$$\sum_{s \leq t : E_s, G_s} \sum_{x \in \mathcal{X}} |\hat{\rho}_{s-1}(x) - \rho(x)| \leq 4|\mathcal{X}| \left( \sqrt{S_t \log(|\mathcal{X}|S_t^2)} + \log S_t + 1 \right). \tag{74}$$

*Proof.* Using the definition of $G_s$,

$$\sum_{s\leq t: E_s, G_s}\sum_{x\in\mathcal{X}}|\hat{\rho}_{s-1}(x)-\rho(x)| \leq |\mathcal{X}|\sum_{s\leq t: E_s, G_s} 2\max\left(\sqrt{\frac{\log(|\mathcal{X}|S_s^2)}{2S_s}}, \frac{2}{s}\right)$$

$$\leq 2|\mathcal{X}|\sum_{s=1}^{S_t}\max\left(\sqrt{\frac{\log(|\mathcal{X}|s^2)}{2s}}, \frac{2}{s}\right)$$

$$\leq 2|\mathcal{X}|\sqrt{\frac{\log(|\mathcal{X}|S_t^2)}{2}}\sum_{s=1}^{S_t}\frac{1}{\sqrt{s}} + 4|\mathcal{X}|\sum_{s=1}^{S_t}\frac{1}{s}$$

$$\leq 4|\mathcal{X}|\left(\sqrt{S_t\log(|\mathcal{X}|S_t^2)} + \log S_t + 1\right),$$

where the last inequality holds since

$$\sum_{t=1}^{m}\sqrt{\frac{1}{t}} \leq 1 + \int_1^m x^{-1/2}dx = 1 + [2x^{1/2}]_1^m = 2\sqrt{m} - 1 < 2\sqrt{m}$$

and $\sum_{t=1}^m \frac{1}{t} \leq \log m + 1$. $\qquad\square$

**Lemma 13.** *Let $t$ be such that both $E_t$ and $G_t$ occur and suppose $\nu \geq 1$. Define*

$$\Psi_t := \sum_{s\leq t: E_s}\sum_{x\in\mathcal{X}}\hat{\rho}_{s-1}(x)\sum_{a\in\mathcal{A}}\omega_s(x,a)\|\phi(x,a)\|_{\bar{V}_{s-1}^{-1}}.$$

*Then,*

$$\Psi_t \leq \frac{4L|\mathcal{X}|}{\sqrt{\nu}}\left(\sqrt{S_t\log(|\mathcal{X}|S_t^2)} + \log S_t + 1\right) + \frac{M_t L}{\sqrt{\nu}} + \frac{L}{\nu}\sqrt{S_t\log S_t}. + \sqrt{2dS_t\log\frac{\nu + S_t L^2/d}{\nu}}.$$

*Proof.* We start by noticing that, for all $x, a$ and $s \geq 0$,

$$\|\phi(x,a)\|_{\bar{V}_{s-1}^{-1}}^2 = \phi(x,a)^T\bar{V}_{s-1}^{-1}\phi(x,a) \leq \sigma_{\max}(\bar{V}_{s-1}^{-1})\underbrace{\|\phi(x,a)\|_2^2}_{\leq L} \leq \frac{L^2}{\sigma_{\min}(\bar{V}_{s-1})} \leq \frac{L^2}{\nu},$$

and thus $\|\phi(x,a)\|_{\bar{V}_{s-1}^{-1}} \leq L/\sqrt{\nu}$. Here $\sigma_{\max}(\cdot)$ and $\sigma_{\min}(\cdot)$ denote the maximum and minimum eigenvalue of a matrix, respectively. Splitting the steps where the good event does and does not hold,

$$\Psi_t = \sum_{s\leq t: E_s, G_s}\sum_{x\in\mathcal{X}}\hat{\rho}_{s-1}(x)\sum_{a\in\mathcal{A}}\omega_s(x,a)\|\phi(x,a)\|_{\bar{V}_{s-1}^{-1}} + \sum_{s\leq t: E_s, \neg G_s}\sum_{x\in\mathcal{X}}\hat{\rho}_{s-1}(x)\sum_{a\in\mathcal{A}}\omega_s(x,a)\|\phi(x,a)\|_{\bar{V}_{s-1}^{-1}}$$

$$\leq \sum_{s\leq t: E_s, G_s}\sum_{x\in\mathcal{X}}(\hat{\rho}_{s-1}(x)-\rho(x))\sum_{a\in\mathcal{A}}\omega_s(x,a)\|\phi(x,a)\|_{\bar{V}_{s-1}^{-1}} + \frac{M_t L}{\sqrt{\nu}}$$

$$+ \sum_{s\leq t: E_s, G_s}\sum_{x\in\mathcal{X}}\rho(x)\sum_{a\in\mathcal{A}}\omega_s(x,a)\|\phi(x,a)\|_{\bar{V}_{s-1}^{-1}}$$

$$\leq \frac{L}{\sqrt{\nu}}\sum_{s\leq t: E_s, G_s}\sum_{x\in\mathcal{X}}|\hat{\rho}_{s-1}(x)-\rho(x)| + \frac{M_t L}{\sqrt{\nu}} + \sum_{s\leq t: E_s, G_s}\sum_{x\in\mathcal{X}}\rho(x)\sum_{a\in\mathcal{A}}\omega_s(x,a)\|\phi(x,a)\|_{\bar{V}_{s-1}^{-1}}$$

$$\leq \frac{4L|\mathcal{X}|}{\sqrt{\nu}}\left(\sqrt{S_t\log(|\mathcal{X}|S_t^2)} + \log S_t + 1\right) + \frac{M_t L}{\sqrt{\nu}} + \sum_{s\leq t: E_s, G_s}\sum_{x\in\mathcal{X}}\rho(x)\sum_{a\in\mathcal{A}}\omega_s(x,a)\|\phi(x,a)\|_{\bar{V}_{s-1}^{-1}},$$

where in the first and second inequality we bounded the expected feature-norms by their maximum value and added/subtracted the first term with the true context distribution. In the last step we applied Lemma 12. We now focus exclusively on the third term. Using the fact that the good event holds at time $t$,

$$\sum_{s\leq t: E_s, G_s}\sum_{x\in\mathcal{X}}\rho(x)\sum_{a\in\mathcal{A}}\omega_s(x,a)\|\phi(x,a)\|_{\bar{V}_{s-1}^{-1}} \leq \sum_{s\leq t: E_s}\sum_{x\in\mathcal{X}}\rho(x)\sum_{a\in\mathcal{A}}\omega_s(x,a)\|\phi(x,a)\|_{\bar{V}_{s-1}^{-1}}$$

$$\leq \sum_{s\leq t: E_s}\|\phi(X_s, A_s)\|_{\bar{V}_{s-1}^{-1}} + \frac{L}{\nu}\sqrt{S_t\log S_t}.$$

Finally, let $\bar{V}_{e,t} := \sum_{s \leq t:E_s} \phi(X_s, A_s)\phi(X_s, A_s)^T + \nu I$ denote the regularized design matrix computed using only the exploration rounds. Then, we have $\bar{V}_t \succeq \bar{V}_{e,t}$ (since sum of rank-one matrices), which implies $\bar{V}_t^{-1} \preceq \bar{V}_{e,t}^{-1}$ and thus $\|\phi(x, a)\|_{\bar{V}_{s-1}^{-1}} \leq \|\phi(x, a)\|_{\bar{V}_{e,s-1}^{-1}}$. Here $\succeq$ denotes the Loewner ordering, i.e., for two symmetric matrices $A, B$ we have $A \succeq B$ ($A \succ B$) if $A - B$ is positive semi-definite (positive definite). Therefore,

$$\sum_{s \leq t:E_s} \|\phi(X_s, A_s)\|_{\bar{V}_{s-1}^{-1}} \leq \sum_{s \leq t:E_s} \|\phi(X_s, A_s)\|_{\bar{V}_{e,s-1}^{-1}} \overset{(a)}{\leq} \sqrt{S_t \sum_{s \leq t:E_s} \|\phi(X_s, A_s)\|_{\bar{V}_{e,s-1}^{-1}}^2}$$

$$\overset{(b)}{\leq} \sqrt{2S_t \log \frac{\det(\bar{V}_{e,t})}{\nu^d}} \overset{(d)}{\leq} \sqrt{2dS_t \log \frac{\nu + S_t L^2/d}{\nu}},$$

where in (a) we equivalently rewritten the first term as a sum over exploration rounds, (b) is from Cauchy-Schwartz inequality, in (c) we used Lemma 11 of [4], and in (d) we used the determinant-trace inequality (Lemma 10 of [4]) to bound the determinant of $\bar{V}_{e,t}$ by $(\nu + S_t L^2/d)^d$. The final statement follows by combining the previous bounds. $\qquad\square$

### I.3 Online Convex Optimization

Here we recall some basic results from online convex optimization. See [e.g., 29] for detailed proofs and discussion of these results.

**Lemma 14** (Recursion bound for subgradient descent). *Let $\sup_{t \geq 1:E_t} |g_t(\omega_t, z_k)|^2 \leq b_\lambda$. For any phase $k \geq 0$, $t \in \mathcal{T}_k^E$, and $\lambda \in \mathbb{R}_+$, the incremental updates to the Lagrange multiplier $\{\lambda_t\}_{t \in \mathcal{T}_k^E}$ of Algorithm 1 satisfy*

$$\sum_{s \leq t:s \in \mathcal{T}_k^E} g_s(\omega_s, z_k)(\lambda_s - \lambda) \leq \frac{1}{2\alpha_k^\lambda}(\lambda - \lambda_1)^2 + \frac{\alpha_k^\lambda b_\lambda^2}{2} S_{t,k}.$$

*Proof.* Recall that the optimization process is reset at the beginning of each phase. Let $\tau_{s,k}$ be a random variable indicating the time at which the $s$-th exploration round of phase $k$ occurs. Note that $\lambda_{\tau_{1,k}} = \lambda_1$. In order to simplify the exposition, and with some abuse of notation, let $\lambda_s = \lambda_{\tau_{s,k}}$ and $g_s = g_{\tau_{s,k}}(\omega_{\tau_{s,k}}, z_k)$. By definition of the update rule, for each $s \geq 1$,

$$(\lambda_{s+1} - \lambda)^2 = (\min\{[\lambda_s - \alpha_k^\lambda g_s]_+, \lambda_{\max}\} - \lambda)^2 = \min\{[\lambda_s - \alpha_k^\lambda g_s]_+ - \lambda, \lambda_{\max} - \lambda\}^2$$

$$\leq (\lambda_s - \alpha_k^\lambda g_s - \lambda)^2 = (\lambda_s - \lambda)^2 + (\alpha_k^\lambda g_s)^2 + 2\alpha_k^\lambda(\lambda - \lambda_s)g_s.$$

Dividing by $2\alpha_k^\lambda$ and rearranging,

$$(\lambda_s - \lambda)g_s \leq \frac{(\lambda_s - \lambda)^2 - (\lambda_{s+1} - \lambda)^2}{2\alpha_k^\lambda} + \frac{\alpha_k^\lambda}{2}g_s^2.$$

Summing over all $s$ up to $S_t$ and noting that the first sum on the right-hand side is telescopic,

$$\sum_{s=1}^{S_t}(\lambda_s - \lambda)g_s \leq \frac{1}{2\alpha_k^\lambda}(\lambda_1 - \lambda)^2 - \frac{1}{2\alpha_k^\lambda}(\lambda_{S_t+1} - \lambda)^2 + \frac{\alpha_k^\lambda}{2}\sum_{s=1}^{S_t}g_s^2.$$

The proof is concluded by upper-bounding the second term by zero and mapping the exploration counter $s$ back to time steps. $\qquad\square$

**Lemma 15.** *[Recursion bound for Online Mirror Descent (OMD)] Let $\omega_1$ be the uniform distribution over actions for each context and $\sup_{t \geq 1:E_t} \|q_t\|_\infty \leq b_\omega$. For any phase $k \geq 0$, $t \in \mathcal{T}_k^E$, and $\omega \in \Omega$, the OMD updates of Algorithm 1 satisfy*

$$\sum_{s \leq t:s \in \mathcal{T}_k^E} h_s(\omega_s, \lambda_s, z_k) - \sum_{s \leq t:s \in \mathcal{T}_k^E} h_s(\omega, \lambda_s, z_k) \geq -\frac{\log |\mathcal{A}|}{\alpha_k^\omega} - \frac{\alpha_k^\omega b_\omega^2}{2} S_{t,k}.$$

*Proof.* We can follow the same steps as before, mapping time steps to exploration counters and then applying the standard recursion bound for OMD [e.g., 29]. $\qquad\square$

**Corollary 2.** *[Recursion bound for primal-dual algorithm] For any phase $k \geq 0$, $t \in \mathcal{T}_k^E$, $\omega \in \Omega$, and $\lambda \in \mathbb{R}_+$, under the same conditions as in Lemma 15 and 14,*

$$\sum_{s \leq t: s \in \mathcal{T}_k^E} f_s(\omega_s) \geq \sum_{s \leq t: s \in \mathcal{T}_k^E} h_s(\omega, \lambda_s, z_k) - \lambda \sum_{s \leq t: s \in \mathcal{T}_k^E} g_s(\omega_s, z_k) - \frac{\log |\mathcal{A}|}{\alpha_k^\omega} - \frac{\alpha_k^\omega b_\omega^2}{2} S_{t,k}$$

$$- \frac{1}{2\alpha_k^\lambda}(\lambda - \lambda_1)^2 - \frac{\alpha_k^\lambda b_\lambda^2}{2} S_{t,k}.$$

*Proof.* The proof is straightforward by expanding $\sum_{s \leq t: s \in \mathcal{T}_k^E} h_s(\omega_s, \lambda_s, z_k) = \sum_{s \leq t: s \in \mathcal{T}_k^E}(f_s(\omega_s) + \lambda_s g_s(\omega_s, z_k))$ and combining Lemma 15 with Lemma 14. $\qquad\square$

## J  Confidence Set for Regularized Least-Squares (Proof of Thm. 1)

The following theorem is the extended version of Thm. 1. It provides a refined confidence set for the parameters estimated by regularized least-squares.

**Theorem 4** (Confidence set over parameters). *Let $\delta \in (0,1)$ and $n \geq 3$. Then,*

$$\mathbb{P}\left\{\exists t \in [n] : \|\widehat{\theta}_t - \theta^\star\|_{\overline{V}_t} \geq \sqrt{c_{n,\delta}}\right\} \leq \delta,$$

*where $\sqrt{c_{n,\delta}} := \frac{\gamma_n}{1 - \frac{1}{\log n}}\sqrt{\kappa_{n,\delta}}$, $\gamma_n := 1 + \frac{1}{\log n}$, and*

$$\sqrt{\kappa_{n,\delta}} = B\sqrt{\nu} + \sqrt{\frac{2\sigma^2 \log\left(\frac{2 + \frac{2nL^2}{d\nu}}{\delta}\right)}{(\log n)^2}} + \sqrt{2\sigma^2 \gamma_n^3 \log\left(\frac{2(1 + \log(n/\chi_n)\log(n))}{\delta}\right) + 2\gamma_n^3 \Upsilon_n}.$$

*Finally, we set $\Upsilon_n := d \log\left(\frac{5}{2} + 2\log n\sqrt{d}\right) + d \log\left(2 + 4d \log\left(4\gamma_n d(\log n)^2 \sqrt{\frac{\nu + L^2 n}{d\nu}}\right)\log n\right)$ and $\chi_n := \frac{\nu^2 v_{\min}^2}{16dL^2(\nu + L^2 n)(\log n)^4 \gamma_n^4}$.*

**Asymptotic dependence**   It is important to note that $\lim_{n \to \infty} \frac{c_{n,1/n}}{2\sigma^2 \log n} = 1$.

### J.1  Proof of Thm. 4

The proof can be summarized in three main steps:

1. We reduce the problem of bounding $\|\widehat{\theta}_t - \theta^\star\|_{\overline{V}_t}$ to one in which we need to bound $(\widehat{\theta}_t - \theta^\star)^T \overline{V}_t^{1/2} v$ for any $v \in \mathcal{C}_1$, where $\mathcal{C}_1 \subset \mathbb{R}^d$ is a (finite) $\epsilon_1$-cover of the $d$-dimensional Euclidean unit ball. We build this cover in such a way that all its elements have norm bounded from below by a strictly positive constant and from above.

2. We extend Theorem 8 of [7] to bound $(\widehat{\theta}_t - \theta^\star)^T \overline{V}_t^{1/2} v$ uniformly over all $v \in \mathcal{C}_1$, instead of the prediction errors $(\widehat{\theta}_t - \theta^\star)^T \phi(x, a)$ uniformly over all contexts/arms. This requires a second $\epsilon_2$-cover (we shall call it $\mathcal{C}_2$) of the set $\{\overline{V}_t^{-1/2} v : t \in [n], v \in \mathcal{C}_1\}$. The result is reported in Lemma 16.

3. The resulting bound is of order $\mathcal{O}(\log(1/\delta) + d \log(1/\epsilon_1))$, which requires tuning $\epsilon_1 = \frac{1}{\log n}$ to cancel the bias of the first cover asymptotically without compromising the size of the cover itself.

**Step 1.**   We start from the fact that

$$\|\widehat{\theta}_t - \theta^\star\|_{\overline{V}_t} = \frac{(\widehat{\theta}_t - \theta^\star)^T \overline{V}_t(\widehat{\theta}_t - \theta^\star)}{\|\widehat{\theta}_t - \theta^\star\|_{\overline{V}_t}} = (\widehat{\theta}_t - \theta^\star)^T \overline{V}_t^{1/2} z_t, \tag{75}$$

where $z_t = \frac{\overline{V}_t^{1/2}(\widehat{\theta}_t - \theta^\star)}{\|\widehat{\theta}_t - \theta^\star\|_{\overline{V}_t}}$ is such that $\|z_t\|_2 = 1$. To handle the fact that $z_t$ is random, we build a linear $(\epsilon_1 > 0)$-cover of the space $\mathcal{Z} = \{z \in \mathbb{R}^d : \|z\|_2 \le 1\}$, which includes $z_t$ for all $t = 1, \ldots, n$. Let $\epsilon_1' > 0$, $\{e_1, e_2, \ldots, e_d\}$ be the canonical basis of $\mathbb{R}^d$, and define

$$\widetilde{\mathcal{C}}_1 := \left\{ \sum_{i=1}^d a_i e_i : a_i \in \left\{ \pm \epsilon_1' \left( \frac{1}{2} + j \right) : j = 0, 1, \ldots, \bar{j} \right\} \forall i \in [d] \right\},$$

where $\bar{j} := \left\lceil \frac{1}{\epsilon_1'} - \frac{1}{2} \right\rceil$. For any vector $z \in \mathcal{Z}$, we can find a vector in $\widetilde{\mathcal{C}}_1$ with at most $\epsilon_1'$ error on each component of $z$, which leads to $\min_{v \in \widetilde{\mathcal{C}}_1} \|v - z\|_2 \le \epsilon_1' \sqrt{d}$ [see e.g., 30, Chap. 27]. Setting $\epsilon_1' = \epsilon_1/\sqrt{d}$ gives an $\epsilon_1$-cover of the unit ball in $\ell_2$-norm. The only problem with this cover is that it contains vectors with norm bigger than 1 and scaling with $d$,[16] which may lead to an undesirable dependency later on. However, we can safely remove the vectors with large norm without affecting the desired accuracy of the cover. Without loss of generality, select $z \in \mathcal{Z}$ in the positive orthant (i.e., $z_i \ge 0$, for any $i \in [d]$) such that we make an error of $\epsilon_1'$ on each component (i.e., the worst-case) and let $w = z + \epsilon_1'$. Then

$$\|w\|_2^2 = \sum_{i=1}^d (z_i + \epsilon_1')^2 = \underbrace{\|z\|_2^2}_{\le 1} + d(\epsilon_1')^2 + 2\epsilon_1' \underbrace{\sum_{i=1}^d z_i}_{\le \|z\|_1 \le \sqrt{d}} \le 1 + \epsilon_1^2 + 2\epsilon_1 = (1 + \epsilon_1)^2.$$

Hence vectors with norm at most $(1 + \epsilon_1)$ actually suffice and thus we can set $\mathcal{C}_1 = \widetilde{\mathcal{C}}_1 \setminus \{v \in \widetilde{\mathcal{C}}_1 : \|v\|_2 > (1 + \epsilon_1)\}$. Then we upper bound the size of this cover as

$$|\mathcal{C}_1| \le |\widetilde{\mathcal{C}}_1| = 2^d (1 + \bar{j})^d \le \left( \frac{5}{2} + \frac{2}{\epsilon_1'} \right)^d = \left( \frac{5}{2} + \frac{2\sqrt{d}}{\epsilon_1} \right)^d.$$

To recap, our cover $\mathcal{C}_1$ has the following properties:

1. $\forall z \in \mathcal{Z} = \{z \in \mathbb{R}^d : \|z\|_2 \le 1\}, \exists v \in \mathcal{C}_1 : \|z - v\|_2 \le \epsilon_1$

2. $|\mathcal{C}_1| \le \left( \frac{5}{2} + \frac{2\sqrt{d}}{\epsilon_1} \right)^d$

3. $\forall v \in \mathcal{C}_1 : \|v\|_2 \le v_{\max} := 1 + \epsilon_1$

4. $\forall v \in \mathcal{C}_1, i \in [d] : |v_i| \ge v_{\min} := \frac{\epsilon_1}{2\sqrt{d}}$ (this follows from the discretization used in $\widetilde{\mathcal{C}}_1$ and it implies that $\|v\|_2 \ge v_{\min} \sqrt{d} = \frac{\epsilon_1}{2}$)

**Step 2.** We use an extension of Thm. 8 of [7] to bound the prediction error at vectors in the cover $\mathcal{C}_1$ after applying the linear transformation $\overline{V}_t^{1/2}$.

**Lemma 16.** *Let $\mathcal{C} \subset \mathbb{R}^d$ be a finite set such that, for any $v \in \mathcal{C}$, $\|v\|_2 \le v_{\max} < \infty$ and $|v_i| \ge v_{\min} > 0, \forall i \in [d]$. Suppose that $n \ge 2$. Then, for any $\delta \in (0, 1)$,*

$$\mathbb{P}\left\{ \exists t \le n, v \in \mathcal{C} : (\widehat{\theta}_t - \theta^\star)^T \overline{V}_t^{1/2} v \ge \sqrt{\kappa_{n,\delta}} \|v\|_2 \right\} \le \delta,$$

*where*

$$\sqrt{\kappa_{n,\delta}} = B\sqrt{\nu} + \sqrt{\frac{2\sigma^2 \log\left( \frac{2 + \frac{2nL^2}{d\nu}}{\delta} \right)}{(\log n)^2}} + \sqrt{2\sigma^2 \gamma_n^3 \log\left( \frac{2(1 + \log(n/\chi_n)\log(n))}{\delta} \right) + 2\gamma_n^3 \Upsilon_n}$$

*and* $\Upsilon_n = \log(|\mathcal{C}|) + d\log\left( 2 + 4d\log\left( 2d\log n \frac{v_{\max}}{v_{\min}} \sqrt{\frac{\nu + L^2 n}{d\nu}} \right) \log n \right)$ *and* $\chi_n = \frac{\nu^2 v_{\min}^2}{4L^2(\nu + L^2 n)(\log n)^2 v_{\max}^2 \gamma_n^2}$.

The specific shape of the bound is obtained by exploiting the properties of the cover $\mathcal{C}_1$ derived in the first step, where $v_{\max} = 1 + \epsilon_1$ and $v_{\min} = \frac{\epsilon_1}{2\sqrt{d}}$.

**Step 3.** We finally tune $\epsilon_1$ to obtain the final bound. With probability at least $1 - \delta$, we have that

$$\|\widehat{\theta}_t - \theta^\star\|_{\overline{V}_t} \overset{(a)}{=} (\widehat{\theta}_t - \theta^\star)^T \overline{V}_t^{1/2} z_t \overset{(b)}{\leq} \max_{z \in \mathcal{Z}} (\widehat{\theta}_t - \theta^\star)^T \overline{V}_t^{1/2} z$$

$$= \max_{z \in \mathcal{Z}} \min_{v \in \mathcal{C}_1} \left\{ (\widehat{\theta}_t - \theta^\star)^T \overline{V}_t^{1/2}(z - v) + (\widehat{\theta}_t - \theta^\star)^T \overline{V}_t^{1/2} v \right\}$$

$$\overset{(c)}{\leq} \max_{z \in \mathcal{Z}} \min_{v \in \mathcal{C}_1} \left\{ \|\widehat{\theta}_t - \theta^\star\|_{\overline{V}_t} \|z - v\|_2 + \sqrt{\kappa_{n,\delta}} \|v\|_2 \right\}$$

$$\overset{(d)}{\leq} \epsilon_1 \|\widehat{\theta}_t - \theta^\star\|_{\overline{V}_t} + (1 + \epsilon_1)\sqrt{\kappa_{n,\delta}},$$

where $(a)$ follows from Eq. 75, (b) from the fact that $z_t \in \mathcal{Z}$, $(c)$ holds with probability at least $1 - \delta$ by Lem. 16 and $(d)$ by properties 1 and 3 of the cover $\mathcal{C}_1$. The statement of the theorem follows by setting $\epsilon_1 = \frac{1}{\log n}$ and rearranging.

## J.2 Proof of Lem. 16

The proof follows similar steps as in [7, Thm. 8].

*Proof.* Take any $v \in \mathcal{C}_1$ and $t \in [n]$. Then,

$$(\widehat{\theta}_t - \theta^\star)^T \overline{V}_t^{1/2} v \overset{(a)}{=} \left( \overline{V}_t^{-1} \sum_{s=1}^t \phi(X_s, A_s) Y_s - \theta^\star \right)^T \overline{V}_t^{1/2} v$$

$$\overset{(b)}{=} \left( \overline{V}_t^{-1} \sum_{s=1}^t \phi(X_s, A_s)(\phi(X_s, A_s)^T \theta^\star + \xi_s) - \theta^\star \right)^T \overline{V}_t^{1/2} v$$

$$\overset{(c)}{=} \left( \overline{V}_t^{-1} V_t \theta^\star + \overline{V}_t^{-1} \sum_{s=1}^t \phi(X_s, A_s)\xi_s - \theta^\star \right)^T \overline{V}_t^{1/2} v$$

$$\overset{(d)}{=} \underbrace{\left( \overline{V}_t^{-1} V_t \theta^\star - \theta^\star \right)^T \overline{V}_t^{1/2} v}_{(i)} + \underbrace{\sum_{s=1}^t v^T \overline{V}_t^{-1/2} \phi(X_s, A_s)\xi_s}_{(ii)}, \qquad (76)$$

where (a) is from the definition of $\widehat{\theta}_t$, (b) since $Y_s = \phi(X_s, A_s)^T \theta^\star + \xi_s$ with $\xi_s \sim \mathcal{N}(0, \sigma^2)$, (c) from the definition of $V_t$, and (d) after rearranging. Let us bound (i). Since $\theta^\star = \overline{V}_t^{-1} \overline{V}_t \theta^\star$, we have

$$(i) = v^T \overline{V}_t^{-1/2}(V_t - \overline{V}_t)\theta^\star = -\nu v^T \overline{V}_t^{-1/2} \theta^\star,$$

where we used $\overline{V}_t = \nu I + V_t$. Therefore,

$$|(i)| \leq \nu |v^T \overline{V}_t^{-1/2} \theta^\star| \leq \nu \|v\|_2 \|\overline{V}_t^{-1/2} \theta^\star\|_2 = \nu \|v\|_2 \|\theta^\star\|_{\overline{V}_t^{-1}},$$

where the second inequality is by Cauchy-Schwartz inequality. Since $\overline{V}_t \succeq \nu I$, $\|\theta^\star\|_{\overline{V}_t^{-1}} \leq \frac{1}{\sqrt{\nu}} \|\theta^\star\|_2 \leq \frac{B}{\sqrt{\nu}}$. This yields

$$|(i)| \leq B\sqrt{\nu} \|v\|_2.$$

Let us consider the second term. Since $\overline{V}_t^{-1/2}$ is random, we proceed using the same covering argument as in the proof in [7, Thm. 8]. Let $\epsilon_2 > 0$ (whose value will be specified later). Recall that our input is a finite set of $d$-dimensional vectors $\mathcal{C}_1$ such that $\|v\|_2 \leq v_{\max} < \infty$ and $|v_i| \geq v_{\min} > 0$ hold for all $v \in \mathcal{C}_1$ and $i \in [d]$. Note that the latter condition implies $\|v\|_2 \geq v_{\min}\sqrt{d}$. Our goal is to build an $\epsilon_2$-covering set of $\{\overline{V}_t^{-1/2} v : t \in [n], v \in \mathcal{C}_1\}$. Since this set is random, we build a deterministic one that contains the former almost surely and cover it instead. Note that, for

any $t \in [n]$, $\overline{V}_t^{-1/2}$ is such that (1) $\overline{V}_t^{-1/2} \succ 0$, (2) $\|\overline{V}_t^{-1/2}\|_2 = \sigma_{\max}(\overline{V}_t^{-1/2}) \leq \frac{1}{\sqrt{\nu}}$, and (3) $\sigma_{\min}(\overline{V}_t^{-1/2}) \geq \frac{1}{\sqrt{\nu+L^2n}}$. Let $\mathcal{D}$ denote the set of $d \times d$ matrices with these properties, that is,

$$\mathcal{D} := \left\{ D \in \mathbb{R}^{d \times d} : D \succ 0, \|D\|_2 \leq \frac{1}{\sqrt{\nu}}, \sigma_{\min}(D) \geq \frac{1}{\sqrt{\nu+L^2n}} \right\}.$$

Then, $\overline{V}_t^{-1/2} \in \mathcal{D}$ for all $t \in [n]$ and our initial set to be covered is almost surely contained into $\mathcal{B} := \{Dv : D \in \mathcal{D}, v \in \mathcal{C}_1\}$. Furthermore, $v_{\min}\sqrt{\frac{d}{\nu+L^2n}} \leq \|b\|_2 \leq \frac{v_{\max}}{\sqrt{\nu}}$ for all $b \in \mathcal{B}$. We shall now cover $\mathcal{B}$. Let $\{e_1, \ldots, e_d\}$ be the canonical basis of $\mathbb{R}^d$ and, for all $v \in \mathcal{C}_1$ we introduce a cover with *geometric scale* as

$$\tilde{\mathcal{C}}_{2,v} := \left\{ \sum_{i=1}^d a_i e_i \middle| \forall i \in [d] : a_i \in \left\{ \pm \frac{\epsilon_2 \|v\|_2 (1+\epsilon_2)^j}{\sqrt{\nu+L^2n}} : j = 0, 1, \ldots, \bar{j} \right\} \right\},$$

where $\bar{j} := \left\lceil \frac{\log\left(\frac{v_{\max}}{\epsilon_2 v_{\min}}\sqrt{\frac{\nu+L^2n}{d\nu}}\right)}{\log(1+\epsilon_2)} \right\rceil$ is such that $\frac{\epsilon_2 \|v\|_2 (1+\epsilon_2)^{\bar{j}}}{\sqrt{\nu+L^2n}} \geq \frac{v_{\max}}{\sqrt{\nu}}$ (i.e., the maximum absolute

value of each element in $\mathcal{B}$). Then, our cover is $\tilde{\mathcal{C}}_2 = \bigcup_{v \in \mathcal{C}_1} \tilde{\mathcal{C}}_{2,v}$. Let us analyze some its properties. First its size is

$$|\tilde{\mathcal{C}}_2| \leq |\mathcal{C}_1| \left( 2 + \frac{\log\left(\frac{v_{\max}}{\epsilon_2 v_{\min}}\sqrt{\frac{\nu+L^2n}{d\nu}}\right)}{\log(1+\epsilon_2)} \right)^d. \tag{77}$$

Then, we can show the following covering property in $l_\infty$-norm.

**Proposition 5.** *For all $v \in \mathcal{C}_1$, $t \in [n]$, there exists $\overline{w}_{v,t} \in \tilde{\mathcal{C}}_2$ such that*

$$\forall i \in [d] : \left| \left[ \overline{V}_t^{-1/2} v - \overline{w}_{v,t} \right]_i \right| \leq \epsilon_2 \max \left\{ \left| \left[ \overline{V}_t^{-1/2} v \right]_i \right|, \frac{\|v\|_2}{\sqrt{\nu+L^2n}} \right\}.$$

*Proof.* For simplicity, denote $b := \overline{V}_t^{-1/2} v$. By definition, we have $b \in \mathcal{B}$ (i.e., the deterministic set that we actually covered). We shall build a vector $w \in \mathcal{C}_2$ which has the desired property. Take any component $b_i$, with $i \in [d]$, then
(1) If $|b_i| < \frac{\epsilon_2 \|v\|_2}{\sqrt{\nu+L^2n}}$, then we can set $w_i = \frac{\epsilon_2 \|v\|_2}{\sqrt{\nu+L^2n}}\text{sign}(b_i)$ and we have

$$|w_i - b_i| \leq |w_i| = \frac{\epsilon_2 \|v\|_2}{\sqrt{\nu+L^2n}}.$$

(2) If $|b_i| \geq \frac{\epsilon_2 \|v\|_2}{\sqrt{\nu+L^2n}}$, by the geometrical cover, we can find a point $w_i$ such that $1 \leq \frac{|w_i|}{|b_i|} \leq 1 + \epsilon_2$. Too see this, without loss of generality, that $b_i$ is positive. Note that, since $b_i$ lies in the range $[\frac{\epsilon_2 \|v\|_2}{\sqrt{\nu+L^2n}}, \frac{v_{\max}}{\sqrt{\nu}}]$ which is covered geometrically, there exists a real value $0 \leq k \leq \bar{j}$ such that $b_i = \frac{\epsilon_2 \|v\|_2}{\sqrt{\nu+L^2n}}(1+\epsilon_2)^k$. Then, if we set $w_i = \frac{\epsilon_2 \|v\|_2}{\sqrt{\nu+L^2n}}(1+\epsilon_2)^{\lceil k \rceil}$, we can easily verify the desired property. This implies

$$|w_i - b_i| \leq |w_i| - |b_i| \leq \epsilon_2 |b_i|,$$

where the left-hand side is from the reverse triangle inequality. The statement follows by combining the two cases. $\square$

An immediate consequence of Proposition 5 is that, for all $v \in \mathcal{C}_1$, $t \in [n]$, there exists $\overline{w}_{v,t} \in \tilde{\mathcal{C}}_2$ which can be written as $\overline{w}_{v,t} = \overline{V}_t^{-1/2} v + \zeta$, where $\zeta \in \mathbb{R}^d$ is a vector of errors such that $|\zeta_i| \leq \epsilon_2 \max \left\{ \left| \left[ \overline{V}_t^{-1/2} v \right]_i \right|, \frac{\|v\|_2}{\sqrt{\nu+L^2n}} \right\}$ for all $i \in [d]$.

Note that, by definition, $\tilde{\mathcal{C}}_2$ contains vectors with norm that scales in $\sqrt{d}$ (e.g., the vector with all components larger or equal to $v_{\max}/\sqrt{\nu}$, which has norm $v_{\max}\sqrt{d/\nu}$ belongs to $\tilde{\mathcal{C}}_2$). These vectors will create an undesirable dependency on $d$ later on, and so we need to perform some pruning before proceeding. Take any $b \in \mathcal{B}$ and suppose that $b = Dv$ for $v \in \mathcal{C}_1$ and $D \in \mathcal{D}$. Let $\mathcal{I} := \{i \in [d] : |b_i| < \frac{\epsilon_2 \|v\|_2}{\sqrt{\nu+L^2 n}}\}$ be the set of components $i$ such that $|b_i|$ is below the starting point of our geometrical grid $\tilde{C}_{2,v}$ and $\mathcal{I}^c = [d] \setminus \mathcal{I}$. From the proof of Proposition 5, we know that the vector $w \in \tilde{\mathcal{C}}_2$ that is the closest to $b$ is such that $|w_i| \leq \frac{\epsilon_2 \|v\|_2}{\sqrt{\nu+L^2 n}}$ for $i \in \mathcal{I}$ and $|w_i|/|b_i| \leq 1 + \epsilon_2$ for $i \in \mathcal{I}^c$. Therefore,

$$\|w\|_2^2 = \sum_{i \in \mathcal{I}} |w_i|^2 + \sum_{i \in \mathcal{I}^c} |w_i|^2 \leq |\mathcal{I}| \frac{\epsilon_2^2 \|v\|_2^2}{\nu + L^2 n} + (1 + \epsilon_2)^2 \sum_{i \in \mathcal{I}^c} |b_i|^2 \leq \frac{d\epsilon_2^2 \|v\|_2^2}{\nu + L^2 n} + (1 + \epsilon_2)^2 \|b\|_2^2.$$

This implies that $\|w\|_2 \leq \frac{\sqrt{d}\epsilon_2 \|v\|_2}{\sqrt{\nu+L^2 n}} + (1 + \epsilon_2)\|b\|_2$. Recall that $\|b\|_2 \leq \frac{v_{\max}}{\sqrt{\nu}}$ and $\|v\|_2 \leq v_{\max}$. Thus, $\|w\|_2 \leq \frac{\sqrt{d}\epsilon_2 v_{\max}}{\sqrt{\nu+L^2 n}} + (1 + \epsilon_2)\frac{v_{\max}}{\sqrt{\nu}} \leq \frac{v_{\max}}{\sqrt{\nu}}\left(1 + \epsilon_2(1 + \sqrt{d})\right)$. This condition holds for all "useful" vectors in our cover (i.e., those that are the closest to some of the vectors we need to cover). Therefore, we can safely set $\mathcal{C}_2 = \left\{w \in \tilde{\mathcal{C}}_2 : \|w\|_2 \leq \frac{v_{\max}}{\sqrt{\nu}}\left(1 + \epsilon_2(1 + \sqrt{d})\right)\right\}$ as our final cover. Note that Proposition 5 still holds for $\mathcal{C}_2$ since we removed only vectors that cannot be the closest to any of the points to be covered. In the following, we set $w_{\max} := \frac{v_{\max}}{\sqrt{\nu}}\left(1 + \epsilon_2(1 + \sqrt{d})\right)$ as the maximum norm of any vector in $\mathcal{C}_2$.

Let us now go back to bounding term (ii) in Eq. 76. Let $\overline{w}_{v,t} := \operatorname{argmin}_{w \in \mathcal{C}_2} \left\|\overline{V}_t^{-1/2} v - w\right\|_1$ be the vector in our cover $\mathcal{C}_2$ which is the closest to $\overline{V}_t^{-1/2} v$ uniformly over all components. Then,

$$
\begin{aligned}
(ii) := \sum_{s=1}^t v^T \overline{V}_t^{-1/2} \phi(X_s, A_s)\xi_s &= \left(\overline{V}_t^{-1/2} v\right)^T \sum_{s=1}^t \phi(X_s, A_s)\xi_s \\
&= \left(\overline{V}_t^{-1/2} v - \overline{w}_{v,t}\right)^T W_t + \overline{w}_{v,t}^T W_t \leq \underbrace{\left\|\overline{V}_t^{-1/2} v - \overline{w}_{v,t}\right\|_{\overline{V}_t}}_{(a)} \underbrace{\|W_t\|_{\overline{V}_t^{-1}}}_{(b)} + \underbrace{\overline{w}_{v,t}^T W_t}_{(c)},
\end{aligned}
$$

where we defined $W_t := \sum_{s=1}^t \phi(X_s, A_s)\xi_s$. We start from (a). Using the error-decomposition property from Proposition 5, we can write $\left\|\overline{V}_t^{-1/2} v - \overline{w}_{v,t}\right\|_{\overline{V}_t} = \|\zeta\|_{\overline{V}_t}$ for some vector $\zeta \in \mathbb{R}^d$ with $|\zeta_i| \leq \epsilon_2 \max\left\{|[\overline{V}_t^{-1/2} v]_i|, \frac{\|v\|_2}{\sqrt{\nu+L^2 n}}\right\}$ for all $i \in [d]$. Since this implies $|\zeta_i| \leq \epsilon_2\left(|[\overline{V}_t^{-1/2} v]_i| + \frac{\|v\|_2}{\sqrt{\nu+L^2 n}}\right)$, we have

$$\|\zeta\|_{\overline{V}_t} \overset{(d)}{\leq} \epsilon_2 \left\|\overline{V}_t^{-1/2} v\right\|_{\overline{V}_t} + \frac{\epsilon_2 \|v\|_2}{\sqrt{\nu + L^2 n}} \|\mathbf{1}_d\|_{\overline{V}_t} \overset{(e)}{\leq} \epsilon_2 \|v\|_2 + \frac{\epsilon_2 \|v\|_2 \sqrt{d}}{\sqrt{\nu + L^2 n}} \left\|\overline{V}_t^{1/2}\right\|_2 \overset{(f)}{\leq} \epsilon_2 \|v\|_2 + \epsilon_2 \|v\|_2 \sqrt{d},$$

where in (d) we used the triangle inequality ($\mathbf{1}_d$ denotes the d-dimensional vector of ones), in (e) we used $\|\mathbf{1}_d\|_{\overline{V}_t} \leq \left\|\overline{V}_t^{1/2}\right\|_2 \|\mathbf{1}_d\|_2$, and in (f) we upper bounded the maximum eigenvalue of $\left\|\overline{V}_t^{1/2}\right\|_2$ by $\sqrt{\nu + L^2 n}$. Therefore, we conclude,

$$(a) := \left\|\overline{V}_t^{-1/2} v - \overline{w}_{v,t}\right\|_{\overline{V}_t} \leq \epsilon_2(1 + \sqrt{d})\|v\|_2.$$

Term (b) can be bounded by Lemma 17. For any $\delta' \in (0, 1)$, with probability at least $1 - \delta'$,

$$(b) := \|W_t\|_{\overline{V}_t^{-1}} \leq \sqrt{2\sigma^2 d \log\left(\frac{1 + \frac{tL^2}{d\nu}}{\delta'}\right)}.$$

Term (c) can be bounded by Lemma 20 (whose bound holds uniformly over all elements in $\mathcal{C}_2$). Recall that $\|w\|_2 \leq w_{\max}$ for all $w \in \mathcal{C}_2$. For any $\chi > 0$ and $\delta' \in (0, 1)$, with probability at least

$1 - \delta'$,

$$(c) := \overline{w}_{v,t}^T W_t \leq \sqrt{2\sigma^2 \gamma_n \max\left\{\chi, \|\overline{w}_{v,t}\|_{\overline{V}_t}^2\right\} \log\left(\frac{\Gamma_{w_{\max}^2 L^2, \chi}|\mathcal{C}_2|}{\delta'}\right)}.$$

Note that, by definition of $\mathcal{C}_2$, $\|\overline{w}_{v,t}\|_{\overline{V}_t}^2 \geq \sigma_{\min}(\overline{V}_t)\|\overline{w}_{v,t}\|_2^2 \geq \frac{\nu d \epsilon_2^2 \|v\|^2}{\nu + L^2 n} \geq \frac{\nu d^2 \epsilon_2^2 v_{\min}^2}{\nu + L^2 n}$. Hence, setting $\chi \leftarrow \chi'_n := \frac{\nu d^2 \epsilon_2^2 v_{\min}^2}{\nu + L^2 n}$,

$$\Gamma_{w_{\max}^2 L^2, \chi'_n} = 1 + \frac{\log(w_{\max}^2 L^2 n / \chi'_n)}{\log \gamma_n} \leq 1 + \log(w_{\max}^2 L^2 n / \chi'_n) \log(n)$$

where the last inequality is from $\log(1 + \frac{1}{\log n}) \geq \frac{1}{2 \log n}$ for $n \geq 2$. This yields

$$(c) \leq \sqrt{2\sigma^2 \gamma_n \|\overline{w}_{v,t}\|_{\overline{V}_t}^2 \log\left(\frac{(1 + \log(w_{\max}^2 L^2 n / \chi'_n) \log(n))|\mathcal{C}_2|}{\delta'}\right)}.$$

Let us now bound $\|\overline{w}_{v,t}\|_{\overline{V}_t}^2$. We have

$$\|\overline{w}_{v,t}\|_{\overline{V}_t}^2 = \left\|\overline{w}_{v,t} \pm \overline{V}_t^{-1/2} v\right\|_{\overline{V}_t}^2 = \left\|\overline{w}_{v,t} - \overline{V}_t^{-1/2} v\right\|_{\overline{V}_t}^2 + \left\|\overline{V}_t^{-1/2} v\right\|_{\overline{V}_t}^2 + 2\left(\overline{w}_{v,t} - \overline{V}_t^{-1/2} v\right)^T \overline{V}_t \left(\overline{V}_t^{-1/2} v\right)$$

$$\leq \left\|\overline{w}_{v,t} - \overline{V}_t^{-1/2} v\right\|_{\overline{V}_t}^2 + \|v\|_v^2 + 2\left\|\overline{w}_{v,t} - \overline{V}_t^{-1/2} v\right\|_{\overline{V}_t} \|v\|_2$$

$$\leq (\epsilon_2)^2 (1 + \sqrt{d})^2 \|v\|_2^2 + \|v\|_2^2 + 2\epsilon_2 (1 + \sqrt{d}) \|v\|_2^2 = \left(1 + \epsilon_2(1 + \sqrt{d})\right)^2 \|v\|_2^2,$$

where in the last inequality we used the previous bound on $(a) = \left\|\overline{w}_{v,t} - \overline{V}_t^{-1/2} v\right\|_{\overline{V}_t}$.

Putting (a), (b), and (c) together we obtain the following bound on (ii):

$$(ii) = v^T \overline{V}_t^{-1/2} W_t \leq \|v\|_2 \epsilon_2 (1 + \sqrt{d}) \sqrt{2\sigma^2 d \log\left(\frac{1 + \frac{nL^2}{d\nu}}{\delta'}\right)}$$

$$+ \|v\|_2 \left(1 + \epsilon_2(1 + \sqrt{d})\right) \sqrt{2\sigma^2 \gamma_n \log\left(\frac{(1 + \log(w_{\max}^2 L^2 n / \chi'_n) \log(n))|\mathcal{C}_2|}{\delta'}\right)}.$$

If we now set $\epsilon_2 \leftarrow \frac{1}{2d \log n}$, we have $\chi'_n = \frac{\nu v_{\min}^2}{4(\nu + L^2 n)(\log n)^2}$. Setting $\chi''_n = \chi'_n / (w_{\max}^2 L^2)$ and using $w_{\max} = \frac{v_{\max}}{\sqrt{\nu}}\left(1 + \epsilon_2(1 + \sqrt{d})\right) \leq \frac{v_{\max}}{\sqrt{\nu}} \gamma_n$, $\chi''_n \geq \frac{\nu^2 v_{\min}^2}{4L^2(\nu + L^2 n)(\log n)^2 v_{\max}^2 \gamma_n^2} = \chi_n$. Thus,

$$(ii) \leq \|v\|_2 \left(\sqrt{\frac{2\sigma^2 \log\left(\frac{1 + \frac{nL^2}{d\nu}}{\delta'}\right)}{(\log n)^2}} + \sqrt{2\sigma^2 \gamma_n^3 \log\left(\frac{(1 + \log(n/\chi_n) \log(n))}{\delta'}\right) + 2\gamma_n^3 \log|\mathcal{C}_2|}\right).$$

Furthermore, using (77), the log-size of the cover $\mathcal{C}_2$ is

$$\Upsilon_n = \log|\mathcal{C}_2| \leq \log(|\mathcal{C}_1|) + d \log\left(2 + \frac{\log\left(2d \log n \frac{v_{\max}}{v_{\min}} \sqrt{\frac{\nu + L^2 n}{d\nu}}\right)}{\log(1 + \frac{1}{2d \log n})}\right)$$

$$\leq \log(|\mathcal{C}_1|) + d \log\left(2 + 4d \log\left(2d \log n \frac{v_{\max}}{v_{\min}} \sqrt{\frac{\nu + L^2 n}{d\nu}}\right) \log n\right)$$

To conclude the proof, we notice that the derivation above holds uniformly for all $v \in \mathcal{C}_1$ and $t \in [n]$ with probability at least $1 - 2\delta'$ since we applied both Lemma 17 (for term (b) in (ii)) and Lemma 20 (for term (c) in (ii)). Thus, the statement follows by setting $\delta = 2\delta'$. $\qquad\square$

### J.3 Auxiliary Results

**Lemma 17.** *[Lemma 9 of [4]] Let $\tau$ be a stopping time with respect to filtration $\{\mathcal{F}_t\}_{t=1}^{\infty}$ and $W_t := \sum_{s=1}^{t} \phi(X_s, A_s)\xi_s$. Then, for any $\delta \in (0,1)$, with probability at least $1 - \delta'$,*

$$\|W_\tau\|_{\overline{V}_\tau^{-1}} \leq \sqrt{2\sigma^2 \log\left(\frac{\det(\overline{V}_\tau)^{1/2}\nu^{-d/2}}{\delta}\right)} \leq \sqrt{2\sigma^2 d \log\left(\frac{1 + \frac{\tau L^2}{d\nu}}{\delta}\right)}.$$

The following result is a specialization of Lemma 2.6 of [31] or Lemma 4.2 of [32].

**Lemma 18.** *Let $n \in \mathbb{N}$ and $\{Y_t\}_{t=1}^{n}$ be a sequence of sub-Gaussian random variables adapted to filtration $\mathcal{F}$ such that $\mathbb{E}[Y_t|\mathcal{F}_{t-1}] = 0$ and*

$$\forall \zeta \in \mathbb{R} : \mathbb{E}\left[e^{\zeta Y_t}|\mathcal{F}_{t-1}\right] \leq e^{\frac{\zeta^2 \sigma_t^2}{2}},$$

*where $\sigma_t^2 := \mathbb{V}\mathrm{ar}[Y_t|\mathcal{F}_{t-1}]$. Then, for all $\epsilon \geq 0, v > 0$,*

$$\mathbb{P}\left\{\exists t \leq n : \sum_{s=1}^{t} Y_s \geq \epsilon, \sum_{s=1}^{t} \sigma_s^2 \leq v\right\} \leq e^{-\frac{\epsilon^2}{2v}}.$$

*Proof.* The result follows straightforwardly from Lemma 2.6 of [31] or Lemma 4.2 of [32] after optimizing for $\zeta$. $\qquad \square$

**Lemma 19** (Lemma 14 of [7])**.** *Let $n \in \mathbb{N}$ and $\epsilon > 0$. Let $\{Y_t\}_{t=1}^{n}$ be a sequence of Gaussian random variables adapted to filtration $\mathcal{F}$ such that $\mathbb{E}[Y_t|\mathcal{F}_{t-1}] = 0$ and $\mathbb{V}\mathrm{ar}[Y_t|\mathcal{F}_{t-1}] \leq b$ for some $b > 0$. Then*

$$\mathbb{P}\left\{\exists t \leq n : \sum_{s=1}^{t} Y_s \geq \sqrt{2\gamma_n P_t \log\frac{\Gamma_{b,\epsilon}}{\delta}}\right\} \leq \delta,$$

*where $P_t = \max\{\epsilon, \sum_{s=1}^{t} \mathbb{V}\mathrm{ar}[Y_t|\mathcal{F}_{t-1}]\}$, $\gamma_n = 1 + \frac{1}{\log n}$, and $\Gamma_{b,\epsilon} = 1 + \frac{\log(nb/\epsilon)}{\log \gamma_n}$.*

*Proof.* The proof uses the same peeling argument as in [7] but follows different steps.

Let $\tau \leq n$ be a stopping time with respect to $\mathcal{F}$ whose value will be specified later. Define $\Upsilon_t := \sum_{s=1}^{t} \mathbb{V}\mathrm{ar}[Y_t|\mathcal{F}_{t-1}]$ as the sum of predictable variances and $f(v) := \sqrt{2\gamma_n \max\{v, \epsilon\} \log\frac{1}{\delta'}}$. Let us define a sequence of scalars $v_{-1}, v_0, \ldots v_{k_n}$, which will be used to discretize the predictable variances, with $v_{-1} = 0$, $v_0$ to be specified later, $v_j = \gamma_n v_{j-1}$ for $j \geq 1$, and $k_n$ such that $v_{k_n} \geq nb$ (which implies $v_{k_n} \geq \Upsilon_n$). Note that the theorem holds trivially when $\Upsilon_\tau = 0$, so we consider the case where this variable is positive. We have

$$\mathbb{P}\left\{\sum_{s=1}^{\tau} Y_s \geq f(\Upsilon_\tau)\right\} \overset{(a)}{\leq} \sum_{j=0}^{k_n} \mathbb{P}\left\{\sum_{s=1}^{\tau} Y_s \geq f(\Upsilon_\tau), \Upsilon_\tau \in (v_{j-1}, v_j]\right\}$$

$$\overset{(b)}{\leq} \sum_{j=0}^{k_n} \mathbb{P}\left\{\sum_{s=1}^{\tau} Y_s \geq f(v_{j-1}), \Upsilon_\tau \leq v_j\right\} \overset{(c)}{\leq} \sum_{j=0}^{k_n} e^{-\frac{f(v_{j-1})^2}{2v_j}},$$

where (a) uses a union bound, (b) holds since $f$ is non-decreasing, and (c) is from Lemma 18. Using the definition of $\{v_j\}_{j \geq -1}$,

$$\sum_{j=0}^{k_n} e^{-\frac{f(v_{j-1})^2}{2v_j}} = e^{-\frac{\gamma_n \epsilon \log\frac{1}{\delta'}}{v_0}} + \sum_{j=1}^{k_n} e^{-\frac{2\gamma_n \max\{v_j/\gamma_n, \epsilon\} \log\frac{1}{\delta'}}{2v_j}} \leq (\delta')^{\frac{\gamma_n \epsilon}{v_0}} + k_n \delta'.$$

Since $v_{k_n} = \gamma_n^{k_n} v_0$, we have that $k_n = \left\lceil \frac{\log(nb/v_0)}{\log(\gamma_n)} \right\rceil$ suffices to have $v_{k_n} \geq nb$. Setting $v_0 \leftarrow \gamma_n \epsilon$,

$$\mathbb{P}\left\{\sum_{s=1}^{\tau} Y_s > f(\Upsilon_\tau)\right\} \leq \delta'\left(1 + \left\lceil \frac{\log(nb/\epsilon) - \log(\gamma_n)}{\log(\gamma_n)} \right\rceil\right) \leq \delta'\left(1 + \frac{\log(nb/\epsilon)}{\log(\gamma_n)}\right) = \delta'\Gamma_{b,\epsilon}.$$

The result follows by setting $\delta \leftarrow \delta'\Gamma_{b,\epsilon}$ and $\tau \leftarrow \min\left\{t \leq n : \sum_{s=1}^{t} Y_s > \sqrt{2\gamma_n P_t \log\frac{\Gamma_{b,\epsilon}}{\delta}}\right\}$. $\qquad \square$

The following result can be derived using a similar argument as in the proof of Lemma 15 of [7].

**Lemma 20.** *Let $\mathcal{C} \subset \{w \in \mathbb{R}^d : \|w\| \leq b\}$ be a finite set of vectors in $\mathbb{R}^d$ with norm bounded by $b > 0$ and $W_t$ as defined in Lemma 17. Then, for all $\epsilon > 0$ and $\delta \in (0, 1)$,*

$$\mathbb{P}\left\{\exists t \leq n, w \in \mathcal{C} : w^T W_t \geq \sqrt{2\sigma^2 \gamma_n \max\{\epsilon, \|w\|_{\overline{V}_t}^2\} \log\left(\frac{\Gamma_{b^2 L^2, \epsilon} |\mathcal{C}|}{\delta}\right)}\right\} \leq \delta,$$

*where $\gamma_n$ and $\Gamma_{b^2 L^2, \epsilon}$ are those defined in Lemma 19.*

*Proof.* Fix $w \in \mathcal{C}$. Note that

$$\frac{w^T W_t}{\sigma} = \sum_{s=1}^{t} \frac{w^T \phi(X_s, A_s)\xi_s}{\sigma}$$

is a sum of Gaussian random variables adapted to $\mathcal{F}$ such that

$$\mathbb{V}\text{ar}\left[\frac{w^T \phi(X_s, A_s)\xi_s}{\sigma}|\mathcal{F}_{s-1}\right] = \frac{(w^T \phi(X_s, A_s))^2}{\sigma^2} \underbrace{\mathbb{V}\text{ar}[\xi_s|\mathcal{F}_{s-1}]}_{=\sigma^2} \leq \|w\|^2 \|\phi(Y_s, A_s)\|^2 \leq b^2 L^2.$$

Furthermore,

$$\sum_{s=1}^{t} (w^T \phi(Y_s, A_s))^2 = \sum_{s=1}^{t} w^T \phi(Y_s, A_s)\phi(Y_s, A_s)^T w = \|w\|_{V_t}^2 \leq \|w\|_{\overline{V}_t}^2,$$

where the last inequality is from $V_t \preceq \overline{V}_t$. Therefore, using Lemma 19, with probability at least $1 - \delta'$,

$$w^T W_t \leq \sqrt{2\sigma^2 \gamma_n \max\{\epsilon, \|w\|_{\overline{V}_t}^2\} \log \frac{\Gamma_{b^2 L^2, \epsilon}}{\delta'}}.$$

The result follows after taking a union bound over all elements in $\mathcal{C}$. $\square$

## K   Additional Experiments

### K.1   Implementation Details

In our implementation of SOLID, we ignore the projection of the parameters computed by regularized least squares onto $\Theta$. Moreover, we remove the restriction that the alternative parameters should lie in $\Theta$. That is, we use

$$\Theta_{\text{alt}} := \{\theta' \in \mathbb{R}^d \mid \exists x \in \mathcal{X}, \ a_{\theta^\star}^\star(x) \neq a_{\theta'}^\star(x)\}, \tag{78}$$

and similarly for $\overline{\Theta}_t$. In this case, for linear bandits with Gaussian noise, the infimum over alternative models in the constraint of (P) can be computed in closed form as

$$2\sigma^2 \inf_{\theta' \in \Theta_{\text{alt}}} \sum_{x,a} \eta(x, a) d_{x,a}(\theta^\star, \theta') = \inf_{\theta' \in \Theta_{\text{alt}}} \|\theta^\star - \theta'\|_{V_\eta}^2 = \min_{\substack{x \in \mathcal{X}, \\ a \neq a_{\theta^\star}^\star(x)}} \frac{\Delta_{\theta^\star}(x, a)^2}{\|\phi(x, a) - \phi_{\theta^\star}^\star(x)\|_{V_\eta^{-1}}^2}, \tag{79}$$

where $V_\eta = \sum_{x,a} \eta(x, a)\phi(x, a)\phi(x, a)^\mathsf{T}$ and $\phi_{\theta^\star}^\star(x) = \phi(x, a_{\theta^\star}^\star(x))$. The same closed-form can be used for the infimum in the constraint (3). Regarding the exploitation test, we restrict the set of alternative reward parameters to those with "incompatible" optimal arm in the last observed context. That is, we use the test

$$\inf_{\theta' \in \widetilde{\Theta}_{t-1}} \|\widehat{\theta}_{t-1} - \theta'\|_{\overline{V}_{t-1}}^2 > \beta_{t-1}, \tag{80}$$

where $\widetilde{\Theta}_{t-1} = \{\theta' \in \mathbb{R}^d \mid a_{\widehat{\theta}_{t-1}}^\star(X_t) \neq a_{\theta'}^\star(X_t)\}$. Once again, the infimum can be computed in closed form as before (without the minimum over contexts).

## K.2 Experiment Configurations

We provide the detailed configurations of the experiments reported in the main paper. We use the same confidence intervals in all experiments. For SOLID, we set $\beta_t = \sigma^2(\log(t) + d \log \log(n))$ and $\gamma_t = \sigma^2(\log(S_t) + d \log \log(n))$ as prescribed by Thm. 1 (without numerical constants). For OAM, we use the same $\beta_t$ for the exploitation test. For LinUCB, we use the confidence set of [4] without numerical constants. Similarly, we implement LinTS as defined in [5] but without the extra-sampling factor $\sqrt{d}$ used to prove its frequentist regret. All plots are the results of 100 runs with 95% Student's t confidence intervals.

In both experiments, for SOLID we set $\alpha_\omega = 1$, $\alpha_\lambda = 0.5$, and we normalize the gradients by context in $l_2$-norm. We do not reset the optimizer at the beginning of each phase. We use the theoretical exponential schedule for $z_k$ and $p_k$ as defined in Thm. 2. We set $z_0 = 1$, $\lambda_1 = 0$ for the first experiment and $z_0 = |\mathcal{A}|$, $\lambda_1 = 50$ for the second one. The reward noise is $\sigma = 0.5$ in the first experiment and $\sigma = 1$ in the second one.

**Generation of Random Problems**   We adopt the following procedure in order to generate the random bandit models for the second experiment. We first randomly sample a sparse $|\mathcal{X}||\mathcal{A}| \times d$ feature matrix and a sparse vector $\theta^\star$ with entries uniformly distributed in $[0, 1]$. We then compute the resulting optimal arms for each context and check whether they span $\mathbb{R}^d$. If they do, we discard the generated features/parameter and repeat the previous procedure. Otherwise we keep the bandit problem. Discarding problems where the features of the optimal arms span $\mathbb{R}^d$ is done in order to avoid easy bandit problems in which exploration is not necessary (see [16])[17].

## K.3 Parameter Analysis

We provide an empirical study of how different choices for the relevant parameters of SOLID affect the algorithm's performance in the toy problem of Sec. 6. We note that the purpose of this section is to build some intuition on how SOLID behaves with different parameters rather than assessing which configurations are globally better.

We use the two-context toy problem of Sec. 6 with $\xi = 0.1$ and $\sigma^2 = 1$. We study the effect of the following parameters, with corresponding default values.

- $z_0$ (default 30): the initial normalization factor;
- $\lambda_1$ (default 0): the initial multiplier;
- $\alpha^\omega$ (default 0.1): learning rate for $\omega$. We keep it fixed instead of decreasing with the phase length as suggested by the theory;
- $\alpha^\lambda$ (default 0.5): learning rate for $\lambda$. We keep it fixed as for $\alpha^\omega$;
- $z_k, p_k$ (default $z_k = z_0 e^k$, $p_k = z_k e^{2k}$): the schedule for the phase length. We use the one for which we derive regret guarantees by default but we also experiment with other schedules. By default we do not reset the optimizer at the beginning of each phase.

We vary each parameter in a suitable range while keeping all the others fixed to their default values. The results are described in the following paragraphs.

**Changing $z_0$**   As mentioned in the main paper, the initial value of the parameter $z$ controls both the feasibility of the optimization problem and the trade-off between minimizing regret and gathering information about the optimal arms when $t$ is small. While a small value of $z_0$ might lead SOLID to collect a large amount of information, this might bring high finite regret as derived in the regret bound. Fig. 3(left) confirms this claim, where the value $z_0 = 1$ suffers high initial regret but the resulting curve has a better slope.

**Changing $\lambda_1$**   Though the initial multiplier has no particular impact on the regret bound, in practice it induces a behavior similar to $z_0$, where larger values lead SOLID to collect more information about $\theta^\star$ in the very first learning steps (see Fig. 3(right)).

Figure 3: The effect of changing $z_0$ (left) and $\lambda_1$ (right).

Figure 4: The effect of changing $\alpha^\omega$ (left) and $\alpha^\lambda$ (right).

**Changing the step sizes** Fig. 4 shows the effect of varying $\alpha^\omega$ and $\alpha^\lambda$. In this particular case, $\alpha^\lambda$ seems to have no remarkable effect on SOLID's performance. On the other hand, the algorithm is quite sensible to the choice of $\alpha^\omega$, with very small values performing poorly since the policy is updated rarely and remains close to uniform for a long time. More aggressive step sizes seem to yield the best performance.

**Phase schedule** We test different schedules for $z_k$ and $p_k$ with respect to the one prescribed by the theory. We have $z_k = z_0 e^k, p_k = z_k e^{2k}$ (exp-exp), $z_k = z_0(1+k), p_k = z_k e^k$ (lin-exp), $z_k = z_0(1+k), p_k = z_k(1+k)^2$ (lin-pol), and $z_k = z_0(1+k), p_k = z_k(1+k)$ (lin-lin). Fig. 5(left) shows the result (here we set $z_0 = 1$ to better highlight the contribution of the different schedules). The exponential schedules are as expected more conservative since the algorithm spends more time optimizing with small values of $z$ (i.e., seeks more information). The linear and polynomial schedules behave, on the other hand, more greedily and suffer less regret, though the resulting curve has larger slope.

We also test the effect of resetting the optimizer (middle and right plots in Fig. 5). We see that resetting the optimizer does not significantly affect the algorithm's performance both in case $z = 1$ and $z = 30$. This is likely due to the fact that phases are long (thanks to the exponential schedule) and that the algorithm spends many steps in the exploit phase, where no optimization is performed.

**Tracking** We compare the sampling strategy adopted by SOLID with the popular direct and cumulative tracking rules. Interestingly, Fig. 6(left) shows that sampling from $\omega$ constitutes a nice trade-off between cumulative tracking and the more aggressive direct tracking. Note that, while our theoretical results can be easily derived for cumulative tracking, we do not know whether the same can be done for direct tracking.

Figure 5: Different phase schedules (left) and effect of resetting the optimizer (middle and right plots).

Figure 6: Different tracking strategies (left) and comparison with the exploitation test used in OAM.

**Exploitation test** We note that the test performed by SOLID in order to decide whether to explore or exploit is slightly different from the one adopted in OAM. In fact, the closed-form of the infimum over the alternative set (Eq. 79) leads to terms of the form $\Delta_{\widehat{\theta}_t}(x,a)^2/\|\phi(x,a) - \phi(x)^\star\|^2_{\overline{V}_t^{-1}}$ while OAM uses $\Delta_{\widehat{\theta}_t}(x,a)^2/\|\phi(x,a)\|^2_{\overline{V}_t^{-1}}$. We verify empirically (Fig. 6(right)) that the two tests lead to very similar performance.

## K.4 Real Dataset

We report additional results on real data. We use the Jester Dataset [33] which consists of joke ratings in a continuous range from $-10$ to $10$ for a total of $100$ jokes and $73421$ users. We select a subset of $40$ jokes and $19181$ users rating all these $40$ jokes.

We build a linear contextual problem as follows. We first extract separate 36-dimensional user (context) and joke (arm) features via a low-rank matrix factorization. Then, we concatenate these user and joke features (thus obtaining vectors with 72 entries) and fit a $64 \times 64$ neural-network with ReLU non-linearities to predict the ratings of a random subset of $75\%$ of the users, using these feature vectors as inputs. We obtain $R^2 \simeq 0.95$ on the remaining $25\%$ users. Finally, we take the features extracted in the last layer of the network as the features for our bandit problem and the parameters of the same layer as $\theta^\star$. Rewards in our bandit problem are generated from this linear model by perturbing the prediction with $\mathcal{N}(0, 0.5^2)$ noise. We thus obtain a problem with $d = 65$ (the $64$ hidden neurons plus the bias term), $40$ arms (the jokes), and a total of $19181$ users.

We run the algorithms for $2 \cdot 10^6$ steps, with each run randomizing a subset of $1\%$ of the total users (hence $|\mathcal{X}| = 191$) and using all $40$ arms. For SOLID, we use the same parameters as in the experiment with random models. Due to the computational bottleneck demonstrated in the previous experiments, we could not run OAM on this problem. The results are shown in Figure 7 and confirm that SOLID achieves superior performance than the other baselines.

Figure 7: Experiment on a real dataset (Jester).