[Reviews · NeurIPS 2020]

Review 1

Summary and Contributions: The paper studies algorithms for the finite-armed linear bandit problem under which the leading order term of the asymptotic growth rate of regret is optimal. The asymptotic limits of these problems seems to be well understood by older work of Lai and Graves but has attracted renewed interest. The paper aims to improve previous asymptotically algorithms and analysis in settings with smaller time horizon. To do this, they reformulate the lower bound to better account for contexts with small probabilities, they use a primal dual algorithm based on UCBs and Lagrangian relaxation of the asymptotic lower bound, and they show the constant terms in regret scale at most linearly with problem parameters like the number of contexts.

Strengths: While the problem's large-deviation-style asymptotics are well understood, its difficult to translate those into practical algorithms and performance. This paper takes a step in that direction. Writing this paper took real technical mastery. It is quite complex and creatively combines many different ideas from the bandit and online optimization literatures.

Weaknesses: From a purely theoretical perspective, the paper provides a bound on the lower order asymptotic terms in an asymptotically optimal regret upper bound. But, it's not clear if these bounds are practically meaningful. First, the asymptotic bounds are pretty misleading for most linear bandit examples. The leading order term depends inversely on the smallest possible gaps between arms' mean rewards, which is often extremely small. Second, the bounds on the "lower order" terms are only smaller than the leading order term when log(n) exceeds the |X|d, meaning when the horizon is exponentially large in the number of contexts times the model dimension. The algorithm seems more practical than some previous ones that were asymptotically optimal in the same sense. But the experiments don't make a serious attempt to judge whether they could supplant e.g. Thompson sampling or a tuned UCB algorithm. The algorithm is inherently based around identifying an exactly optimal policy, even if other policies are extremely similar. I worry it could perform very poorly when several policies are statistically indistinguishable (which is very common).

Correctness: I don't see an clear issues with correctness, but the full paper consists of 49 extremely technical pages. Realistically, if accepted, the paper is being accepted without having being 'verified' in peer review.

Clarity: The body of the paper is relatively well written, though there is a way more material crammed in than fits in an eight page paper.

Relation to Prior Work: The paper should credit, from the very beginning, works like: Asymptotically Efficient Adaptive Choice of Control Laws in Controlled Markov Chains by Graves and Lai or Asymptotically efficient adaptive allocation schemes for controlled iid processes: Finite parameter space by A Rajeev, D Teneketzis, V Anantharam. Or even earlier work in experimental design that gives similar lower bounds. It should be clear, from the beginning, that the main ideas underlying asymptotic optimality are quite old are not a recent development in ML.

Reproducibility: Yes

Additional Feedback: Please clarify what parameters you used for LinUCB and LinTS? Are these the extremely conservative versions for which regret bounds exist, or the versions with more realistic parameters that are implemented in practice?


Review 2

Summary and Contributions: The paper proposes a novel algorithm for linear contextual bandits (with contexts in a finite support and finite number of actions). The paper is both asymptotically optimal and computationally efficient. The authors present as well the finite-time guarantee of the algorithm. Experiments, as well as theoretical statements, support this finding.

Strengths: The authors make a significant improvement on previous works (on constructing bandit algorithms with asymptotically optimal regret guarantee) by combining various techniques: - introducing a constraint in the optimization problem to guarantee bounded number of pulls - reformulating the optimization problem to better adapt to the context distribution - reformulating the Lagrangian dual formulation to an optimistic version, which removes requirement of wasteful forced exploration - introducing a new incremental learning method for scalability Each technique is well motivated, and described in a clear manner.

Weaknesses: -The paper seems to be much inspired by the work of Degenne et al. (2020). The authors do not explicitly state how their contributions differ from this work in the main body of the paper.

Correctness: The claims are clear. The empirical methodology is correct.

Clarity: The paper is well written and the exposition is clear. The contributions as well as how these were made are described in a clear manner.

Relation to Prior Work: The authors extensively review the prior works that address the same problem. The proposed method overcomes weakness of prior works in various aspects (computational efficiency, empirical performance, dependency on forced exploration and number of arms, dependency on the distribution of contexts)

Reproducibility: Yes

Additional Feedback: -In Preliminaries, the authors state that the Gaussian assumption on the errors can be relaxed to the sub-Gaussian assumption. It would be helpful know how the theoretical statements would change under sub-Gaussian assumption. (For example, can we still use the KL divergence formula as it is?) ==== Comments after rebuttal ==== Thank you for a detailed answer to my question. I also agree that an algorithm with low, asymptotic problem-dependent regret bound can be more practical than an algorithm with low, finite-time worst case regret bound.


Review 3

Summary and Contributions: Provided an asymptotically optimal algorithm for linear contextual bandits

Strengths: primal-dual algorithm applied to linear contextual bandits seems novel to me

Weaknesses: finite-time regret bound--the most important metric--is not good enough

Correctness: yes

Clarity: reasonable

Relation to Prior Work: yes Post rebuttal: I had read the authors' response and had decided to keep the current score.

Reproducibility: Yes

Additional Feedback: I read the paper with interest, but got a bit disappointed in the end. Asymptotic optimality seems to be the focus of the paper, and this is the point I disagree with. Certainly, having asymptotic optimality is good, but only performing well on that--rather than finite-time optimality--is not enough given that linear contextual bandits have been studied extensively. In particular, a simple epsilon-greedy algorithm with epsilon decreasing to 0 at an appropriate rate is already asymptotically optimal. So in my view, finite-time regret must be the clear performance metric for evaluating an algorithm. Additionally, the finite-time regret bound given in Theorem 2 is not very good. In particular, it has a dependence on the number of contexts: in the normal linear contextual bandits, there are infinite (and even uncountably infinite) contexts and hence the bound is vacuous except in the simplest setting. The authors need to take a look at "Contextual Bandits with Linear Payoff Functions/Chu, Li, Reyzin and Schapire, 2011", one of the pioneering papers (missing in the references too) that gave finite-time regret bound and established a \sqrt{dT} regret without any dependence on the number of contexts.


Review 4

Summary and Contributions: This papers presents an asymptotically optimal algorithm for Linear Contextual Bandits which improves performances compared to recent algorithms. The algorithms is based on a primal dual incremental implementation of a solution of the relaxed lower bound optimization problem. A finite time regret guarantee as well as empirical evaluations are provided. The authors use a new concentration inequality on the performance of the estimated regression parameter that can be of independent interest for the community.

Strengths: This paper improves recent algorithms on several aspects: - the context distribution is decoupled from the exploration policy, the regret bound is thus independent of the context distribution and in particular its minimum probability - allows a direct sampling from the exploration strategy A finite time regret bound is provided showing that the algorithm is asymptotically optimal and empirical evaluations show the improvement in performance compared to other recent algorithms. Finally the authors derive a new concentration inequality for the estimated regression parameters that can be of independent interest for the community.

Weaknesses: None as far I understand the main tools of the algorithm and the main results.

Correctness: I did not check the proofs of the main results

Clarity: I ernjoy reading the paper. It is well written and it is easy to understand the main tools and the main arguments for the suggested algorithm.

Relation to Prior Work: The authors clearly discussed the differences with previous contributions.

Reproducibility: Yes

Additional Feedback: l 15 : please give the chapter/section number when citing [1] l 120 : by -> be

[Author Response · NeurIPS 2020]

**1 We would like to thank the reviewers for their valuable feedback.**

[R1, R3] Asymptotic problem-dependent vs finite-time worst-case regret. We agree that finite-time regret is the
performance measure of interest. At the same time, problem-dependent optimality is a stronger notion than worst-case
optimality, as it requires an algorithm to perform optimally in *every single* instance. Linear contextual bandits have been
studied extensively from a *worst-case* perspective and minimax optimal strategies exist. These strategies are robust
to worst-case instances, but in general they fail to adapt to the structure of the problem (e.g., they ignore informative
arms) and may perform poorly in practice (see e.g., [7] and our experiment in Fig.1). In recent years, several attempts
have been made to design more adaptive algorithms by leveraging *asymptotic lower bounds*, which effectively capture
all problem-specific characteristics (e.g., set of arms, possible constraints on the parameters, reward distribution) into
the regret bound. While the resulting algorithms directly inherit asymptotic optimality, the question is whether their
more problem-adaptive behavior translates to competitive finite-time performance w.r.t. worst-case optimal algorithms.
While this is the case for best-arm identification (see e.g., "Explicit Best Arm Identification in Linear Bandits Using
No-Regret Learners" [Zaki et al., 2020] and "Gamification of Pure Exploration for Linear Bandits" [Degenne et al.,
2020]), it still remained as an open question for regret minimization in linear bandit, where algorithms like OSSB and
OAM have several practical limitations and they are rarely preferable over LinUCB or LinTS. We believe our paper is
a significant step forward in addressing this question: SOLID resolves most of the issues of existing asymptotically
optimal algorithms, significantly improves their finite-time regret guarantees, and it is shown to be empirically better
than *practical* versions of LinUCB and LinTS in a variety of settings (including real data, see App. K.3).

[R1, R3] Dependence on contexts. We significantly improved the regret guarantees w.r.t. [14] by removing any
dependency on $1/\rho_{\min}$ (which is at least as large as $|\mathcal{X}|$). Yet, we conjecture the dependence on $|\mathcal{X}|$ could be improved
further. SOLID optimizes and updates a context-arm exploration strategy and this may suggest a polynomial dependency
on the size of the exploration strategy is unavoidable. Nonetheless, we managed to push the dependency on the number
of arms to a logarithmic term (greatly improving previous results) and a similar approach could be used for contexts by
avoiding concentrating $\hat{\rho}$ to $\rho$ (see e.g., Lemma 11), which is currently the main source of dependency on $|\mathcal{X}|$. This
conjecture is also supported by empirical evidence. On the Jester dataset in App. K.3, SOLID's performance is not
significantly affected by the number of contexts (almost 200) and it still performs better than LinUCB/LinTS. We will
run additional experiments on Jester for different values of $|\mathcal{X}|$ to further investigate the dependency.

[all] Non-contextual case. We would like to bring to the reviewers' attention that while the paper is framed in the
general contextual case, our contribution should also be assessed in the simpler and yet significant non-contextual case.
Our algorithm and analysis resolve many open questions in this setting, including the dependence on $|\mathcal{A}|$, the derivation
of confidence sets over parameters with optimal asymptotic scaling (Thm. 1), and the efficient incremental computation
of the lower bound. After the submission, we have also analyzed the *finite-time worst-case* properties of SOLID when
$|\mathcal{X}| = 1$. In this case, a simple proof following almost directly from the proof of Thm. 2, we derived an $\tilde{\mathcal{O}}(\sqrt{dn})$ regret
bound that holds for *any horizon* $n$. This implies that SOLID is the first algorithm that is both *finite-time minimax*
*optimal* and *asymptotically problem-dependent optimal* for linear (non-contextual) bandits.

[R1] Experiments. We ran the *practical* version of LinUCB/LinTS, using confidence sets without numerical constants,
with log-determinant of the design matrix for LinUCB, and without the *theoretical* oversampling $\sqrt{d}$-factor for LinTS.

[R1] "The leading order term depends inversely on the smallest possible gaps between arms' mean rewards". This is not
the case in problems with structure (e.g., linear). Examples like the one of the first experiment (which extends the one
in [7]) show that there exist problems in which one can make some arm gap arbitrarily small, yet the optimal regret rate
$v^{\star}(\theta^{\star})$ does not scale with it since pulls are allocated to other informative arms. As a result, the algorithm's behavior
and performance are not negatively affected by the existence of policies that are extremely similar to the optimal one.

[R2] Sub-Gaussian assumption. The lower-bound remains the same, except that the KL divergence needs to be
computed for some distribution in the sub-Gaussian family. The analysis would be almost identical thanks to the
Lipschitz property of KL divergences between sub-Gaussian distributions (see, e.g., [15]) and the results would be the
same with a distribution-dependent Lipschitz constant in lower order terms.

[R3] "a simple epsilon-greedy algorithm [...] is already asymptotically optimal". An algorithm is asymptotically optimal
if its regret scales as $\log(n)$ with a leading problem-dependent constant matching the $v^{\star}(\theta^{\star})$ in the lower bound. This
is very important in practice because it certifies that the algorithm effectively adapts to the problem's structure. In
this sense, an epsilon-greedy algorithm is far from being asymptotically optimal as it only recovers a $O(\log n)$ regret
possibly with a prohibitively large constant (e.g., scaling linearly with the number of context-arms or inverse of the
gaps, which can be extremely small). This directly translates into a poor performance in practice.

[R3] "The authors need to take a look at Chu et al.". We cite [4], which refine the original results ofChu et al. [2011].

[R4] Thanks for the supportive review and for the suggested corrections. We have already updated the paper accordingly.

[Meta-Review · NeurIPS 2020]

Four knowledgeable reviewers refereed the paper. Several issues were raised, and not all reviewers recommended acceptance. The first points that was brought up in several reviews is the motivation for looking at asymptotically optimal algorithms, and how much these are relevant in practice. The authors successfully addressed this in the rebuttal, and I recommend that this answer is taken as the starting point to reinforce the motivation part of the paper. Futher, reviewer #3 raised the missing comparison with Chu, Li, Reyzin and Schapire, 2011 as a major objection. However, this is in my opinion successfully addressed by the rebuttal through a chain of citations and by pointing out that past bounds are weaker than the reviewer believed. I am therefore lowering the confidence on the low score of reviewer 3. What remains is that the paper extends a series of works proving instance-optimal regret guarantees based on information theoretic lower bounds, now to address the contextual bandit setting, providing finite-time guarantees. This is therefore a worthwhile contribution, passing the acceptance threshold. I strongly encourage the authors to integrate the discussion points from the rebuttal (relation asymptotic optimality and practicality, comparing the dependence of the new and previous bounds in their various problem parameter dependences).